# Large-scale testing of antimicrobial lethality at single-cell resolution predicts mycobacterial infection outcomes

In vitro antibiotic testing is important for guiding therapy and drug development. Current methods are focused on growth inhibition in bulk bacterial populations but often fail to accurately predict treatment responses. Here we introduce Antimicrobial Single-Cell Testing (ASCT), a large-scale live-cell imaging approach that quantifies bacterial killing in real time at single-cell resolution. By tracking over 140 million mycobacteria and analysing ~20,000 time–kill curves, we identify key determinants of antibiotic killing and its clinical relevance. For *Mycobacterium tuberculosis*, we found that drug-specific killing dynamics in starved bacteria, rather than growth inhibition or killing of growing cells, predict regimen efficacy in mice and humans. Extending this approach to *Mycobacterium abscessus* and comparing 405 bacterial strains, we show that antibiotic killing is also a genetically encoded bacterial trait (drug tolerance). We demonstrate that tolerance patterns cluster by antibiotic targets, identify a phage protein that modulates antibiotic killing, and show that strain-specific killing dynamics are associated with individual patient outcomes independent of drug resistance. Together, these findings establish a framework that reveals how drug properties and bacterial diversity shape treatment responses, offering a path to more effective and personalized therapies.

Bacteria have evolved diverse strategies to overcome toxic exposures. The best-known is antibiotic resistance, where genetic modifications alter the target site or change the effective drug concentration at the target, enabling bacterial growth during drug treatment[1]. Another bacterial strategy involves transient bacterial survival during antibiotic treatment, allowing bacterial regrowth once the drug is cleared. This phenomenon is known as drug tolerance, referring to delayed killing of the overall population, and persistence, when a highly tolerant subpopulation is present[2]. Despite decades of recognition and the identification of multiple tolerance mechanisms[3–6], key aspects of tolerance biology, especially in human infections, and its impact on antibiotic treatment remain poorly understood[7].

Even in the absence of antibiotic resistance, treatment outcomes for bacterial infections, including urinary tract, respiratory and bloodstream infections, are often poor[8–11]. This challenge is highlighted in mycobacterial infections, where limited treatment efficacy necessitates prolonged multidrug therapy lasting months to years. For drug-susceptible *Mycobacterium tuberculosis*, which caused more than 1 million deaths in 2023[12], it took decades of research to shorten treatment from 6 to 4 months[13]. These extended treatment durations are costly, associated with treatment-related toxicity and increase the risk of non-adherence, potentially driving relapses and transmission. The therapeutic challenge is even more acute in *Mycobacterium abscessus* infections, now one of the most prevalent mycobacterial infections in developed countries[14]. For *M. abscessus*, no consistently effective drug regimen exists, and treatment outcomes are often poor, with failure rates frequently exceeding 50% despite months or years of treatment with multiple in vitro active drugs[15].

✉e-mail: lucas.boeck@unibas.ch

This gap between in vitro growth inhibition and in vivo efficacy motivated us to develop strategies beyond standard susceptibility testing to better predict treatment outcomes. Bacterial killing could be such a factor, which is not captured by minimum inhibitory concentrations (MICs) and has been challenging to study due to the labour-intensive, low-throughput nature of colony-forming unit (c.f.u.) assays[16–18]. To overcome this gap, we established Antimicrobial Single-Cell Testing (ASCT), a highly scalable method for quantifying bacterial viability at single-cell resolution. We first applied ASCT to *Mycobacterium tuberculosis* to examine whether the killing properties of different drug regimens are linked to in vivo treatment responses in mouse and human populations. Extending this approach to *Mycobacterium abscessus*, we then compared hundreds of clinical isolates to determine how bacterial strain variation shapes antibiotic killing dynamics and individual patient outcomes.

## ASCT captures bacterial killing at large scale

To quantify bacterial growth and antibiotic killing in bacterial populations and subpopulations at a scale comparable to that of MIC assessments, we developed ASCT (Fig. 1 and Supplementary Video 1), a workflow that simultaneously tracks the behaviour and viability of millions of single cells across diverse conditions in 1,536-well plates. In ASCT, bacteria are dispensed into multiwell plates, immobilized in agar pads containing the viability dye propidium iodide (PI) and exposed to drugs (Fig. 1a and Methods). High-content live-cell imaging captures brightfield and fluorescence images of over 10,000 fields every 2–4 h for up to 7 days, generating up to one million images per experiment. Time-lapse images of every field are sequentially analysed using sparse and low-rank decomposition to correct for background fluorescence[19], supervised random forest classifiers for bacterial segmentation and viability classification[20], drift correction and single-cell tracking based on object position and homology (Fig. 1b and Extended Data Fig. 1a–e). To analyse time–kill kinetics at the population level, single-cell data are pooled, and overall killing is quantified using the area under the time–kill curve (AUC; Fig. 1c). The AUC captures drug tolerance, representing the killing dynamics of the whole population, and persistence, reflecting the enhanced survival of smaller bacterial populations. ASCT showed no detectable effect on bacterial viability in antibiotic-free conditions (Extended Data Fig. 1f). *M. abscessus* time–kill kinetics were reproducible, drug- and concentration-specific, and largely consistent across gel volumes and concentrations (Fig. 1d,e and Extended Data Fig. 1g,h). Although ASCT can also quantify homogeneous and heterogenous bacterial growth (for example, resistance or hetero-resistance), single-cell tracking of growing cells is constrained by the small well size and overlapping bacteria. This study therefore focuses on the determinants and consequences of antibiotic killing.

## PI provides a single-cell readout of mycobacterial viability

Bacterial viability was assessed using PI, a cell-impermeable fluorescent dye that accumulates in cells upon membrane disruption and serves as a proxy for bacterial death, as previously demonstrated in multiple bacterial species[21–23]. To validate this method, we treated *M. abscessus* with a number of antibiotics for 24 h and tracked more than 30,000 single bacteria for a further 24 h post antibiotic washout. Only one PI-positive bacterium resumed growth (a misclassified doublet of a live and dead bacterium), while 1–11% of the PI-negative bacteria regrew in antibiotic-free media (Extended Data Fig. 2a and Supplementary Video 2), confirming that PI accumulation is a reliable marker of bacterial death.

To further assess the viability of PI-negative bacteria, we used an inducible *M. abscessus* reporter strain to test their capacity to synthesize new proteins. After 12 h of antibiotic exposure, drugs were removed and GFP expression was induced with anhydrotetracycline. Following cefoxitin or moxifloxacin treatment, most PI-negative bacteria expressed GFP shortly after induction (Fig. 1f,g and Extended Data Fig. 2b,c), indicating that they remained viable. Bacteria treated with amikacin, a direct inhibitor of protein synthesis, showed delayed GFP expression after washout. Importantly, the fraction of bacteria capable of GFP synthesis far exceeded the fraction of regrowing bacteria (4.2% for cefoxitin, 2.7% for moxifloxacin and 3.4% for amikacin), demonstrating that regrowth assays underestimate survival and that most metabolically competent cells fail to regrow in standard media conditions. Moreover, repeated assessments after antibiotic removal revealed a progressively increasing PI-positive population (Extended Data Fig. 2c and Supplementary Video 2), consistent with substantial post-exposure killing and explaining some of the mismatch between real-time ASCT and delayed c.f.u. measures[24].

## Time–kill kinetics predict in vivo outcomes of *M. tuberculosis* drug regimens

Using ASCT, we first investigated whether the drug's bactericidal properties correspond to in vivo outcomes in mice and humans. Specifically, we analysed the potential of ASCT to identify more effective treatments for tuberculosis. We tested antibiotic killing of 65 drug regimens in the two avirulent *M. tuberculosis* strains H37Ra and mc²7000, in nutrient-rich and starvation conditions. Each drug was dosed at the maximum blood concentration ($C_{max}$) achievable during therapeutic dosing in humans. Drug regimens containing isoniazid, rifampicin or ethambutol (all part of the standard *M. tuberculosis* regimen) were most effective in killing growing *M. tuberculosis* (Extended Data Fig. 3a–c). In contrast, combinations including bedaquiline, clofazimine or pretomanid, were more effective under starvation conditions, consistent with previous reports that these drugs retain activity in non-replicating *M. tuberculosis* (Fig. 2a,b and Extended Data Fig. 3d), and with clinical data demonstrating treatment-shortening potential of bedaquiline- and pretomanid-containing regimens[25–29]. Their efficacy during starvation is probably due to a disruption of energy metabolism, respiration or redox balance, which are essential processes for survival in nutrient-limited environments.

To evaluate ASCT's ability to predict infection outcomes, we used established classifications of *M. tuberculosis* drug regimen efficacy, categorizing them as either similar to standard-of-care or better than SOC, on the basis of mouse and clinical studies[30,31]. While time–kill kinetics of growing *M. tuberculosis* did not discriminate between similar-to-SOC and better-than-SOC drug combinations, we found strong associations between in vivo outcomes and in vitro killing under starvation conditions, independent of whether quantified by AUC or live-cell fractions at single timepoints (Fig. 2c, Extended Data Fig. 4a and Supplementary Table 1). These findings suggest that non-growing, metabolically less active bacteria, commonly found in intra- and extracellular infection niches, are more challenging to clear and probably responsible for long-term infection outcomes[32–35].

For example, killing of pantothenate-starved mc²7000 (the auxotrophic *M. tuberculosis* strain H37Rv ΔpanCD ΔRD1) was highly associated with outcomes in multiple mouse models, including the relapsing mouse model, the bactericidal model of common mouse strains and the C3HeB/FeJ strain, mirroring different outcomes (bacteriologic burden and relapses) and host pathobiology[36,37]. Moreover, in vitro mc²7000 killing upon pantothenate starvation was associated with clinical outcomes in phase 2 clinical studies, as measured by bactericidal activity during treatment[31]. Similar results were obtained with 14-day phosphate-buffered saline (PBS) starvation of the mc²7000 and H37Ra *M. tuberculosis* strains. Overall, 11 out of 12 associations between *M. tuberculosis* killing under starvation and clinical outcomes reached statistical significance.

In contrast, c.f.u.-based killing assessments were not associated with in vivo outcome measures (Extended Data Fig. 5a–c), and correlations of the lowest MIC, median MIC or mean MIC (mean MIC of the two most potent drugs) within each combination with in vivo outcome

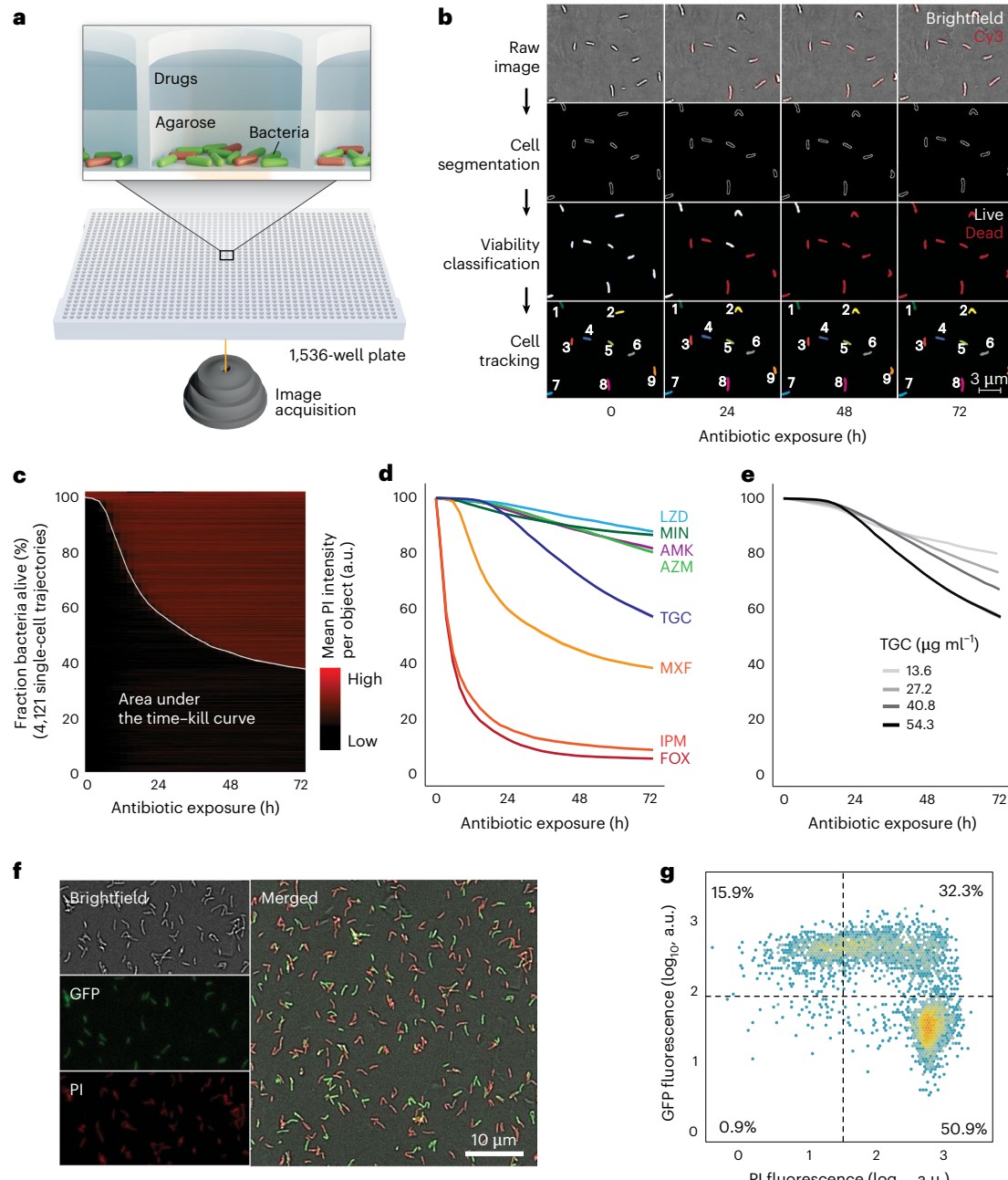

**Fig. 1 | ASCT enables high-throughput assessments of bacterial viability.**
**a**, Illustration of the ASCT setup. Bacteria (dead cells in red) are immobilized in agar pads containing the viability dye PI at the bottom of wells of a 1,536-well plate and exposed to drugs. Live-cell imaging (brightfield and fluorescence) of 9 fields per well and up to 1,536 wells was performed at 2–4-h intervals from the plate bottom. **b**, Schematic of the automated image analysis. Raw brightfield images (to assess cell morphology) and fluorescence images (Cy3 settings, to quantify PI accumulation) were processed to segment cells (lines indicate cell borders), classify bacterial viability (white, viable; red, dead) and track individual cells (track IDs indicated by numbers). **c**–**e**, ASCT-derived time–kill curves of *M. abscessus* (isolate number 328). **c**, Example of 4,212 single-cell trajectories during moxifloxacin exposure from a single imaging well, with mean PI intensity over time highlighted. Bacterial trajectories are ordered by the time of PI positivity.

The white line represents the proportion of live bacteria across 29 timepoints and reveals population time–kill kinetics. The AUC quantifies overall killing, integrating drug tolerance and persistence phenotypes. Time–kill curves (mean ± s.e.m. of three replicates; s.e.m. barely exceeds the line thickness) across multiple antibiotics (**d**) and tigecycline concentrations (**e**). AMK, amikacin; AZM, azithromycin; FOX, cefoxitin; IPM, imipenem; LZD, linezolid; MIN, minocycline; MXF, moxifloxacin; TGC, tigecycline. The number of tracked bacteria per imaging well was 4,246 ± 1,227 (mean ± s.d.). **f**,**g**, Green fluorescent protein (GFP) induction after 12-h cefoxitin treatment and washout with PI fluorescence. **f**, Representative brightfield, GFP and PI fluorescence images with merged overlays acquired -1 h after antibiotic removal and GFP induction. The experiment was repeated twice with similar results. **g**, Corresponding single-cell fluorescence intensities, with gating of GFP- and PI-positive and negative populations.

measures were poor (Fig. 2d and Extended Data Fig. 6). Because MICs within combinations may shift due to drug–drug interactions, we systematically measured all possible pairwise combinations among the 13 drugs tested (9 × 9 concentration matrices in quadruplicate for each of the 78 drug pairs, -25,000 assays in total; Extended Data Fig. 7a).

From these data we predicted high-order drug effects and compared them with Bliss expectations derived from single-drug activities[38]. Interactions were generally weak and not associated with in vivo outcomes, except for clinical trial data where antagonism correlated with regimens classified as better-than-SOC (Extended Data Fig. 7b,c).

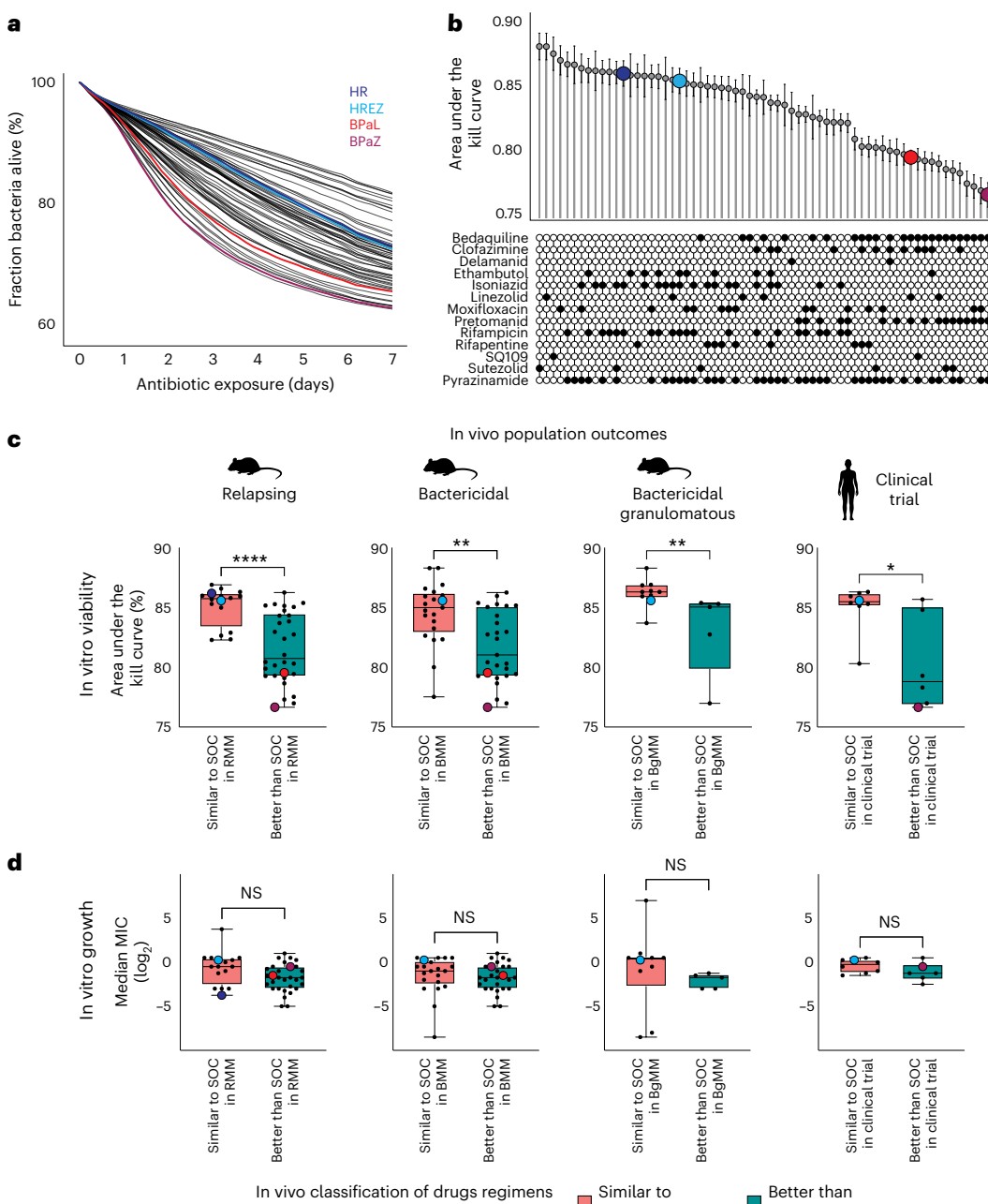

**Fig. 2 | Time–kill kinetics predict in vivo outcomes of *M. tuberculosis* drug regimens. a**, ASCT-based time–kill kinetics of PBS-starved *M. tuberculosis* (mc²7000) exposed to 65 drug regimens, with the following regimens highlighted: isoniazid-rifampicin (HR, dark blue), isoniazid-rifampicin-ethambutol-pyrazinamide (HREZ, light blue), bedaquiline-pretomanid-linezolid (BPaL, red) and bedaquiline-pretomanid-pyrazinamide (BPaZ, purple). **b**, Areas under the kill curve for *M. tuberculosis* drug regimens, averaged across three *M. tuberculosis* starvation conditions (mean ± s.e.m.), with individual drugs indicated. **c**, Drug regimens were previously classified as similar to standard-of-care (SOC) or better than SOC on the basis of their performance in relapsing mouse models (RMM), bactericidal mouse models (BMM) of common mouse strains, the granulomatous C3HeB/FeJ mouse strain and in clinical studies[30,31].

Time–kill curves (AUC averaged across three *M. tuberculosis* starvation models) of similar-to-SOC drug regimens were compared with better-than-SOC regimens using a two-sided Mann–Whitney *U*-test. Each dot represents a drug regimen (colours indicate drug regimens in **a**). Boxplots show the median, interquartile range and total range (central line, box and whiskers, respectively) of AUC values (RMM: $n = 46$, $P = 2.7 \times 10^{-5}$; BMM common strains: $n = 48$, $P = 0.0015$; BMM C3HeB/FeJ: $n = 15$, $P = 0.0047$; clinical bactericidal activity: $n = 14$, $P = 0.020$). *$P < 0.05$, **$P < 0.01$, ****$P < 0.0001$; NS, not significant. **d**, Median MICs of drug regimens, calculated from the MICs of individual drugs (averaged from mc²7000 and H37Ra strains), were compared between similar-to-SOC and better-than-SOC regimens. Sample sizes, statistical tests and box plots are identical to those described in **c**. BgMM, BMM C3HeB/FeJ.

Logistic regression models combining MIC with interactions metrics showed only moderate predictive performance in the relapsing mouse model and no predictive value in the bactericidal mouse model (Extended Data Fig. 7d). Notably, all combinations contained at least one drug administered at concentrations above the MIC, highlighting the value of MICs for identifying active agents, even though regimens with very low MICs did not translate into improved outcomes. To complement ASCT, we also performed population-level resazurin assays as an alternative measure of antibiotic killing. As expected for a metabolic readout, starved *M. tuberculosis* showed lower fluorescence levels than growing cultures. Across all regimens, relative fluorescence changes correlated with ASCT-derived killing only under starvation,

where modest associations with in vivo outcomes were also observed (Supplementary Fig. 1a,b).

The performance of ASCT in predicting clinical study outcomes using *M. tuberculosis* starvation models yielded areas under the receiver operating curve (AUC-ROC) between 0.76 and 0.94 (Extended Data Fig. 4b), suggesting that the killing activity of *M. tuberculosis* regimens upon starvation is a critical in vitro marker that corresponds to in vivo efficacy. These analyses, based on non-clinical *M. tuberculosis* strains, show that time–kill dynamics in starved conditions can quantify regimen performance at the population level. While these findings highlight the potential of ASCT to prioritize antibiotic treatments in development, they do not extend to predicting individual patient outcomes.

## Drug tolerance is a distinct measure of antibiotic efficacy

To investigate bacterial factors that influence antibiotic killing and individual infection outcomes, we next focused on different bacterial isolates. Due to the biosafety constraints of *M. tuberculosis* live-cell imaging, we studied *M. abscessus*, an emerging concern in developed countries. To evaluate how phenotypic and genetic diversity within a naturally evolved bacterial species correlate with antibiotic killing (that is, drug tolerance), we obtained a single bacterial isolate from each of 405 patients with *M. abscessus* lung infection across Europe and Australia[39,40] (Fig. 3a). Using ASCT, we generated time–kill profiles for each clinical isolate and the *M. abscessus* laboratory strain against 8 antibiotics that are commonly used and recommended in clinical treatment[41]. Each drug was tested at 2 concentrations, and each condition was assessed in triplicate. In total, we assessed 18,244 time–kill curves and tracked ~130 million bacterial cells over 29 timepoints and 72 h. Because all clinical *M. abscessus* isolates of a given antibiotic condition were assessed simultaneously, they were frozen in logarithmic phase and exposed to drugs shortly after thawing, when they were probably in a lag phase. Isolate–drug pairs that exhibited bacterial growth (that is, drug resistance) or did not meet quality criteria were excluded from analyses (Methods). Our findings revealed highly diverse but reproducible time–kill kinetics under identical drug exposure, indicating that specific strain characteristics modulate antibiotic killing (Fig. 3b).

Drug tolerance and persistence have been linked to slow bacterial growth or prolonged lag times, associated with metabolic dormancy[42,43]. To evaluate how bacterial replication affects antibiotic killing, we measured optical densities of 226 *M. abscessus* strains during planktonic growth in nutrient-rich conditions and fitted Gompertz functions to provide model-based estimates of each isolate's lag time and growth rate (Fig. 3c,d). Apart from revealing a strong direct correlation between growth rate and lag time, reflecting a trade-off between bacterial growth and adaptation[44], we identified correlations between growth rates or lag times and drug tolerance phenotypes (Fig. 3e and Supplementary Fig. 2). For instance, drug tolerance to imipenem and cefoxitin, which target cell wall synthesis specifically during bacterial replication, correlated with slow growth but not lag time. In contrast, moxifloxacin killing (targeting the DNA) was associated with longer lag times only. These associations were weak, explaining only up to 13% of the variation in the drug tolerance phenotype, suggesting that replication patterns were not the main driver of drug tolerance. However, potential technical factors, such as variability in growth assessments, curve fitting or isolate clumping, may have reduced the strength of these associations. We also found weak associations ($R^2$ up to 0.11) between drug tolerance and resistance phenotypes (especially for cefoxitin and minocycline), mainly driven by outliers with MICs close to the tested concentration. Using a non-parametric test (Spearman correlation), most of these associations disappeared (Supplementary Fig. 2), supporting the idea that drug tolerance is a marker of antibiotic efficacy independent of drug resistance[45]. Drug tolerance phenotypes, especially for antibiotics targeting protein

synthesis, were strongly associated, with some correlating weakly with MICs (Extended Data Fig. 8a), suggesting shared mechanisms or adaptive evolution reminiscent of collateral sensitivity and cross-resistance.

## Drug tolerance is heritable

While drug resistance is well established as a genetic trait, drug tolerance has traditionally been viewed as a largely phenotypic characteristic. Recent studies, however, have identified multiple genetic mechanisms underlying drug tolerance in mycobacteria and other bacterial species[42,46–48]. We employed whole-genome sequencing and estimated the proportion of phenotypic variance attributable to genetic variability, known as heritability[49]. Specifically, we mapped 1.3 million *M. abscessus* unitigs (sequences of variable length), which capture most genomic variation arising from vertical inheritance, horizontal gene transfer and mutations including single nucleotide polymorphisms, insertions, deletions and gene presence–absence, to respective phenotypes using linear mixed models[50,51]. Resistance phenotypes for several antibiotics (particularly macrolide MICs) were strongly determined by bacterial genetics, whereas others, such as imipenem and cefoxitin, showed low heritability, probably reflecting chemical instability and limited biological variation in MIC measurements (Fig. 4a). Across all drugs, tolerance phenotypes demonstrated substantial heritability (32–97%), far exceeding random chance (1.1%, Student's $t$-test $P < 2.2 \times 10^{-16}$), indicating that killing phenotypes are strain-specific and largely genetically determined.

Mapping these phenotypes to the *M. abscessus* phylogeny (Fig. 4b and Extended Data Fig. 8b), we observed that some of the high- or low-tolerance phenotypes evolved in parallel, resulting in homoplastic traits, while others were conserved in phylogenetically related isolates, indicating inherited clades. One such example is a tigecycline low-tolerance clade within the dominant circulating clone of *M. abscessus massiliense*[52]. Many of these isolates carry high-level mutational aminoglycoside and macrolide resistance and have been linked to increased virulence[40]. Our finding of low tigecycline tolerance highlights a potential vulnerability in otherwise highly drug-resistant *M. abscessus*, where treatment success rates can fall below 20%.

## Drug tolerance predicts individual infection outcomes

We then investigated whether strain-specific kill kinetics influence individual patient outcomes, given the limited direct evidence linking drug tolerance to treatment failures[7]. Our analysis showed that, among eight tested drugs, only macrolide MICs were associated with clinical outcomes (Fig. 5a and Extended Data Fig. 9a,b). However, tolerance to amikacin, cefoxitin and imipenem, measured at low and high drug concentrations, correlated with *M. abscessus* clearance in patients (Extended Data Fig. 10). Specifically, rapid bacterial killing was associated with favourable clinical outcomes, whereas high tolerance was linked to poor outcomes. Notably, the effect sizes of drug tolerance in predicting infection outcomes were comparable to those of macrolide MICs, currently the main in vitro measure guiding *M. abscessus* treatment decisions. Incorporating a single drug tolerance measure alongside macrolide resistance improved clinical outcome prediction, increasing the AUC-ROC from 0.69 to 0.78 (Fig. 5b, and Supplementary Tables 2 and 3). These findings underscore drug tolerance as an independent and clinically relevant marker of antibiotic efficacy.

## Drug tolerance is target specific

By applying principal component analysis to 5,056 *M. abscessus* time–kill curves and mapping antibiotics in drug tolerance space, we identified drug clusters related to the drug's mode of action (Fig. 6a). These clusters indicate that bacterial survival mechanisms are linked to the drug's target, probably driven by shared downstream effects, such as conserved stress responses that modulate antibiotic killing[53].

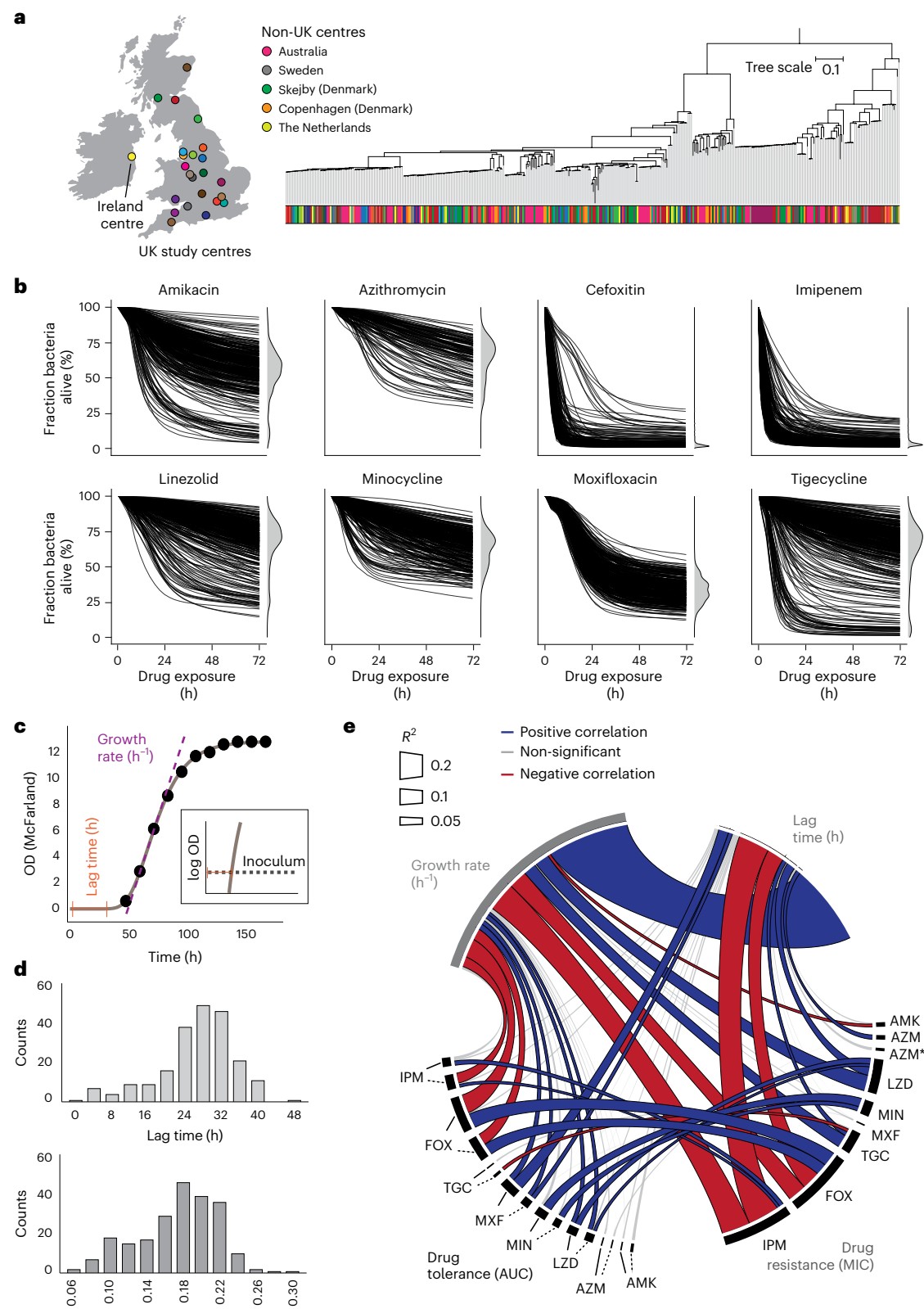

**Fig. 3 | Antibiotic tolerance is a distinct measure of antibiotic efficacy.**
**a**, Study centres in the United Kingdom, Ireland, mainland Europe and Australia. Maximum-likelihood phylogenetic tree of 376 *M. abscessus* isolates obtained from different patients, with each centre indicated by colour. **b**, Time–kill curves of 406 *M. abscessus* isolates across 8 antibiotics (the lower drug concentrations are shown), with viability distributions at 72 h, excluding growing isolates. **c**, Gompertz function fitting to optical density (OD) measurements (black dots), illustrating the determination of bacterial growth rates and lag times.

The insert highlights lag time estimation from the Gompertz function and inoculum size on a logarithmic scale. **d**, Distributions of lag times (top) and growth rates (bottom) from 226 fitted *M. abscessus* growth curves. **e**, Pearson correlation analyses between drug tolerance (high drug concentration, continuous line; low concentration, dashed line), bacterial growth rate, lag time and the corresponding MIC. Line colour indicates the direction of the correlation; line width represents the corresponding $R^2$ value. AZM*, inducible azithromycin resistance.

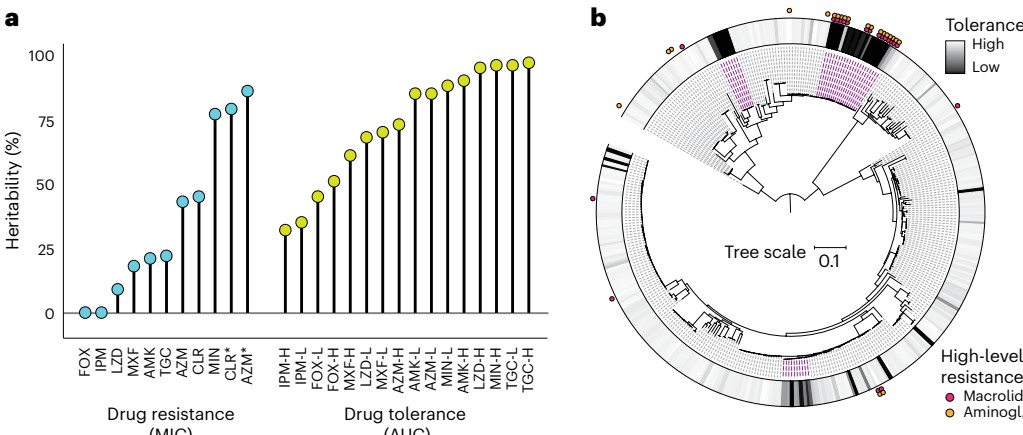

**Fig. 4 | Tolerance is driven by the genetic bacterial background. a**, Heritability estimates for drug resistance (MIC; $n = 221$) and drug tolerance phenotypes (AUC; $n = 371$, excluding growing isolates) based on 1.3 million genetic *M. abscessus* variants (unitigs) derived from whole-genome sequencing. CLR, clarithromycin; CLR*, inducible clarithromycin resistance; L, low drug concentration; H, high drug concentration. **b**, Maximum-likelihood phylogenetic tree of 353 *M. abscessus* isolates, aligned with the tigecycline tolerance heat map (high concentration) and mutational macrolide and aminoglycoside (Aminogl.) resistance. Purple lines mark clades with reduced tigecycline tolerance.

To explore the genetic basis of drug tolerance, we mapped ~280,000 *M. abscessus* genotypes (single nucleotide polymorphisms, insertions and deletions) extracted from whole-genome sequences of each isolate to drug tolerance phenotypes, using mixed-effects models corrected for population structure. This analysis identified multiple genotypes and genes strongly associated with drug tolerance phenotypes (Fig. 6b). Tolerance-associated genes grouped by the antibiotic target, with specific gene sets linked to protein synthesis, DNA and cell wall-active drugs.

We further examined *MAB_0233*, a putative phage tail tape measure protein, in which several deletions were associated with drug tolerance to drugs targeting protein synthesis (Fig. 6c). As such, *MAB_0233* could facilitate intracellular drug accumulation, enhance metabolic activity to potentiate drug effects, interfere with stress responses, or directly synergize with the drug. Using oligonucleotide-mediated recombineering followed by Bxb1 integrase targeting (ORBIT[54]), we generated a *MAB_0233* knockout strain in *M. abscessus*, which showed increased tolerance for antibiotics targeting translation (amikacin, tigecycline and linezolid), but not the cell wall, while MICs remained stable (Fig. 6d and Supplementary Fig. 3a,b). *MAB_0233* complementation restored the low-tolerance phenotype for amikacin and tigecycline. However, for linezolid, the complemented strain retained an altered time–kill profile, suggesting partial complementation or the involvement of additional mechanisms influencing linezolid tolerance.

## Discussion

Antibiotics that demonstrate activity in vitro may fail to achieve clinical success, highlighting a critical disconnect between microbiological testing and patient outcomes. This gap represents a major challenge in antibiotic development and patient care. To address it, we developed antimicrobial single-cell testing, a strategy that quantifies bacterial viability in real time and at large scale, providing new insights into an underexplored dimension of antibiotic activity. Applying ASCT, we demonstrate that drug-specific killing (in *M. tuberculosis*) and strain-specific killing (in *M. abscessus*) are key predictors of drug trial failures and poor clinical outcomes, and thus determinants of in vivo efficacy. This dual perspective of conserved drug effects and variable strain-specific responses shows how both components shape antibiotic killing and ultimately affect treatment outcomes.

Several large-scale approaches to quantify bacterial viability have been recently developed, typically coupling a viability marker with an appropriate readout technology. Markers include conventional regrowth, metabolic dyes, biosensors and fluorescent reporters, while readouts range from bulk optical or fluorescent measurements to live-cell imaging, flow cytometry and label-free biophysical methods[16,17]. Although these strategies move beyond classical culture assays, most remain constrained by limited throughput, sensitivity or single-cell resolution, leaving critical aspects of bacterial survival unresolved. To overcome these limitations, we established a platform for repetitive, large-scale, real-time, single-cell assessments of bacterial viability across diverse clinical isolates. We used PI as a viability marker and confirmed that PI-positive cells do not regrow and can therefore be considered dead[23]. To further characterize PI-negative cells, we employed a reporter strain with inducible GFP expression and found that most PI-negative bacteria retained the ability to synthesize new proteins, in contrast to the very small fraction of bacteria that regrew. Notably, many PI-negative cells that initially produced GFP later died after drug washout. This post-exposure killing probably reflects incomplete drug removal, classical post-antibiotic effects or nutrient shock[24,55,56]. These findings indicate that conventional methods underestimate bacterial survival and highlight the importance of real-time, single-cell measures of bacterial viability. Still, defining true bacterial viability remains challenging, and PI may only partially capture it. We therefore focused on identifying phenotypes predictive of in vivo outcomes, a goal achieved with greater performance and throughput by ASCT through real-time single-cell PI quantifications.

We applied ASCT to investigate the time–kill kinetics of multiple *M. tuberculosis* drug regimens across two strains and various experimental conditions. While combinations including the standard drugs isoniazid, rifampicin and ethambutol were most lethal under growth conditions, only killing under starvation predicted in vivo population outcomes. Antibiotic killing in these starvation conditions, reflective of metabolic inactivity in intra- and extracellular infection niches[57], correlated strongly with outcomes in multiple in vivo models, including the relapsing and bactericidal mouse models as well as humans during phase 2 clinical trials. Thus, even using non-clinical, avirulent *M. tuberculosis* strains, we were able to predict in vivo outcomes of virulent tuberculosis at the population level, something not achievable with c.f.u. counts, MICs, drug interaction measures or resazurin-based assessments. Importantly, our data also show that not only the extent of killing matters, but also the physiological bacterial state in which killing is assessed, highlighting the importance of infection-relevant contexts when evaluating killing dynamics. Extending this approach to more host-like conditions may further improve its predictive performance.

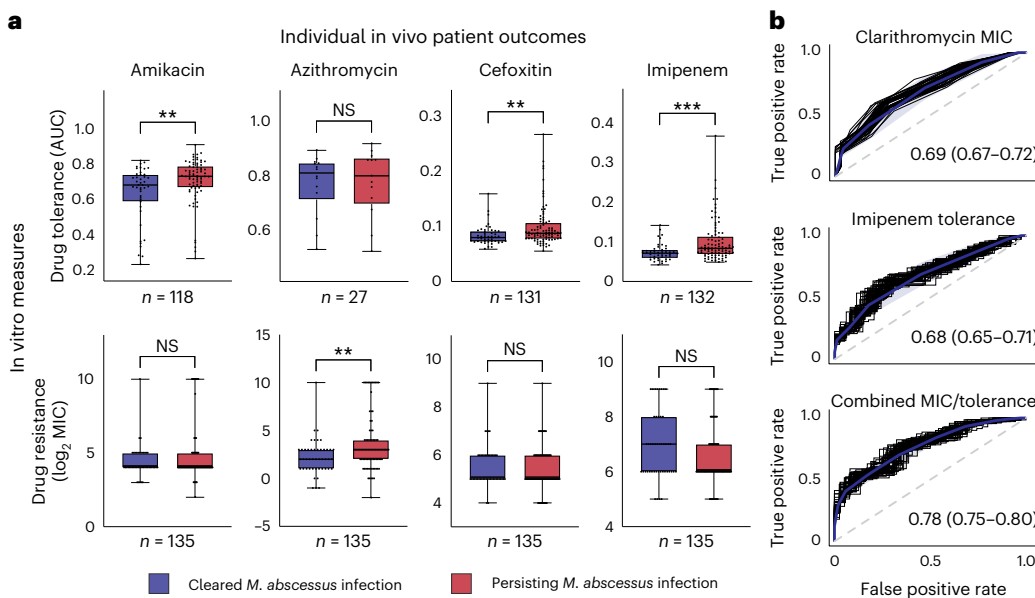

**Fig. 5 | Drug tolerance predicts individual infection outcomes. a**, Comparison of *M. abscessus* drug tolerance (AUC, low concentration) and drug resistance (MICs) between isolates from patients with poor and favourable clinical outcomes, using a two-sided Mann–Whitney *U*-test (*n* = 135, excluding growing isolates). Boxplots show the median, interquartile range and total range (central line, box and whiskers, respectively) of AUC values. \*\**P* < 0.01, \*\*\**P* < 0.001.

**b**, Logistic regression predicting clinical outcomes from clarithromycin resistance (MIC) and imipenem tolerance (low concentration), individually and combined. Fifty ROC curves were generated by randomly selecting 80% of the samples for each condition. The blue line represents the mean ROC curve and the shaded area denotes 1 s.d. The mean AUC-ROC and 95% confidence intervals are shown.

Given the high attrition rates in antibiotic development and clinical trials[58], there is an urgent need for more efficient strategies to prioritize therapeutic regimens, which is a challenging task, considering the growing number of active lead compounds and possible combinations[59]. By enabling the evaluation of thousands of drug combinations in vitro—with demonstrated relevance in vivo—ASCT has the potential to identify more effective and much shorter treatment regimens against *M. tuberculosis* infection and disease. This advancement could accelerate global health initiatives, such as the World Health Organization's goal to eliminate most tuberculosis burden by 2035[12,60].

Highly effective antibiotics are the cornerstone of successful antibiotic treatments, yet clinical outcomes vary widely, with some patients responding to short courses and others failing despite prolonged multidrug treatment[61]. We show that antibiotic killing is highly variable across clinical *M. abscessus* isolates, further challenging the prevailing clinical view that bacterial killing is solely determined by the drug's bactericidal properties[62]. This variability in killing, different from drug resistance, highlights drug tolerance as a fundamental bacterial trait that may greatly affect treatment efficacy. Notably, there exists scarce evidence that drug tolerance, rather than limited drug penetration or reinfections, underlies antibiotic treatment failures[63–65]. We found that *M. abscessus* strains poorly killed by amikacin, cefoxitin or imipenem (in high and low drug concentrations) were linked to worse clinical outcomes in individual patients, providing some of the most compelling evidence so far that drug tolerance is a critical determinant of treatment success. Because drug treatments are complex and often change, actual drug exposure was unclear, probably underestimating the impact of drug tolerance. These findings indicate that tolerance phenotypes could complement conventional drug susceptibility testing to improve outcome prediction and clinical decision-making.

We also show that drug tolerance (such as resistance) is largely determined by the bacterial genetic background and, thus, a heritable and evolvable bacterial trait. For example, within the subspecies *M. abscessus massiliense*, we identified a low-tolerance clade among highly resistant strains, revealing vulnerabilities that could be exploited

to enhance bacterial clearance. Tolerance phenotypes clustered with the drug's mode of action, suggesting conserved survival pathways among drugs targeting similar cellular functions. Using phenogenomic analysis, we identified numerous genes and genotypes associated with high- or low-tolerance phenotypes. Through gene knockout and complementation experiments, we further validated a phage protein that modulates bacterial killing. These tolerance mechanisms represent potential targets for sterilizing antibiotic treatments, while tolerance genotypes could be integrated into diagnostics, analogous to molecular susceptibility testing. Presumably, molecular testing, or phenotyping via ASCT, could facilitate individualized, tolerance-tailored antibiotic regimens and help improve patient outcomes beyond the development of new drugs and regimens.

We note several limitations. Because PI measures membrane compromise, it directly reflects killing by cell wall damage but only indirectly captures other killing mechanisms. Since all dead cells eventually lose membrane integrity, PI therefore detects killing by non-lytic antibiotics only after a delay. In *M. abscessus* (but not in *M. tuberculosis*) we observed a delay of ~3 h for such drugs. Given the extended duration of our experiments and in vivo validation, this effect is probably minimal in mycobacteria but may become more relevant when applying ASCT to other bacterial species. While our findings highlight bacterial killing as a key determinant of antibiotic activity, clinical treatment outcomes are influenced by many additional factors. These include host immune responses, fluctuating drug exposures and penetration into tissues, drug interactions, toxicity and adherence, as well as the complexity of infection environments, such as intracellular niches and granulomas in mycobacterial infections, or acidic compartments with altered drug activity (for example, pyrazinamide)[34,66,67]. ASCT does not capture these non-bacterial determinants and therefore cannot fully predict treatment outcomes, but it provides a scalable and biologically meaningful readout of bacterial susceptibility that complements existing tools. We validated ASCT in mycobacteria, but this approach is probably applicable to other bacterial species and multidrug-resistant pathogens. However, adjustments in experimental design, imaging setup and analysis will be required to account for the species-specific

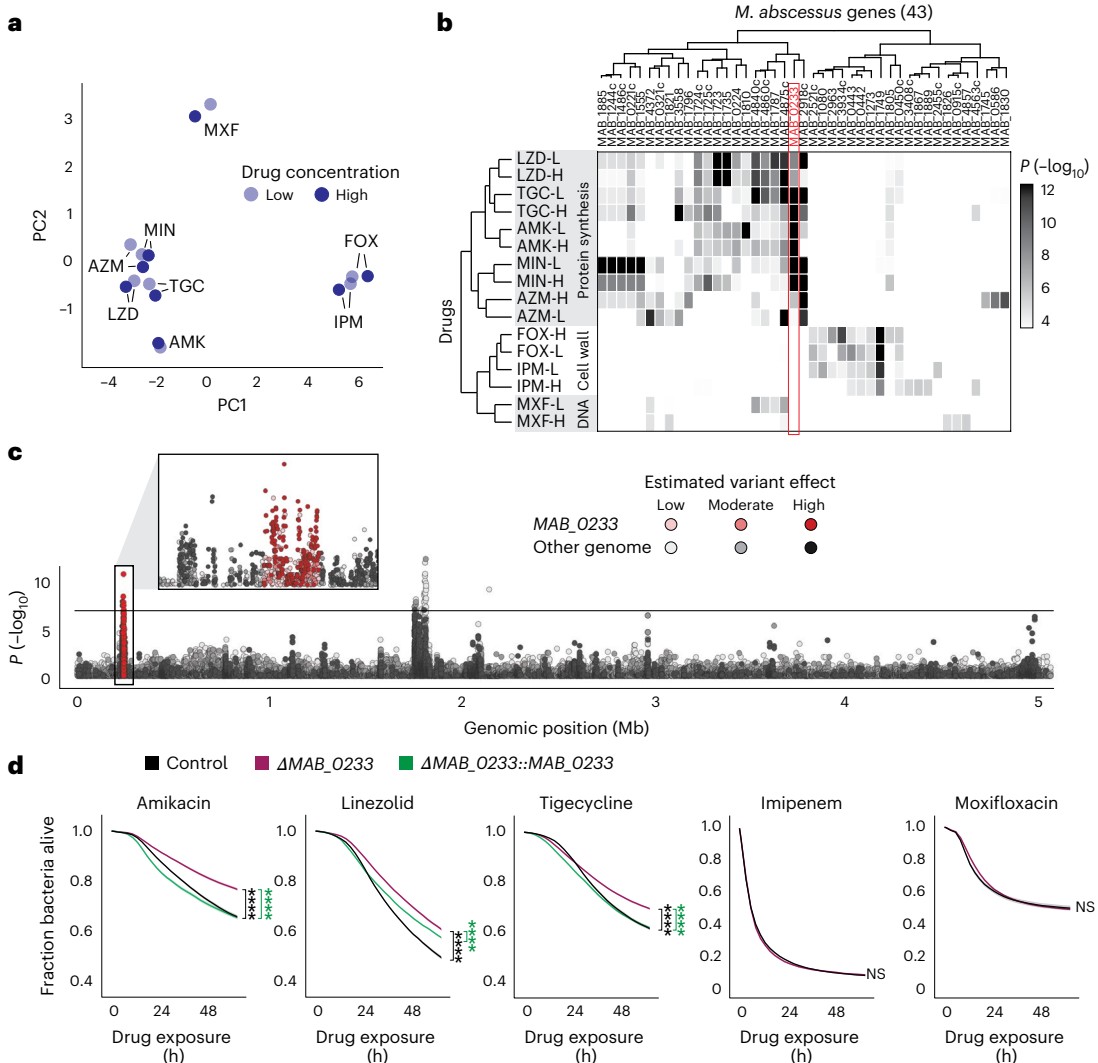

**Fig. 6 | Drug tolerance is target specific. a**, Principal component analysis of 8 drugs at 2 concentrations in drug tolerance space, based on Spearman correlations from 350 clinical *M. abscessus* isolates. **b**, Genome-wide association results for 16 drug tolerance phenotypes, highlighting the top 5 associated genes (moderate- or high-effect genotypes) per phenotype. The heat map shows association of 43 genes (top genotype per gene) with tolerance phenotypes using a mixed-effects model corrected for population structure (two-sided Wald test). Associations with $P > 0.0001$ are in white. **c**, Manhattan plot of 272,351 *M. abscessus* genotypes and their association with amikacin (low concentration)

kill kinetics (two-sided Wald test), with the Bonferroni correction threshold (black line, $1.7 \times 10^{-7}$). Multiple variants in *MAB_0233* were strongly linked to amikacin killing (insert). **d**, Time-kill kinetics of control, Δ*MAB_0233* and complemented strains (mean ± s.e.m., nine replicates per condition). AUC values of Δ*MAB_0233* were compared with control or complemented strains using a two-sided Mann–Whitney *U*-test. *P* values for amikacin, linezolid and tigecycline were $4 \times 10^{-5}$, except for tigecycline time–kill kinetics between Δ*MAB_0233* and the control strain ($P = 8 \times 10^{-5}$). ****$P < 0.0001$.

morphology and physiology, particularly in regard to viability assessments. The in vivo relevance of drug tolerance and nutrient limitation is also likely to vary across pathogens and infection sites.

In our study we interrogated antibiotic killing at two levels: first, whether drug-specific killing in *M. tuberculosis* correlates with in vivo outcomes at the population level; and second, whether strain-specific variation in killing across clinical *M. abscessus* isolates associates with individual patient responses. Although these approaches differ (drug regimen versus strain), both converge on the same principle that killing dynamics are critical determinants of treatment success. While we examined these factors independently, clinical treatment responses are shaped by the interplay of drug properties and bacterial behaviours. By studying millions of single-cell fates across hundreds of conditions, our approach provides a scalable framework to translate in vitro killing into in vivo efficacy, opening new avenues for drug development and personalized therapy.

## Methods
### Sample collection
We used the *Mycobacterium abscessus* laboratory strain ATCC-19977, the avirulent *Mycobacterium tuberculosis* strain H37Ra and the auxotrophic *M. tuberculosis* strain mc²7000 (H37Rv ΔpanCD ΔRD1)[68]. Clinical *M. abscessus* isolates were obtained from respiratory samples (sputum or bronchoalveolar lavage fluid) collected from patients with pulmonary *M. abscessus* infection as described previously[40]. All work with clinical *M. abscessus* isolates and avirulent *M. tuberculosis* strains was conducted in a biosafety level 2 (BSL-2) facility, in accordance with institutional biosafety regulations. Included patients came from all major cystic fibrosis centres in the United Kingdom, the Republic of Ireland (Dublin), Sweden (Gothenburg), Denmark (Copenhagen and Skejby), the Netherlands (Nijmegen) and Australia (Queensland). *M. abscessus* isolates were retrieved from the original mycobacterial growth indicator tubes (MGIT) or, if otherwise unavailable, from

subcultured isolates. The study was approved in England and Wales by the National Research Ethics Service (12/EE/0158) and the National Information Governance Board (ECC 3-03 (f)/2012) and in other centres by respective local review boards.

### Bacterial growth assessments

All consumables are provided in Supplementary Table 4. Planktonic growth rates and the lag times of *M. abscessus* isolates were assessed following suspension in nutrient-rich media. Approximately $10^5$ c.f.u.s of *M. abscessus* were inoculated in 2.5 ml of Middlebrook 7H9 medium (supplemented with 0.4% glycerol, 10% OADC [oleic acid, albumin, dextrose, catalase] and 0.05% Tween 80) in 15 ml glass tubes, which were then incubated at 37 °C with 150 r.p.m. orbital shaking. Optical densities ($OD_{565}$) were measured with a densitometer (DEN-1B; Biosan) and quantified in McFarland units. $OD_{565}$ readings were performed every 12 h for a minimum of 5 days until the readings stabilized for at least 24 h (change in McFarland less than 0.5). Background-corrected $OD_{565}$ measurements were used to fit Gompertz functions to estimate growth rates, where $N$ is the number of bacteria at a given time ($t$), with $A$ representing carrying capacity (maximum population size), $B$ the initial growth factor affecting displacement along the $y$ axis and $C$, the growth rate constant. Lag times were calculated from the fitted Gompertz function and the known inoculum size.

$$N(t) = Ae^{-Be^{-Ct}} \qquad (1)$$

Given that conventional formulas are imprecise when considering that the growth rate is a proportion of the carrying capacity per time[69], we calculated the growth rate ($\mu$) with the following equation:

$$\mu = C \frac{\text{Lambert}W\left(\frac{A}{eN_0}\ln\left(\ln\left(\frac{A}{N_0}\right)\right)\right)}{\ln\left(\ln\left(\frac{A}{N_0}\right)\right)} \qquad (2)$$

The lag time ($\lambda$) was calculated as:

$$\lambda = -\ln\left(\ln\left(\frac{A}{N_0}\right)/B\right)/C \qquad (3)$$

### Drug susceptibility testing

Antibiotic resistance was quantified using MICs, following the Clinical Laboratory Standards Institute (CLSI; M24 3rd edition) guidelines[70]. B.W. received training at the National Center for Mycobacteria, Zürich (Switzerland), to align with standard clinical microbiology practices. *M. abscessus* isolates were cultured in Middlebrook 7H9, supplemented with 0.4% glycerol, 10% OADC and 0.05% Tween 80, for 3 days and then diluted in PBS to a McFarland standard of 0.5. A 140 µl aliquot of this suspension was mixed into 14 ml cation-adjusted Mueller–Hinton broth (CAMHB) to achieve an approximate density of $10^6$ c.f.u.s ml$^{-1}$. Antibiotics were tested in log$_2$-fold concentration steps at the following ranges: amikacin (0.25–512 µg ml$^{-1}$), azithromycin (3.9 ng l$^{-1}$–512 µg ml$^{-1}$), cefoxitin (0.5–256 µg ml$^{-1}$), clarithromycin (3.9 ng l$^{-1}$–512 µg ml$^{-1}$), clofazimine (7.8 ng ml$^{-1}$–16 µg ml$^{-1}$), imipenem (0.13–56 µg ml$^{-1}$), linezolid (0.25–128 µg ml$^{-1}$), minocycline (15.6 ng ml$^{-1}$–256 µg ml$^{-1}$), moxifloxacin (15.6 ng ml$^{-1}$–32 µg ml$^{-1}$) and tigecycline (15.6 ng ml$^{-1}$–8 µg ml$^{-1}$). For each experiment CAMHB was freshly prepared. Antibiotics and CAMHB were added to 96-well plates to reach a volume of 50 µl per well. After mixing, the 50 µl bacterial suspension was added to each well to obtain a final bacterial concentration of ~5 × 10$^5$ c.f.u.s ml$^{-1}$. Each plate included growth and media control wells. *Mycobacterium peregrinum* (ATCC-700686) MICs were assessed for quality control in each experimental batch. Plates were sealed and incubated at 30 °C and visually evaluated for growth after 3–5 days once growth was visible in the growth control wells. Azithromycin and clarithromycin conditions were reassessed after 14 days of incubation to evaluate inducible macrolide resistance. MICs were recorded as the

lowest drug concentration preventing visible mycobacterial growth. For further analyses, we used either log$_2$-transformed MIC values or applied previously reported *M. abscessus* resistance breakpoints (clarithromycin MIC ≥ 8 µg ml$^{-1}$; amikacin MIC ≥ 64 µg ml$^{-1}$ (ref. 71)).

*M. tuberculosis* drug susceptibility testing was done analogous to *M. abscessus*, except that *M. tuberculosis* isolates were cultured in 5 ml of Middlebrook 7H9 broth, supplemented with 0.4% glycerol, 10% OADC and 0.05% Tween 80 in 50 ml tubes (*M. tuberculosis* mc$^2$7000 was additionally supplemented with 100 µg ml$^{-1}$ pantothenate), and that MIC plates were assessed after 14 days. Bedaquiline, clofazimine, delamanid, ethambutol, linezolid, moxifloxacin, pretomanid, rifampicin, rifapentin, SQ109 and sutezolid were assessed in log$_2$-fold concentration steps between 2 ng ml$^{-1}$ and 64 µg ml$^{-1}$, whereas isoniazid and pyrazinamide were assessed between 4 ng ml$^{-1}$ and 128 µg ml$^{-1}$.

### C.f.u.-based time–kill kinetics

*M. tuberculosis* mc$^2$7000 was cultured in 10 ml of Middlebrook 7H9 broth supplemented with 0.4% glycerol, 10% OADC, 0.05% Tween 80 and 100 µg ml$^{-1}$ pantothenic acid. Cultures were incubated at 37 °C with shaking (150 r.p.m.) until mid-log phase, then washed and resuspended in PBS before incubation at 37 °C for 14 days. PBS-starved bacteria where standardized to ~10$^7$ c.f.u.s ml$^{-1}$, and 180 µl was dispensed into 96-well U-bottom-plates. Individual drugs or drug combinations were added (30 µl) to achieve maximum therapeutic blood concentrations ($C_{max}$): bedaquiline 1.1 µg ml$^{-1}$, clofazimine 1.25 µg ml$^{-1}$, delamanid 0.5 µg ml$^{-1}$, ethambutol 4 µg ml$^{-1}$, isoniazid 4.5 µg ml$^{-1}$, linezolid 19 µg ml$^{-1}$, moxifloxacin 4 µg ml$^{-1}$, pretomanid 7.8 µg ml$^{-1}$, pyrazinamide 40 µg ml$^{-1}$, rifampicin 16 µg ml$^{-1}$, rifapentin 19 µg ml$^{-1}$, SQ109 0.026 µg ml$^{-1}$ and sutezolid 1.48 µg ml$^{-1}$. Each condition was tested in triplicate. Plates were sealed with parafilm, covered with lids and incubated at 37 °C. At days 0, 3 and 7, 20 µl from each condition were transferred into 180 µl of PBS. Serial dilutions were performed, and 5 µl of each dilution were spotted onto 7H11 agar plates supplemented with 100 µg ml$^{-1}$ pantothenic acid (in duplicate). In addition, 50 µl of undiluted samples were washed and directly plated on pantothenate-supplemented 7H11 agar. Plates were incubated at 37 °C until visible colonies emerged. Colonies were manually counted at dilutions where 2–200 colonies were present, and c.f.u.s ml$^{-1}$ were calculated. The detection limit was 20 c.f.u.s ml$^{-1}$.

### In vivo data

The in vivo classifications of *M. tuberculosis* drug regimens 'as good or worse than SOC (where isoniazid-rifampicin-pyrazinamide-ethambutol or isoniazid-rifampicin-pyrazinamide were considered SOC)' or 'better than SOC' were obtained from two previous studies. In vivo classifications based on mouse models were obtained from ref. 30, and classifications based on phase 2a and 2b clinical trials (assessing bactericidal activity) were obtained from ref. 31. All similar-to-SOC or better-than-SOC drug combinations from the relapsing mouse model ($n = 46$) and clinical studies ($n = 14$) were assessed with ASCT. These drug combinations and classifications of single drugs were also used to evaluate the performance of ASCT in bactericidal mouse models (common mouse strains: $n = 48$; C3HeB/FeJ mouse strain: $n = 15$). Clinical metadata of patients with respiratory *M. abscessus* infection were available for a subset of patients. Treatment outcomes were assessed in patients meeting the ATS/IDSA criteria of NTM pulmonary disease[72]. Patients with positive cultures after 6 months of *M. abscessus* treatment were considered as having persisting infections, and patients with sustained culture conversion (microbiological clearance) as cleared infections. Growing isolates were excluded from the analysis of bacterial killing. The discriminative performance for predicting the success of drug regimens in mouse or patient populations (*M. tuberculosis*) and the clinical outcome in individual patients (*M. abscessus*) was assessed with logistic regression and quantified with AUC-ROC. For each condition, 50 bootstrap iterations were performed by randomly

selecting 80% of the samples. Sensitivity, specificity, positive and negative predictive values, and F1 scores were calculated at the optimal cut-off determined by the Youden index.

## ACST experimental setup

*M. abscessus* and *M. tuberculosis* isolates were cultured in 5 ml of Middlebrook 7H9 broth, supplemented with 0.4% glycerol, 10% OADC and 0.05% Tween 80 in 50 ml tubes (*M. tuberculosis* mc²7000 was additionally supplemented with 100 µg ml⁻¹ pantothenate). The cultures were incubated at 37 °C with shaking at 150 r.p.m. until mid-log phase (McFarland: 5–8). In *M. tuberculosis*, we also assessed two starvation conditions: PBS starvation and pantothenate starvation. For PBS starvation, mid-log phase bacteria (H37Ra or *M. tuberculosis* mc²7000) were washed and resuspended in PBS, the 50 ml tube filled with PBS and incubated for 14 days at 37 °C without shaking. For pantothenate starvation, *M. tuberculosis* mc²7000 (H37Rv ΔpanCD ΔRD1) was incubated for 14 days with Middlebrook 7H9, supplemented with 0.4% glycerol, but not pantothenate.

After growth or starvation, mycobacteria were centrifuged at 3,000 *g* for 10 min, and the resulting pellet was resuspended in Middlebrook 7H9 (for growth conditions and pantothenate starvation), in Middlebrook 7H9 with pantothenate supplementation (for growth conditions in *M. tuberculosis* mc²7000) or in PBS (for PBS starvation). To achieve single-cell suspensions, large clumps were removed by low-speed centrifugation (200 *g*, 3 min), followed by serial filtration of the supernatant through 5 µm and 1.2 µm filters (Sartorius Minisart). In every sample, bacterial densities were assessed using $OD_{565}$ measurements, and the live-cell fraction was quantified using propidium iodide staining and imaging. Bacterial samples with a live-cell fraction below 95% were discarded and reprocessed. Single-cell suspensions were immediately used (*M. tuberculosis*) or frozen at −80 °C (*M. abscessus* clinical isolates).

ASCT uses a dual-layer approach: the first layer consists of an agar pad containing and immobilizing bacteria, and the second layer comprises drug-containing solutions (Fig. 1a). To prepare the agar pad, ultra-low gelling temperature agarose (ULGA; Lonza SeaPrep) and Middlebrook 7H9 were dissolved in hot $dH_2O$. Additional dissolution was facilitated through heating, where any evaporated water was replaced to maintain constant 7H9 and agarose concentrations. Once the solution cooled to 50 °C or lower, OADC and glycerol were added. The mixture was then filtered through a 0.22 µm filter and propidium iodide was added. This agarose solution was aliquoted as needed. The final gel pad solution contained 0.4 agarose, 1× Middlebrook 7H9, 10% OADC, 0.4% glycerol, 8 µg ml⁻¹ propidium iodide and ~5 × 10⁶ bacteria per ml. For non-starving *M. tuberculosis* mc²7000, the agar pad also contained 100 µg ml⁻¹ pantothenate. In pantothenate-starvation conditions, the gel pad did not contain pantothenate, and in PBS-starvation media, all supplements were replaced with 1× PBS.

All preparations were made in 96-well plates using a 96-channel pipette (Mini96, Integra Biosciences). Of the isolate–agar–PI solution, 7 µl was dispensed, using liquid handling (I.DOT, Dispendix), into each well of a 1,536-well plate (Greiner, Screenstar). At least one column and row were left as buffer wells at the edges. All materials used in the agarose preparation, including chemicals, pipette tips, filters, the plate and so on, were preheated to 37 °C. The 1,536-well plate was centrifuged at 37 °C and 3,000 *g* for 30 min to position the bacteria at the bottom of the wells for imaging. This step was followed by cold centrifugation at 4 °C and 1,500 *g* for 20 min to solidify the ULGA. After leaving the plate for 30 min at room temperature, a defined drug solution (4 µl) dissolved in Middlebrook 7H9 or PBS (for PBS starvation) was added to each well. The plate was then sealed with parafilm, covered with a lid and remained for another 60 min at room temperature before being placed into the microscope's preheated live-cell chamber (37 °C; Life Imaging Services). To evaluate the impact of gel pad volumes and concentrations on antibiotic killing, we tested various volumes (4, 5, 6 and 7 µl) and concentrations (0.3, 0.4 and 0.5%; Extended Data Fig. 1g,h).

Images were captured with a Nikon Ti2-E inverted microscope using a CFI Plan Apo Lambda ×40 NA 0.95 or a CFI Plan Apo Lambda ×100 NA 1.45 objective, and the Perfect Focus System for maintenance of the focus over time. PI fluorescence was excited with a Spectra III Light Engine (Lumencor) at 555 nm and collected with a penta-edge 408/504/581/667/762 dichroic beam splitter and a 440/40, 520/21, 606/34, 694/34, 809/81 penta-bandpass filter. Images (×40 12-bit and ×100 (optionally with ×1.5 zoom lens) 16-bit) were acquired with a Photometrics Kinetics camera, controlled with the Nikon NIS acquisition software. For ×40, the fluorescence excitation light intensity was set to 10% and the camera exposure time to 20 ms, and the brightfield illumination light intensity was set to 50% and the camera exposure time to 8 ms. These settings were kept the same for all experiments and conditions for comparability. For ×100 acquisition, the excitation light intensity and camera exposure were adjusted accordingly.

The NIS JOBS module was used for image acquisition automation. Briefly, 9 equally spaced fields of view for each well and imaging timepoint were acquired. For *M. abscessus*, images were captured every 2.5 h for a total duration of 72 h and 29 timepoints; for *M. tuberculosis*, images were acquired every 4 h for a duration of 168 h (42 timepoints in total). Imaging one timepoint took ~130 min (1,260 wells with 9 fields of view per well). At the end of image acquisition, imaging data (*M. abscessus*: up to 11,340 movies and 7.8 TB per experiment; *M. tuberculosis*: up to 11 TB per experiment) were transferred to the high-performance computing cluster at the University of Basel (sciCORE) for subsequent image and data analysis.

To evaluate ASCT-based time–kill kinetics of *M. tuberculosis*, we exposed *M. tuberculosis* to previously published drug combination regimens[30,31] at the maximum blood concentration ($C_{max}$) achievable during therapeutic dosing in humans (9 replicates for each condition): bedaquiline 1.1 µg ml⁻¹, clofazimine 1.25 µg ml⁻¹, delamanid 0.5 µg ml⁻¹, ethambutol 4 µg ml⁻¹, isoniazid 4.5 µg ml⁻¹, linezolid 19 µg ml⁻¹, moxifloxacin 4 µg ml⁻¹, pretomanid 7.8 µg ml⁻¹, pyrazinamide 40 µg ml⁻¹, rifampicin 16 µg ml⁻¹, rifapentin 19 µg ml⁻¹, SQ109 0.026 µg ml⁻¹ and sutezolid 1.48 µg ml⁻¹.

For drug tolerance profiling, ASCT was performed on all clinical *M. abscessus* isolates using a standardized panel of eight antibiotics, regardless of the treatment each patient received. *M. abscessus* isolates were assessed within a single ASCT plate for a single antibiotic condition in triplicate. The drug concentrations used were based on the MICs for ATCC-19977, determined similar to CLSI criteria[71]. In contrast to CLSI, they were assessed in 7H9 supplemented with 0.4% glycerol, 10% OADC and 0.05% Tween 80 and quantified using $OD_{600}$. Growth inhibition at increasing drug concentrations was fitted with a four-parameter log-logistic model using the R drc package[73]. The MIC was defined as the drug concentration that prevented 90% growth ($IC_{90}$). Antibiotic time–kill kinetics for all 8 drugs were evaluated at 2 different drug concentrations: a lower concentration (10-fold the MIC of ATCC-19977) and a higher concentration (20-fold the MIC of ATCC-19977). The specific drug concentrations (low and high) tested were as follows: amikacin 26 and 52.1 µg ml⁻¹, azithromycin 63.5 and 127 µg ml⁻¹, cefoxitin 63.1 and 126.2 µg ml⁻¹, imipenem 37.7 and 75.4 µg ml⁻¹, minocycline 67.2 and 134.4 µg ml⁻¹, moxifloxacin 8.5 and 17 µg ml⁻¹, linezolid 38.5 and 77.1 µg ml⁻¹, and tigecycline 27.2 and 54.3 µg ml⁻¹.

## ACST image analyses

Image processing of every image time-lapse (all imaging timepoints of a single field) consists of five individual image analysis steps: background control, cell segmentation, cell classification, drift correction and cell tracking.

To increase the accuracy of fluorescence intensity quantifications, we applied BaSiC, a method that automates background correction through low-rank and sparse decomposition via Fiji[19,74]. This method adjusts for spatial and temporal variations in background

fluorescence, which commonly arise from uneven and repetitive illumination. Employing the default BaSiC settings, we achieved consistent fluorescence signals across different conditions, well positions and time (Extended Data Fig. 1b).

Bacteria were segmented using a combined pixel and object classification approach implemented in ilastik[20]. In the pixel classification task, each pixel was subjected to supervised random forest classifiers (100 trees) to differentiate between the background and cellular structures. Within the object classification task, the pixel classification map, estimating the probability of each pixel belonging to the background or cellular class, was smoothed and thresholds were applied to generate object features such as intensity statistics and shape descriptors. These features were used to train the following object classifiers: PI+ single cells, PI− single cells and clumps. Pixel and object classifiers were trained on ~30 time-lapses, including different *M. abscessus* strains and antibiotic conditions. These classifiers were then applied to all data (over 200,000 time-lapses) in batch mode. To assess the fate of PI+ and PI− bacteria upon antibiotic washout, we trained bacteria with ambiguous PI signals as an additional object class. This class was excluded from single-cell growth assessments. To validate segmentation accuracy, 3 randomly selected time-lapse datasets were manually annotated (with automated support) to generate ground-truth pixel prediction maps. The same time-lapses were then processed by ASCT pixel classification. Ground-truth and ASCT pixel classifications were compared using the ImageJ CLIJ2 plugin[74]. Segmentation accuracy (that is, object overlap) was quantified with the Jaccard Index (Extended Data Fig. 1a). To quantify PI classification accuracy, 1,600 PI-positive and 1,600 PI-negative single cells were manually annotated within 5 timelapse datasets, including different *M. abscessus* isolates and treatment conditions. The ASCT PI classification algorithm was than compared with manually annotated objects using the caret R package[75] (Extended Data Fig. 1c).

To optimize downstream analysis and especially bacterial tracking, we corrected for any drift in imaging fields, which mainly occurred at the beginning of each experiment due to thermal changes (thermal drift). The drift of fields was determined by identifying the maximum cross-correlation between two consecutive images by shifting the subsequent image along the $x$ and $y$ coordinates. Drift correction for large-scale time-lapse data is computationally expensive. Therefore, we used smaller field sections (600 × 600 pixels rather than the 2,400 × 2,400 pixels of the whole field of view) and adapted the process to distinct imaging timepoints (frames), given that the drifts were minor after the initial imaging frames. We extracted a window of 600 × 600 pixels from the binary map of segmented objects in frame 0 (centred at $c_1 = (800,800)$), the very first image of a time-lapse. We also extracted a window of 600 × 600 pixels from frame 2 and shifted this section, one pixel at a time, up to 200 pixels away from $c_1$ in each direction, creating 401 × 401 windows. The windows from frame 1 were overlaid on the frame 0 window to find maximum alignment. Given that cells are not always uniformly distributed or can lose focus, the same strategy was performed with two other windows ($c_2 = (1,300,1,300)$ and $c_2 = (1,700,1,700)$). The maximum of the cross-correlation across the 3 locations was used. From frame 3 onward, windows were shifted by fewer pixels to achieve a 75% alignment likelihood between the frames. The shifts were initially set to 25 pixels and consecutively adjusted to 25, 50, 100 and 200 pixels if the alignment likelihood was not achieved.

To automatically follow the behaviours of individual cells during the experiment, we established a custom script to track segmented objects across imaging frames. For each frame, we extracted object features from the segmentation output obtained via brightfield imaging using the MATLAB function 'regionprops'. Our cell-tracking algorithm aims to link corresponding objects between the initial frame 0 and subsequent frames, with $n_t$ denoting an object $n$ at timeframe $t$. Object homology $H$ between frames was determined by comparing features of individual objects. A graph of linking objects ($n_t$, $n_0$) at frame $t$ was represented by a 3-dimensional tensor $H(n_0, n_0 + N, t)$, where $N_0$ is the population of objects at frame 0. The $H$ tensors at each frame were composed of dissociation weights based on the following features: (1) the distances between centroids of objects $n_t$ and $n_0$; (2) the changes in the area of objects; (3) changes in object orientation; and (4) changes in the convex hull area of the object. We assigned higher weight to distances (raised to the power of 4) and a weight of one to the other features.

$$H(n_0, n_t, t) = \text{Dist}(\text{cent}(n_t), \text{cent}(n_0))^4 \frac{|\text{Area}(n_0) - \frac{\text{Area}(n_t)}{\text{d}A/\text{d}t}|}{\text{Area}(n_0)}$$

$$\frac{|\text{CoHull}(n_0) - \frac{\text{CoHull}(n_t)}{\text{d}A/\text{d}t}|}{\text{CoHull}(n_0)} \text{ang}(n_0, n_t) \tag{4}$$

All possible homology values were integrated into an object matrix. Rather than comparing each frame's object to find the closest match, we employed a comprehensive local minimum strategy. The pair of two objects with the highest homology (lowest values in the matrix) was identified as a link in an array and subsequently removed from the matrix. This process was iteratively repeated to identify and remove further links until a homology threshold of $1 \times 10^8$ was reached. This local minimum strategy allowed individual objects to be linked throughout the time-lapse to generate single-cell trajectories. Objects of abnormal size, merging objects, moving objects, objects that increased or decreased in object area, objects with swapping labels, or objects that were not tracked for more than two frames within the time-lapse were discarded. Wells with less than 1,000 tracked bacteria for *M. abscessus* and less than 500 tracked bacteria for *M. tuberculosis* were discarded. The time of PI positivity of a single cell was defined as the first PI+ frame of two consecutive PI+ frames.

## ASCT data analyses

After quantifying the morphology and viability trajectories of each bacterium, the imaging data were further analysed. Bacterial growth in ASCT was defined as an increase in total object area greater than 3.1-fold during antibiotic exposure. This threshold was first defined by comparing growing and non-growing cells and then validated in *M. abscessus* isolates with and without inducible macrolide resistance during azithromycin exposure (Extended Data Fig. 1d). If growth was detected in two-thirds or more of the replicates for a given isolate–drug condition, the condition was considered growing and removed from the killing analyses.

The reproducibility of *M. abscessus* time–kill kinetics was assessed by comparing live-cell fractions at distinct timepoints (3, 6, 9, 12, 24, 36, 48, 60 and 72 h). Exact live-cell fractions at given timepoints were interpolated from overall time–kill kinetics. The coefficient of variation (CoV) was calculated for each triplicate at every timepoint. All triplicates were included in downstream analyses if the mean CoV of all 3 replicates fell within 3 s.d. of the overall mean triplicate CoV distribution per drug condition. Triplicates exceeding this threshold were reduced to the best-performing duplicate. These duplicates were retained if their mean duplicate CoV was within 3 s.d. of the overall mean duplicate CoV per drug condition; otherwise, the isolate–drug pair was excluded from further analyses. A minimum of 2 reproducible time–kill curves per isolate–drug condition was required for downstream analyses of *M. abscessus* kill analyses. To assess for outliers in our *M. abscessus* drug tolerance assessment, we performed principal component analyses for every antibiotic condition on the basis of live-cell fractions at 3, 6, 9, 12, 24, 36, 48, 60 and 72 h. Isolate–drug conditions with a distance of more than 3 s.d. from the mean, assessed with Mahalanobis distance and calculated using the first 2 principal components, were excluded from further analyses. In total, 957 (14.7%) *M. abscessus* isolate–drug pairs were excluded from time–kill analysis due to bacterial growth (mostly azithromycin and minocycline) and 156 (2.4%) due to poor reproducibility or being outliers. Due to the larger

number of replicates, the strategies to account for reproducibility and outliers were not applied to *M. tuberculosis*.

Imaging data were acquired consecutively for each well at different timepoints. The live-cell fraction at 0 h (timepoint 0) was extrapolated by fitting a logistic function to the live-cell fractions from the first 5 imaging frames, corresponding to ~13 h (all conditions except imipenem and cefoxitin). Due to the rapid killing observed in imipenem and cefoxitin conditions, resulting in inaccurate fitting, the maximum live-cell fraction observed for each isolate in any of the other drug conditions was used as the isolate-specific live-cell fraction at 0 h. For *M. abscessus*, isolates with a live-cell fraction below 80% and isolates that had less than 1,000 cells per well across multiple wells were excluded from further analyses. Using the 0-h live-cell fraction, all frames were normalized to an initial live-cell fraction of 100%. Overall, antibiotic killing was quantified as the arithmetic mean of the area under the 72-h (*M. abscessus*) or 168-h (*M. tuberculosis*) time–kill curve, ranging from 1 to 0 (maximum survival to most rapid killing, respectively).

### Drug tolerance analyses

We assessed the relationship between *M. abscessus* drug tolerance phenotypes, growth rates, lag times and MICs using Pearson correlation and $R^2$ values. Spearman correlation was employed to examine the contribution of outliers. For drug clustering in drug tolerance space, a Spearman correlation matrix was generated on the basis of pairwise comparisons of the area under the time–kill curves from 350 clinical *M. abscessus* isolates. Only isolates with 4 or less missing drug tolerance values (out of 16) were used. Principal component analysis was applied to the correlation matrix to visualize drug clustering.

### Single-cell growth assessment

We also applied the tracking algorithm to identify whether single cells are able to grow (form microcolonies) after antibiotic washout. To achieve tracking accuracy and computational efficacy, we employed a backward approach. We identified and categorized objects with a size 5–15 times the median size of single cells as microcolonies and recorded the area, appearance time and centroids. Analogous to the homology matrix to track single non-growing cells, we used the homology index to track microcolonies backwards to the final single-cell object, that is, the initial originating cell. We focused on two features: the object area, which is important particularly to track large objects, which grow and potentially move; and centroids, which are critical to track single non-moving cells. With this approach, we could determine whether specific objects generated microcolonies.

### Whole-genome sequencing

*M. abscessus* isolates were cultured on solid media and colony sweeps were collected[40,76]. DNA extraction was performed using the Qiagen QIAamp DNA mini kit. DNA libraries were constructed with unique identifiers for each isolate and sequenced using multiplexed paired-end sequencing. De novo genome assemblies were assessed for quality. Assemblies with a length longer than 6 Mb, more than 300 contigs, an average depth below 30×, a coverage of the reference genome below 50%, or presumed mixed infection were discarded. Sequence reads were mapped to the *M. abscessus* ATCC-19977 genome using BWA, followed by INDEL realignment[77,78]. Single nucleotide polymorphisms (SNPs) and small insertions/deletions (INDELs) were identified using bcftools and annotated with SNPeff[79,80]. SNPs were filtered to require a minimum base call quality of 50, a minimum mapping quality of 20, and at least 8 matching reads covering an SNP (3 per strand). To assess larger deletions, ATCC-19977 was partitioned into regions of 20 bp with 10 bp overlaps[39]. The coverage of these regions in clinical isolates was assessed with sambamba[81]. Large deletions were defined as 2 consecutive windows with a mean coverage of 5× or below, occurring in at least 5% of all genomes. Variants with identical distributions were collapsed into a single variant. Maximum-likelihood trees were generated

using FastTree, inferred from core SNPs and visualized with iTOL[82,83]. Mutational aminoglycoside resistance was evaluated with mutations in the *rrs* genotype and macrolide resistance with mutations in *rrl*[84,85].

### Heritability estimations

Unitigs for all isolates were extracted using the unitig-caller tool, which utilizes an FM-index built around the Bifrost API[51,86]. A similarity matrix was generated from phylogenetic distances, which was then used to correct for population structure. Using pyseer, FaST linear mixed models were applied to estimate narrow-sense heritability ($h^2$), representing the proportion of variance in the phenotype attributable to genetic variation[50]. Random chance was assessed by shuffling each drug tolerance phenotype across *M. abscessus* isolates and calculating heritability (10 times for each drug tolerance phenotype; 160 times in total).

### Phenogenomic analysis

We performed genome-wide association studies (GWAS) to analyse ~300,000 *M. abscessus* genetic variants, including SNPs, INDELs and large deletions, in relation to drug tolerance phenotypes. Variants were classified by presumed genetic effects: low effect (intergenic variants, synonymous SNPs), moderate effect (non-synonymous SNPs, inframe INDELs) and high effect (frameshift variants, start/stop alterations, large deletions). We applied linear mixed models to account for population structure, integrating a relatedness matrix, and quantified associations using the Wald test[39,87]. We used a Bonferroni threshold of $1.7 \times 10^{-7}$ to control for multiple hypothesis testing. To summarize GWAS hits, we extracted the top 5 genes showing the strongest association per phenotype (for moderate- or high-effect variants). The top associations of these genes with all tolerance phenotypes were then shown in a heat map (moderate- or high-effect variants). Associations were plotted using LocusZoom[88].

### GFP-induction experiments

*M. abscessus* ATCC-19977 was transformed with pUV15tetORm (Addgene, plasmid 17975; gift from Sabine Ehrt) to generate a reporter strain expressing GFP under the control of the tetracycline-inducible promoter Pmyc1tetO[89]. Broth cultures were grown in Middlebrook 7H9 supplemented with hygromycin (1 mg ml⁻¹) until mid-log phase, after which hygromycin was removed. Single-cell suspensions were generated as described for ASCT and treated for 12 h in non-shaking culture conditions with amikacin (26 μg ml⁻¹), cefoxitin (63.1 μg ml⁻¹) or moxifloxacin (8.5 μg ml⁻¹). Following treatment, antibiotics were washed off and bacteria were transferred to the ASCT platform containing 2 μg ml⁻¹ anhydrotetracycline to induce GFP expression. Bright-field and fluorescence images (PI and GFP) were acquired for 12 h. An untreated, induced control sample was used for GFP and PI gating. For each channel, $\log_{10}$-transformed fluorescence intensities were fitted with a 2-component Gaussian mixture model using the Mclust function. Cell counts and proportions were determined per treatment and timepoint. Data were visualized as hexbin density plots (100 bins) of PI versus GFP intensity with overlaid thresholds.

### Drug interaction assessments

To assess *M. tuberculosis* drug combinations, 13 drugs were combined in every pairwise combination using 9 × 9 checkerboards with log√2-fold dilution steps. Each condition was assessed using 4 technical replicates. Drugs were diluted in 7H9 medium, transferred to 384-well plates and inoculated with ~1.2 × 10⁵ c.f.u.s of the H37Ra strain per well. Plates were sealed and incubated at 37 °C with shaking at 250 r.p.m. After 21 days, OD₆₀₀ was assessed and the bacterial fitness was determined as the background-corrected OD₆₀₀ divided by the median OD₆₀₀ of all growth control wells per plate. Single drugs and pairwise dose–response curves were fitted with a 4-parametric log-logistic model (drc R package). Pairwise interaction scores were determined using only the equipotent concentrations[90] and further used to predict

high-order interactions[38]. Bliss interaction scores of high-order combinations were determined by computing the deviation between the predicted high-order curve (derived from measured pairwise interactions) and the bliss-independent expectation of the single-drug effects. Deviations were expressed as $\log_2$ interaction strengths. To evaluate the combined predictive performance of interaction and MIC parameters (median MIC), a logistic regression model was implemented using the glm function in R. Models predicted binary outcomes (similar or better than SOC) in relapsing and bactericidal mouse models. Predictive performance was evaluated across 50 bootstrap iterations (randomly sampling 80% of the conditions), with mean ROC-AUC values assigned to each predictor. Statistical significance was evaluated using a $z$-test against the null hypothesis of random performance (AUC = 0.5).

### Resazurin assay

As an alternative proxy of antibiotic killing, we measured the reduction of resazurin to resorufin in *M. tuberculosis* mc²7000 across all drug regimens. *M. tuberculosis* growth and starvation conditions were prepared as described for ASCT. Approximately $5 \times 10^6$ bacteria suspended in 49.5 µl 7H9 were dispensed per well into clear, flat-bottom 384-well plates and exposed to 0.5 µl drug solution. Drug regimens and concentrations were identical to those tested in ASCT. Each condition was tested in quadruplicate and included medium only and untreated bacterial controls. Plates were sealed with parafilm and incubated at 37 °C in a non-shaking incubator. At days 0, 7 and 14, selected plates were removed and 5 µl of a 0.13% resazurin stock solution was added to each well. After further incubation at 37 °C for 12 h, plates were briefly opened to reduce condensation before fluorescence was measured (excitation 560 nm, emission 590 nm) on a Synergy H1 plate reader (BioTek). Mean relative fluorescence units (RFUs) were background corrected. Antibiotic killing was quantified as the RFU change between days 0 and 14, which was then compared with ASCT-derived killing and in vivo outcomes.

### Generation of the MAB_0233 knockout mutant

*MAB_0233* knockout mutants (Δ*MAB_0233*) were generated on the *M. abscessus* ATCC-19977 background using ORBIT[54]. pKM444 (RecT-Int-expressing plasmid, Kan^r) and pKM496 (for gene deletion by integration into target gene, Zeo^r) were gifted by Kenan Murphy (Addgene, plasmid 108319 and 109301)[54]. pMV261 (replicative vector for gene complementation) was obtained from NovoPro Biosciences[91]. To amplify plasmids, *Escherichia coli* DH5α cells were grown on LB agar plates supplemented with 50 µg ml⁻¹ kanamycin (pKM444 or pMV261) or 50 µg ml⁻¹ zeocin (pKM496).

Liquid cultures of *M. abscessus* ATCC-19977 were incubated at 37 °C and 150 r.p.m. for 2 days. Cells were diluted to an $OD_{600}$ of 0.03 in 20 ml of Middlebrook 7H9, supplemented with 0.4% glycerol, 10% OADC and 0.05% Tween 80 in a 250 ml baffled flask. The culture was placed at 30 °C and 100 r.p.m. for 18–20 h. The culture was then washed 3× with 20 ml ice-cold sterile 10% glycerol supplemented with 0.05% Tween 80 (washing solution). Following the third wash, the cells were collected by centrifugation and resuspended in 200 µl of washing solution. The ORBIT plasmid pKM444 (200 ng) was mixed with competent *M. abscessus* and rested on ice for 5 min. The cells were electroporated (2.5 kV, 1,000 Ω and 25 µF) using a 2-mm-gap-width electroporation cuvette. Following electroporation, the cells were resuspended in 1.5 ml Middlebrook 7H9 medium and placed at 30 °C and 100 r.p.m. for 18–20 h. The recovered cells were pelleted at 3,000 *g* (25 °C, 5 min), resuspended in 100 µl Middlebrook 7H9 medium and plated on 7H11 plates supplemented with 0.5% glycerol, 10% OADC and 250 µg ml⁻¹ kanamycin. The plates were incubated at 30 °C for 3–4 days.

Colonies resulting from pKM444 transformation were picked and grown on Middlebrook 7H11 agar plates supplemented with 0.5% glycerol, 10% OADC and 250 µg ml⁻¹ kanamycin. Cells were collected by centrifugation and resuspended in TRIzol (Thermo Fisher). Cell disruption was performed by bead beating using Lysing Matrix B

beads (200 µl) in a FastPrep-24 Classic instrument at 6.5 m s⁻¹ for 60 s, followed by 60 s incubation on ice. This process was repeated for 3 cycles. Total DNA was extracted using the DNA miniprep kit (Zymo Research). The presence of the pKM444 plasmid was verified by PCR, using Dream*Taq* DNA polymerase (Thermo Fisher). Primers specific to the plasmid (forward: CACGTTGTGTCTCAAAATCTC; reverse: CGATAACGTTCTCGGCTC) were used to amplify a target segment of the plasmid. PCR products were resolved by agarose gel electrophoresis, and bands of the expected size were excised and submitted for Sanger sequencing to verify plasmid presence.

The oligonucleotide sequence consists of the 48 bp Bxb1 *attP* site (or the reverse complement) flanked by 60 bp upstream of the initiation codon and 60 bp downstream of the stop codon[54]. The targeting oligonucleotide (sequence for *MAB_0233*: CGGCCACATGTTTTGTGCCGCTAGGGGAAATCAGCTCGGCATCGTCCGTGTGCCGTTGTTGGTTTGTACCGTACACCACTGAGACCGCGGTGGTTGACCAGACAAACCCACATACCCTGATGCGAGTTCAACTGCCATCTGTTGCCCCCTTCGGGCAATCGGTGCAGC) was acquired from IDT with 4 nmole Ultramer DNA Oligo property and diluted in nuclease-free water to 1 µg µl⁻¹.

*M. abscessus* ATCC-19977, harbouring the pKM444 plasmid, was cultured in Middlebrook 7H9 medium with 5 µg ml⁻¹ anhydrotetracycline (ATc). A 20 ml culture was prepared in a 250 ml baffled flask wrapped in aluminum, washed as described above and resuspended in 200 µl washing solution. Oligonucleotide (2 µl, 1 µg µl⁻¹) guiding the payload plasmid pKM496 to the target gene was aliquoted into 1.5 ml microcentrifuge tubes and denatured at 95 °C for 5 min to prevent secondary structure formation. The denatured oligonucleotides were cooled on ice for at least 5 min before adding 200 ng pKM496 (ref. 54). Subsequently, 200 µl of cells were added to the tube, gently mixed by pipetting and incubated for 5 min. Electroporation, recovery and plating were performed as described above, except that recovered cells were plated on 7H11 plates supplemented with 100 µg ml⁻¹ zeocin. DNA extraction and PCR were performed as described using primers flanking *MAB_0233* and pKM496 (forward: CGCTCACAACTGAATACCC; reverse: CCTGGTATCTTTATAGTCCTGTC).

### Gene complementation

To complement Δ*MAB_0233*, the gene was amplified by PCR (KAPA HiFi, high-fidelity enzyme) and cloned into pMV261 downstream of the mycobacterial strong constitutive promoter, *pHSP60* (ref. 91). In addition, the T2 terminating region from *E. coli rrnB* was added to the end of the open reading frame (ORF) as an efficient terminator of transcription. To eliminate the pKM444 plasmid, which confers kanamycin resistance, the *M. abscessus* knockout strain Δ*MAB_0233* was subjected to serial dilutions and growth cycles until cured. Cured strains were transformed with pMV261 following the same transformation protocol as described for pKM444.

As control, MAB-ATCC-pKM444 was used. The RNA of log phase cultures of the control strain, Δ*MAB_0233* and Δ*MAB_0233*::*MAB_0233* was isolated using the RNA miniprep kit (Zymo Research). Complementary (c)DNA was prepared using the High-Capacity cDNA Reverse Transcription kit (Applied Biosystems). cDNA levels were quantified by quantitative real-time PCR (qPCR) on an Applied Biosystems qPCR machine using a PowerUp SYBR Green Master Mix (Thermo Fisher) and analysed by the $\Delta\Delta C_t$ method. Gene expression was controlled using the housekeeping genes *MAB_3009* (sigA) and *MAB_3869c*. Two sets of primers covering ~150 bp sequences of the beginning and the end of the *MAB_0233* ORF were designed (primer_1_FWD: GCGAAGCCTTCGCCAAAGCTC, primer_1_REV: CCTCGTTGAGCTTTTCCAGCGC, primer_2_FWD: CGCAACGTCTCCGATGGGAAACC, primer_2_REV: GAGTTGAGCGGCGTCCATGC).

### Reporting summary

Further information on research design is available in the Nature Portfolio Reporting Summary linked to this article.

## Data availability

Whole-genome sequence data are available at the European Nucleotide Archive (ENA). Representative images to test ASCT are available on Zenodo at https://doi.org/10.5281/zenodo.17232777 (ref. 92). The full raw imaging dataset is not publicly deposited due to size constraints but is available from the lead contact upon request. All downstream imaging data and other data are available in source data.

## Code availability

All original code has been deposited on GitHub at https://github.com/BoeckLab/ASCT and archived at Zenodo under https://doi.org/10.5281/zenodo.17775709 (ref. 93).

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

## Acknowledgements

We thank B. Schulthess and P. Sander for training in MIC assessments; A. Trauner, D. Pinschewer, M. Zampieri, A. Harms, D. Portevin, B. Aldridge, D. Bumann and U. Jenal for expert input; the scientific computing center at the University of Basel (sciCORE) and specifically R. M. Cabezón, I. M. de Ilarduya, G. Losilla and M. Jacquot for computational support; J. Sollier for critical reading and editing of the paper; the DBM microscopy core facility; and M. Roth and M. Tamm for lab space. Icons were obtained from PhyloPic (http://phylopic.org). L.B. received funding from the Swiss National Science Foundation (grant nos 177799, 185792, 215557), the Bangerter–Rhyner Foundation, Goldschmidt Jacobson Foundation, Helmut Horten Foundation, Swiss Society for Pneumology, Cloëtta Foundation, and NCCR AntiResist, a National Center of Competence in Research, funded by the Swiss National Science Foundation (grant no. 180541). R.A.F and D.M.G. received funding from the Wellcome Discovery award 226602/Z/22/Z and LifeArc/CF Trust Innovation Hub THUB01.

## Author contributions

L.B. conceptualized the project. A.J. and L.B. curated data. A.S., A.J. and L.B. conducted formal analysis. L.B. acquired funding. A.S., A.J., F.K.B., B.W., S.E.C.M., G.C.G., S.T., L.S., A.R., A.W., A.L., S.B. and L.B. conducted investigations. A.S., A.J., F.K.B. and L.B. designed the methodology. L.B. administered and supervised the project. A.S., A.J., T.P. and L.B. designed software. D.M.G., N.E.W., P.D., J.N., H.P., R.T., S.C.B., S.G., J.M.B., A.H.D., R.A.F., M.A. and L.B. procured resources. A.S., A.J., F.K.B. and L.B. performed validation. A.S., A.J., S.E.C.M. and L.B. performed visualization. L.B. wrote the original paper draft. A.S., A.J., F.K.B., B.W., S.E.C.M., G.C.G., S.T., L.S., A.R., A.W., A.L., S.B., D.M.G., N.E.W., P.D., J.N., H.P., R.T., S.C.B., S.G., J.M.B., T.P., A.H.D., R.A.F., M.A. and L.B. reviewed and edited the paper.

## Competing interests

The authors declare no competing interests.

## Additional information

**Extended data** is available for this paper at https://doi.org/10.1038/s41564-025-02217-y.

**Correspondence and requests for materials** should be addressed to Lucas Boeck.

Alexander Jovanovic[1,17], Frederick K. Bright[1,17], Ahmad Sadeghi[1,17], Basil Wicki ®[1], Santiago E. Caño Muñiz[1], Greta C. Giannini ®[1], Sara Toprak[1], Loïc Sauteur ®[1], Anna Rodoni[1], Andreas Wüst[1], Andréanne Lupien[2], Sonia Borrell[3,4], Dorothy M. Grogono ®[5], Nicole E. Wheeler ®[6], Philippe Dehio[1], Johannes Nemeth ®[7], Hans Pargger ®[8], Rachel Thomson ®[9,10], Scott C. Bell ®[11,12], Sebastien Gagneux ®[3,4], Josephine M. Bryant ®[13], Tingying Peng[14], Andreas H. Diacon[15], R. Andres Floto ®[5,16], Michael Abanto[1] & Lucas Boeck ®[1,8] ✉

¹Department of Biomedicine, University of Basel, Basel, Switzerland. ²Department of Medicine, McGill University, McGill International TB Centre, Montreal, Quebec, Canada. ³Swiss Tropical and Public Health Institute, Allschwil, Switzerland. ⁴University of Basel, Basel, Switzerland. ⁵Cambridge Centre for Lung Infection, Royal Papworth Hospital, Cambridge, UK. ⁶Institute of Microbiology and Infection, University of Birmingham, Birmingham, UK. ⁷Department of Infectious Diseases and Hospital Epidemiology, University Hospital Zürich, Zurich, Switzerland. ⁸Pulmonary Medicine, University Hospital

Basel, Basel, Switzerland. [9]Gallipoli Medical Research, Greenslopes Private Hospital, Brisbane, Queensland, Australia. [10]Greenslopes Clinical School, Faculty of Health, Medicine and Behavioural Sciences, The University of Queensland, Brisbane, Queensland, Australia. [11]School of Medicine and Dentistry, Griffith University, Southport, Queensland, Australia. [12]Gold Coast University Hospital, Southport, Queensland, Australia. [13]Wellcome Sanger Institute, Hinxton, UK. [14]Helmholtz Munich – German Research Center for Environment and Health, Munich, Germany. [15]TASK, Cape Town, South Africa. [16]Molecular Immunity Unit, University of Cambridge Department of Medicine, MRC Laboratory of Molecular Biology, Cambridge, UK. [17]These authors contributed equally: Alexander Jovanovic, Frederick K. Bright, Ahmad Sadeghi. ✉e-mail: lucas.boeck@unibas.ch

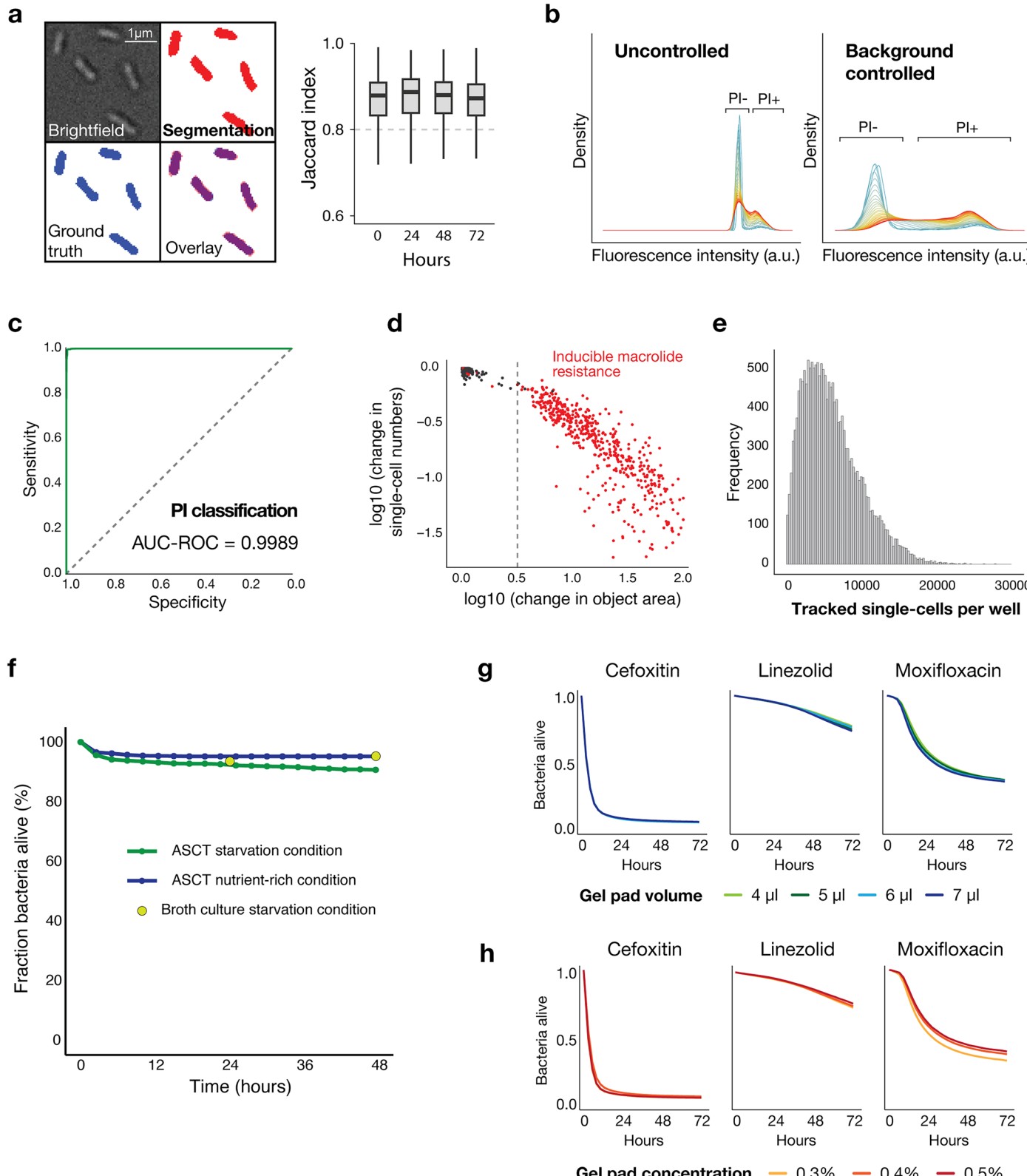

**Extended Data Fig. 1 | See next page for caption.**

**Extended Data Fig. 1 | Antimicrobial Single-Cell Testing.** (**a**) Overlay of manually annotated "ground truth" bacterial segmentation with automated object segmentation (pixel classification) of *M. abscessus* brightfield images. The Jaccard index quantifies overlay accuracy across imaging time points during time-lapse acquisition of 5,887 bacteria. Boxplots show the median, interquartile range, and data range within 1.5 x IQR (central line, box, and whiskers, respectively) of Jaccard indices. The dashed line indicates high segmentation accuracy (Jaccard index of 0.8). (**b**) Changes in dynamic range following BaSiC background correction. Mean PI values per single *M. abscessus* bacterium before and after correction over 72-hour time-lapses (first imaging time point: blue, last time point: red) shown in density plots. (**c**) Accuracy of automated PI classification in predicting "ground-truth" classes of PI-positive and PI-negative single cells. (**d**) Changes in single-cell count (which declines during growth) and object area (assessed with ASCT) over 72 hours under azithromycin treatment across 406 *M. abscessus* isolates. Isolates with (red) and without (black) inducible macrolide resistance are highlighted. The dotted line indicates the ASCT growth threshold. (**e**) Number of single *M. abscessus* bacteria tracked over 72 hours per well within 1536-plates. Data from 406 *M. abscessus* isolates, and about 130 million tracked bacteria across eight drugs and two concentrations (data from Fig. 3). (**f**) Fraction of viable *M. abscessus* immobilised in ASCT under nutrient-rich and starvation conditions without antibiotic exposure (dividing and PI-negative cells were considered viable). Yellow dots indicate the fraction of PI-negative bacteria in starved broth culture at 24 and 48 hours (no single-cell tracking). Effect of different (**g**) gel pad volumes and (**h**) gel pad agarose concentrations on *M. abscessus* time-kill kinetics.

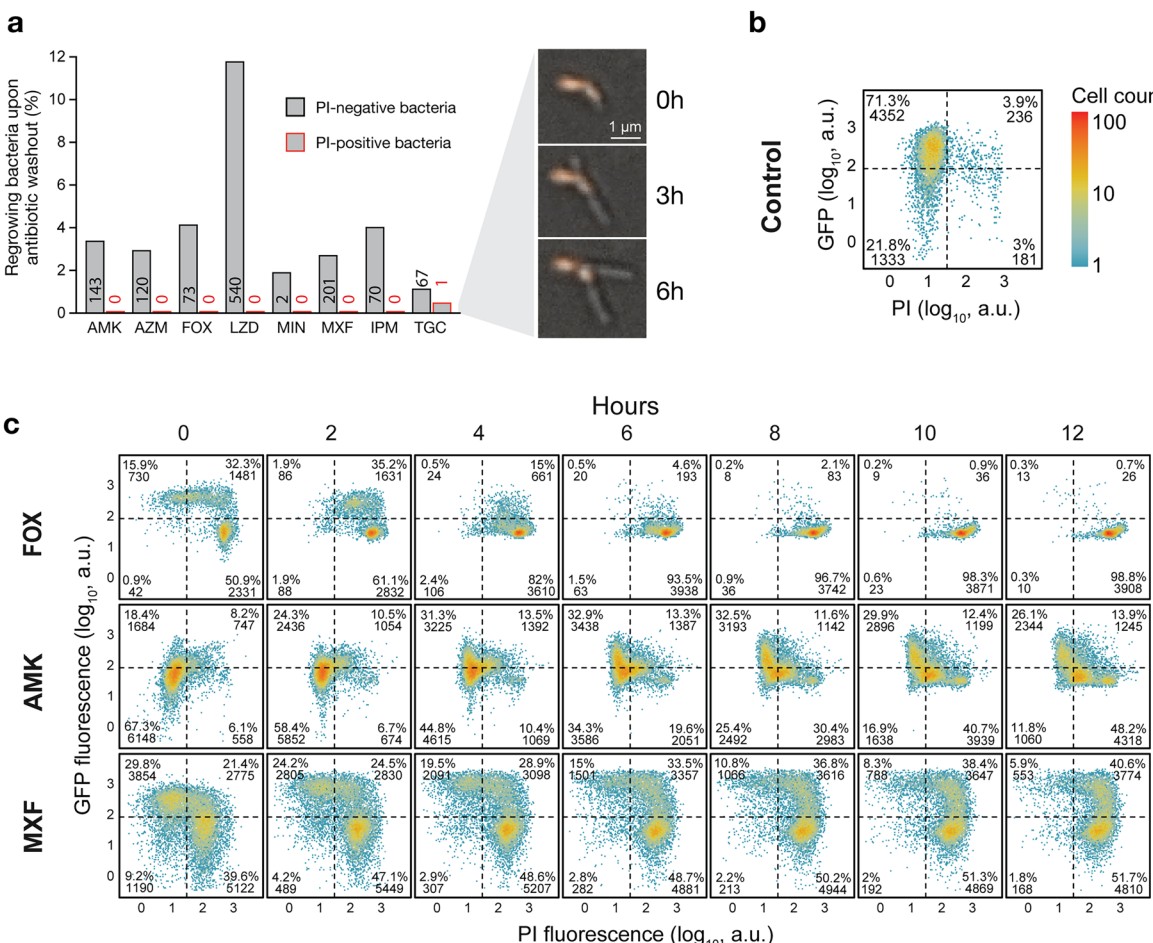

**Extended Data Fig. 2 | Benchmarking propidium iodide as a mycobacterial viability marker.** (**a**) Fraction of bacteria (including numbers) showing regrowth after 24-hour antibiotic treatment and antibiotic washout across PI-negative and PI-positive bacteria. The single regrowing PI-positive bacterium is highlighted in the insert. (**b-c**) Induced GFP expression after antibiotic washout. (**b**) Image gating strategy for GFP and PI in untreated *M. abscessus* controls. (**c**) GFP and PI fluorescence intensities of *M. abscessus* populations treated with amikacin (AMK), cefoxitin (FOX), or moxifloxacin (MXF) following antibiotic removal. Time (hours) indicates imaging time points, with 0 set to the start of imaging ( ~ 1 hour after GFP induction).

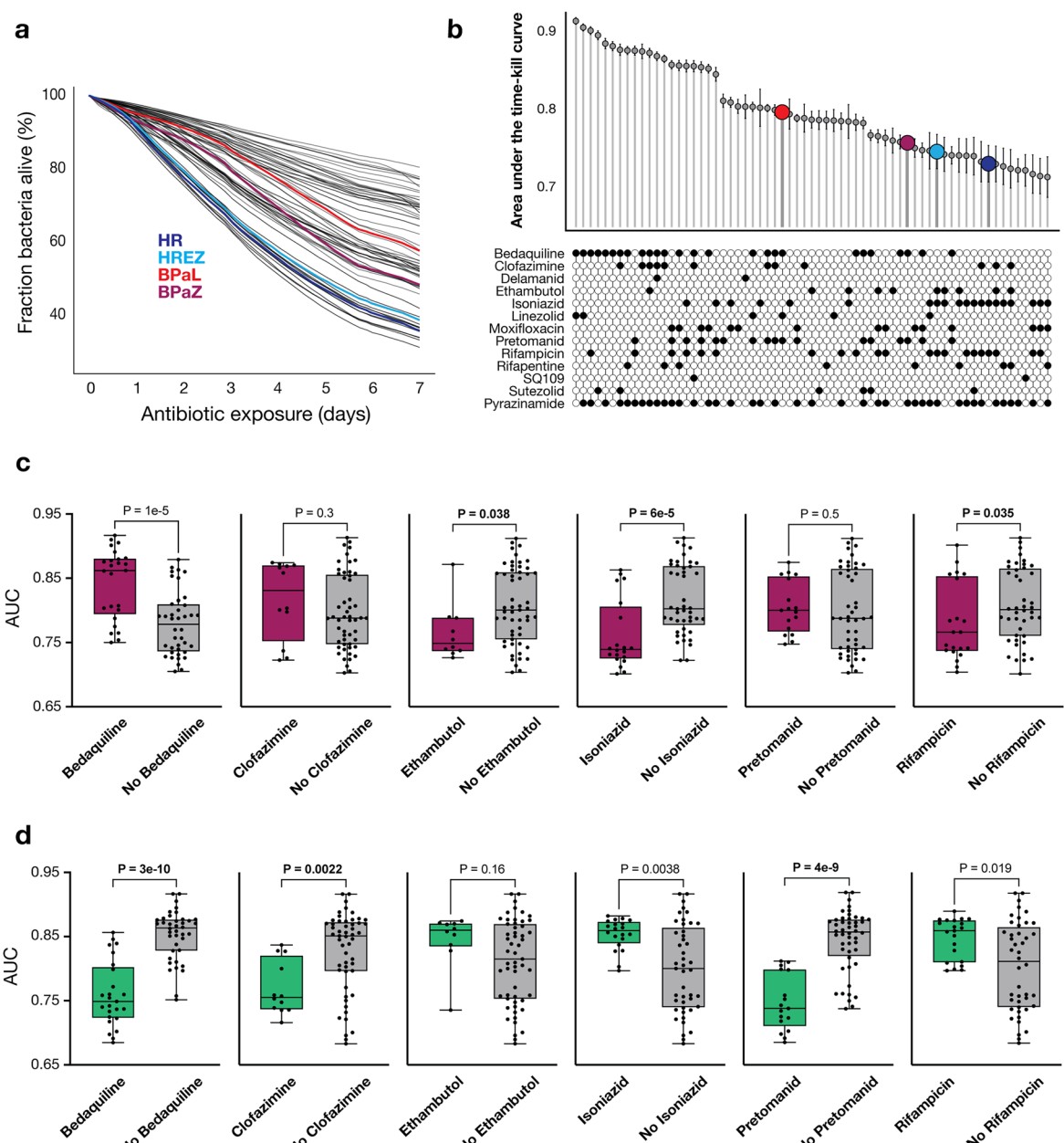

**Extended Data Fig. 3 | *M. tuberculosis* time-kill kinetics. (a)** ASCT-based time-kill kinetics of growing *M. tuberculosis* (H37Ra) exposed to 65 drug regimens, with the following regimens highlighted: isoniazid-rifampicin (HR; dark blue), isoniazid-rifampicin-ethambutol-pyrazinamide (HREZ; light blue), bedaquiline-pretomanid-linezolid (BPaL; red), and bedaquiline-pretomanid-pyrazinamide (BPaZ; purple). **(b)** Mean area under the kill curve for *M. tuberculosis* drug regimens under growth conditions, averaged across the two *M. tuberculosis* strains (H37Ra and mc²7000). Each drug regimen was tested in nine technical replicates (poor-quality curves were excluded), and data are shown as mean ± SEM. **(c-d)** Killing dynamics (AUCs) of *M. tuberculosis* drug combinations were assessed in regard to the presence and absence of individual drugs. Drugs that increased killing under **(c)** growth conditions (averaged from mc²7000 and H37Ra strains) or **(d)** starvation conditions (averaged from three starvation conditions) are shown. Each condition was tested in nine technical replicates (poor-quality curves were excluded). Groups were compared using a two-sided Mann-Whitney U test. Each dot indicates a drug regimen. Boxplots show the median, interquartile range, and total range (central line, box, and whiskers, respectively) of AUC values.

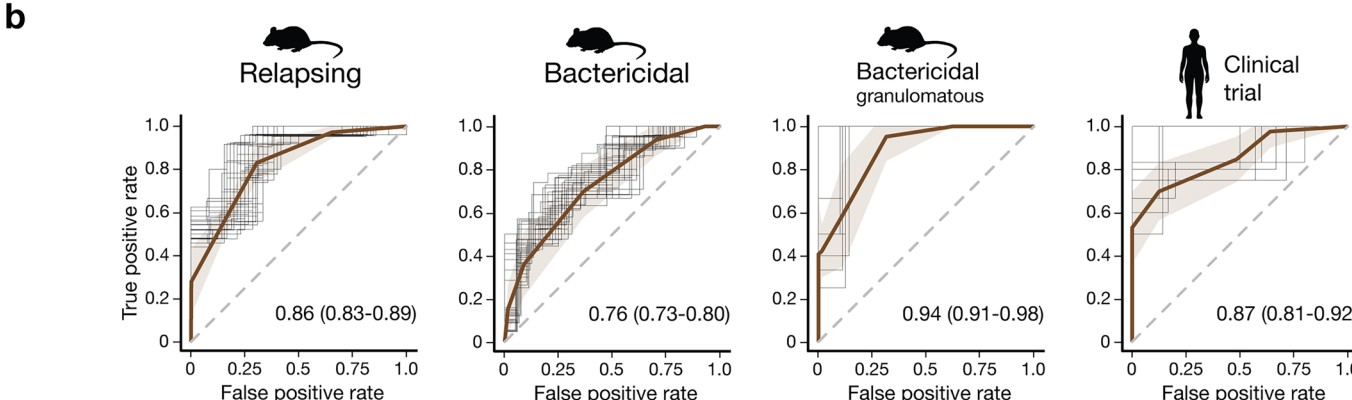

**Extended Data Fig. 4 | Association of ASCT-based killing assessments with *in vivo* outcomes of *M. tuberculosis* drug regimens.** (**a**) Time-kill curves (AUCs) upon growth and starvation conditions of two *M. tuberculosis* isolates were used to compare similar-to-SOC versus better-than-SOC classifications, based on outcomes in relapsing mouse models (RMM), bactericidal mouse models (BMM) of common mouse strains, the granulomatous C3HeB/FeJ strain and clinical studies[30,31]. Comparisons were performed using a two-sided Mann-Whitney U test (P values indicated). Each dot represents a drug regimen. Boxplots show the median, interquartile range and total range (central line, box, and whiskers,

respectively) of AUC values. (RMM: n = 46, BMM common strains: n = 48, BMM C3HeB/FeJ: n = 15, clinical bactericidal activity: n = 14). PA indicates pantothenate, PBS phosphate-buffered saline. (**b**) Performance of time-kill kinetics (AUC averaged across three *M. tuberculosis* starvation models) for predicting *in vivo M. tuberculosis* outcomes (similar-to-SOC versus better-than-SOC drug regimens). Using logistic regression, fifty ROC curves were generated for every condition by randomly selecting 80% of the samples. The brown line represents the mean receiver operating characteristic (ROC) curve, the shaded area one standard deviation. Mean area under the ROC curves and 95% confidence intervals are presented.

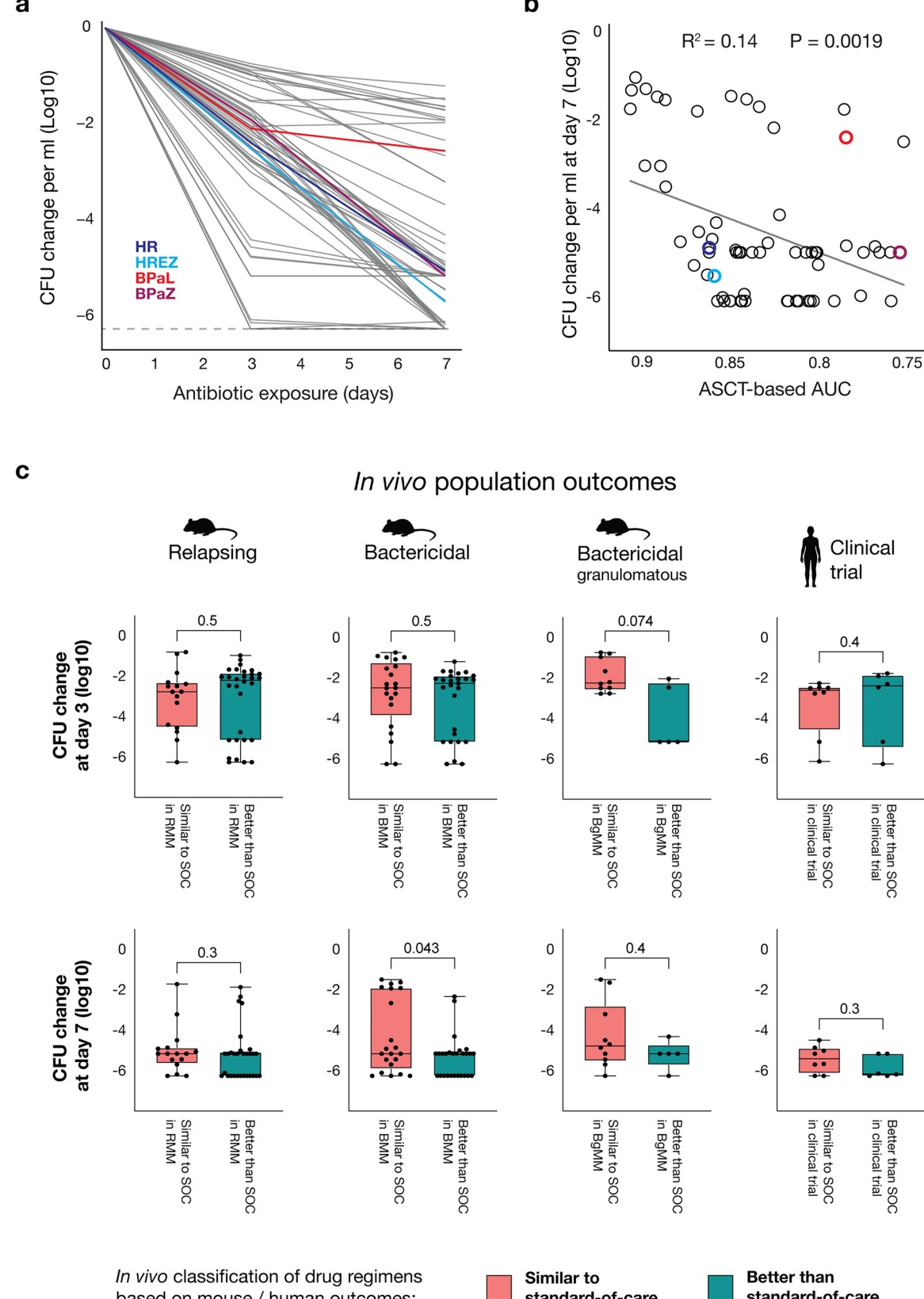

**Extended Data Fig. 5 | See next page for caption.**

**Extended Data Fig. 5 | Association of CFU-based killing assessments with *in vivo* outcomes of *M. tuberculosis* drug regimens.** (**a**) Colony forming unit (CFU) change of PBS-starved *M. tuberculosis* mc²7000 during drug treatment. 65 drug regimens were tested. (**b**) Correlation of CFU-based killing of 65 drug regimens with ASCT-based time-kill kinetics, both assessed in the PBS-starved *M. tuberculosis* mc²7000 strain and quantified with two-sided Pearson correlation. (**c**) CFU-based killing of PBS-starved *M. tuberculosis* mc²7000 at day 3 and 7 was used to compare similar-to-SOC versus better-than-SOC classifications, based on outcomes in relapsing mouse models (RMM), bactericidal mouse models (BMM) of common mouse strains, the granulomatous C3HeB/FeJ strain and clinical studies[30,31]. Comparisons were performed using a two-sided Mann-Whitney U test (P values indicated). Each dot represents a drug regimen. Boxplots show the median, interquartile range and total range (central line, box, and whiskers, respectively; RMM: n = 46, BMM common strains: n = 48, BMM C3HeB/FeJ: n = 15, clinical bactericidal activity: n = 14).

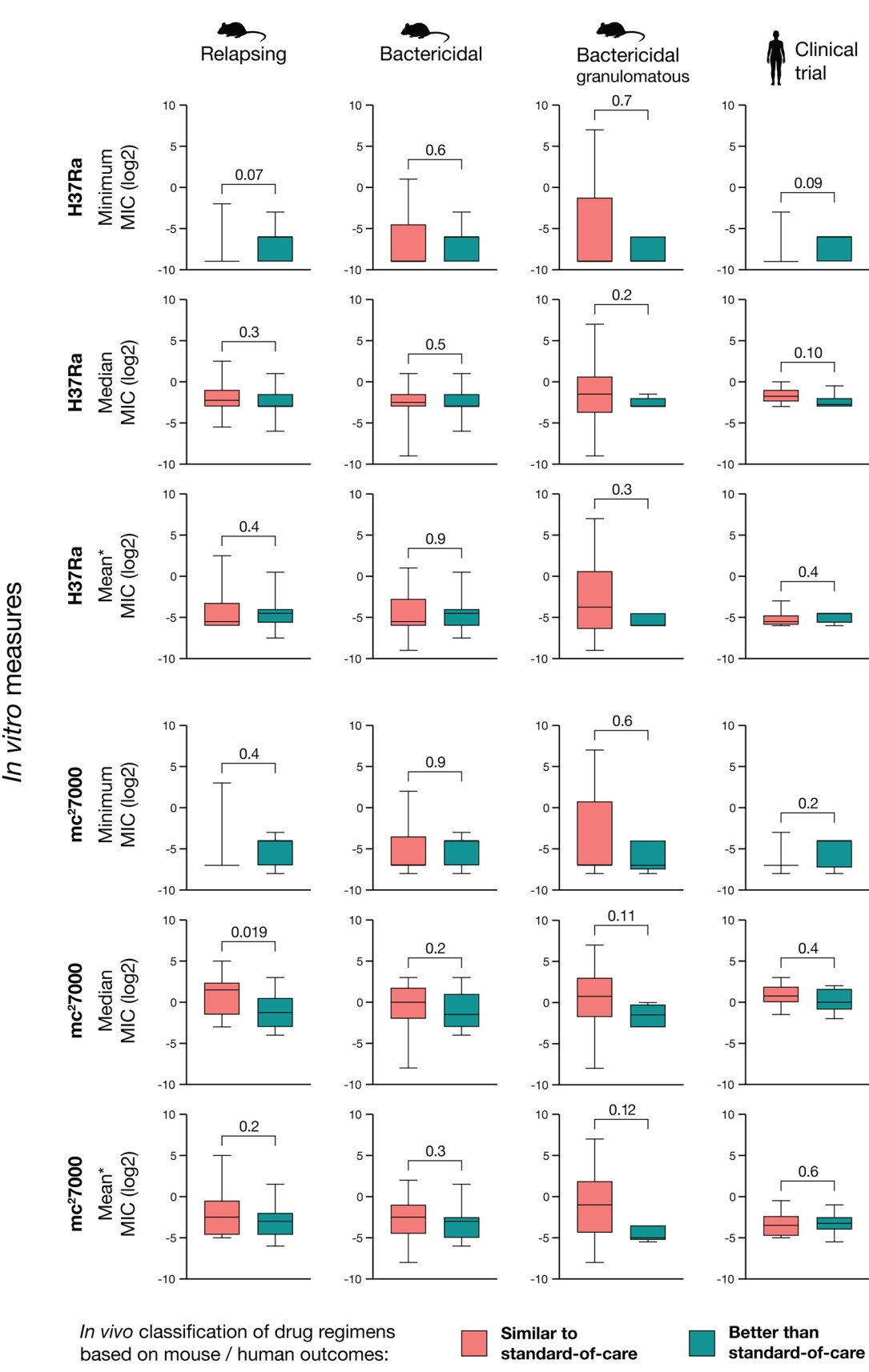

**Extended Data Fig. 6 | See next page for caption.**

**Extended Data Fig. 6 | Association of minimum inhibitory concentrations (MICs) with *in vivo* outcomes of *M. tuberculosis* drug regimens.** The minimum MIC or median MIC of all drugs within each combination, or the mean MIC of the two most potent drugs (mean*), was assessed and compared between similar-to-SOC and better-than-SOC, based on outcomes in relapsing mouse models (RMM), bactericidal mouse models (BMM) of common mouse strains, the granulomatous C3HeB/FeJ strain and clinical studies[30,31]. Comparisons were performed using a two-sided Mann-Whitney U test (P values indicated). Each dot represents a drug regimen. Boxplots show the median, interquartile range and total range (central line, box, and whiskers, respectively) of MIC values (RMM: n = 46, BMM common strains: n = 48, BMM C3HeB/FeJ: n = 15, clinical bactericidal activity: n = 14).

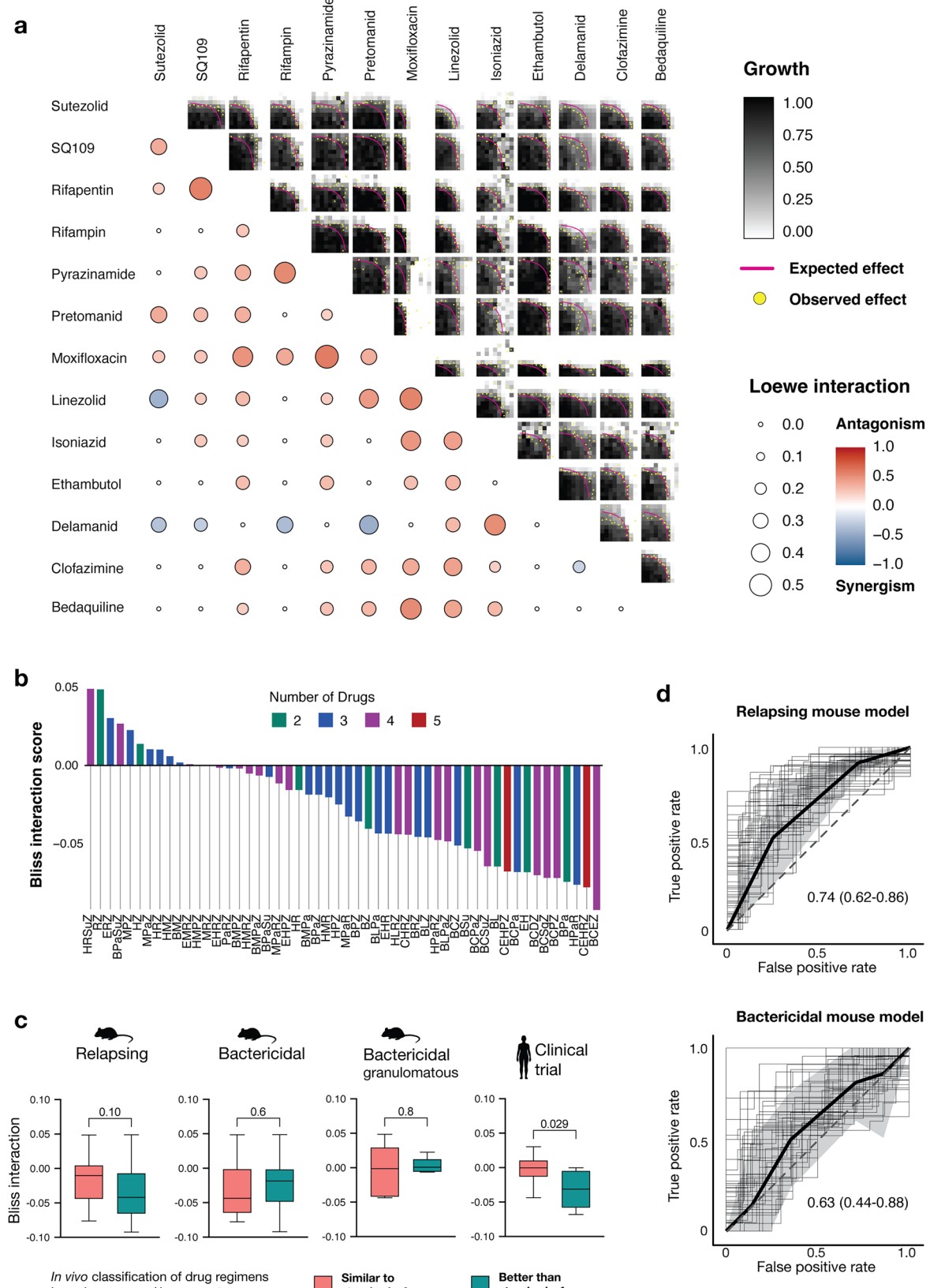

**Extended Data Fig. 7 | See next page for caption.**

**Extended Data Fig. 7 | *M. tuberculosis* drug interactions. (a)** Pairwise *M. tuberculosis* drug-drug interactions of 13 x 12 drugs, each performed in 9x9 checkerboards, highlighting expected and observed growth inhibition. Pairwise interactions are quantified by the Loewe interaction score. **(b)** Measured pairwise and predicted high-order drug interactions of 52 *M. tuberculosis* drug regimens. B indicates bedaquiline, C clofazimine, D delamanid, E ethambutol, H isoniazid, L linezolid, M moxifloxacin, Pa pretomanid, Z pyrazinamide, R rifampicin, P rifapentine, Sq SQ109 and Su sutezolid. **(c)** Drug interactions were compared across similar-to-SOC versus better-than-SOC classifications, based on outcomes in relapsing mouse models (RMM), bactericidal mouse models (BMM) of common mouse strains, the granulomatous C3HeB/FeJ strain and clinical

studies [30,31]. Comparisons were performed using a two-sided Mann-Whitney U test (P values indicated). Boxplots show the median, interquartile range and total range (central line, box, and whiskers, respectively; RMM: n = 46, BMM common strains: n = 48, BMM C3HeB/FeJ: n = 15, clinical bactericidal activity: n = 14). **(d)** The performance of combined MIC (median regimen MIC) and drug interaction measures for predicting *in vivo outcomes* was assessed using logistic regression models. Fifty ROC curves were generated for every condition by randomly selecting 80% of the samples. The black line represents the mean receiver operating characteristic (ROC) curve, the shaded area one standard deviation. Mean area under the ROC curves and 95% confidence intervals are presented.

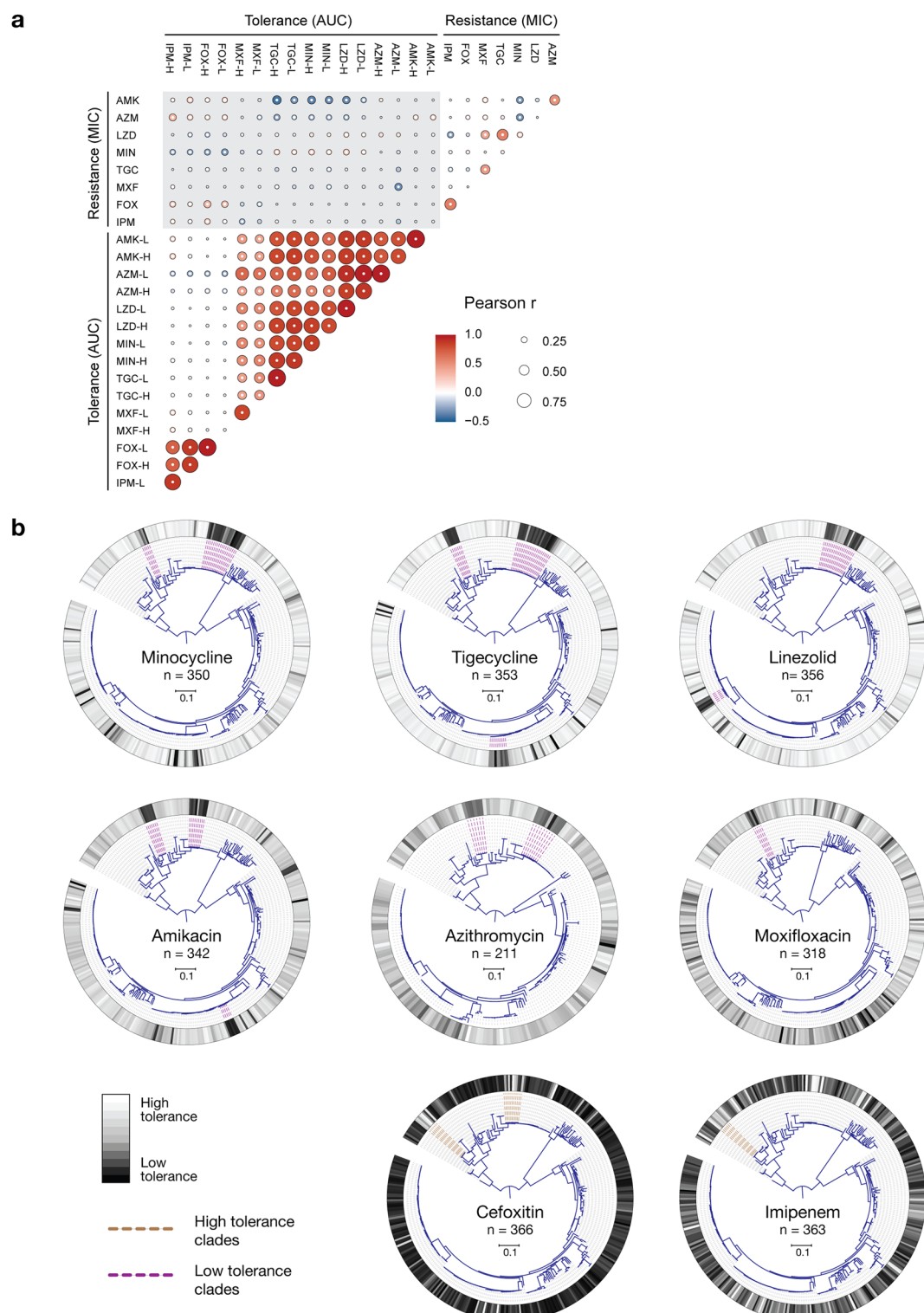

**Extended Data Fig. 8 | Drug correlations and tolerance mapping to the *M. abscessus* phylogeny.** (**a**) Pearson correlation of *M. abscessus* drug resistance and tolerance phenotypes. Correlation strength is indicated be circle size, correlation direction by colour. White central dots highlight significant correlations (two-sided Pearson test, p < 0.05). AUC indicates the area under the kill curve, MIC minimum inhibitory concentration, AMK amikacin, AZM azithromycin, FOX cefoxitin, IPM imipenem, LZD linezolid, MIN minocycline, MXF moxifloxacin, TGC tigecycline, L refers to low drug concentration, and H to high drug concentration. (**b**) *M. abscessus* phylogenetic trees (maximum-likelihood trees) aligned with drug tolerance heatmaps.

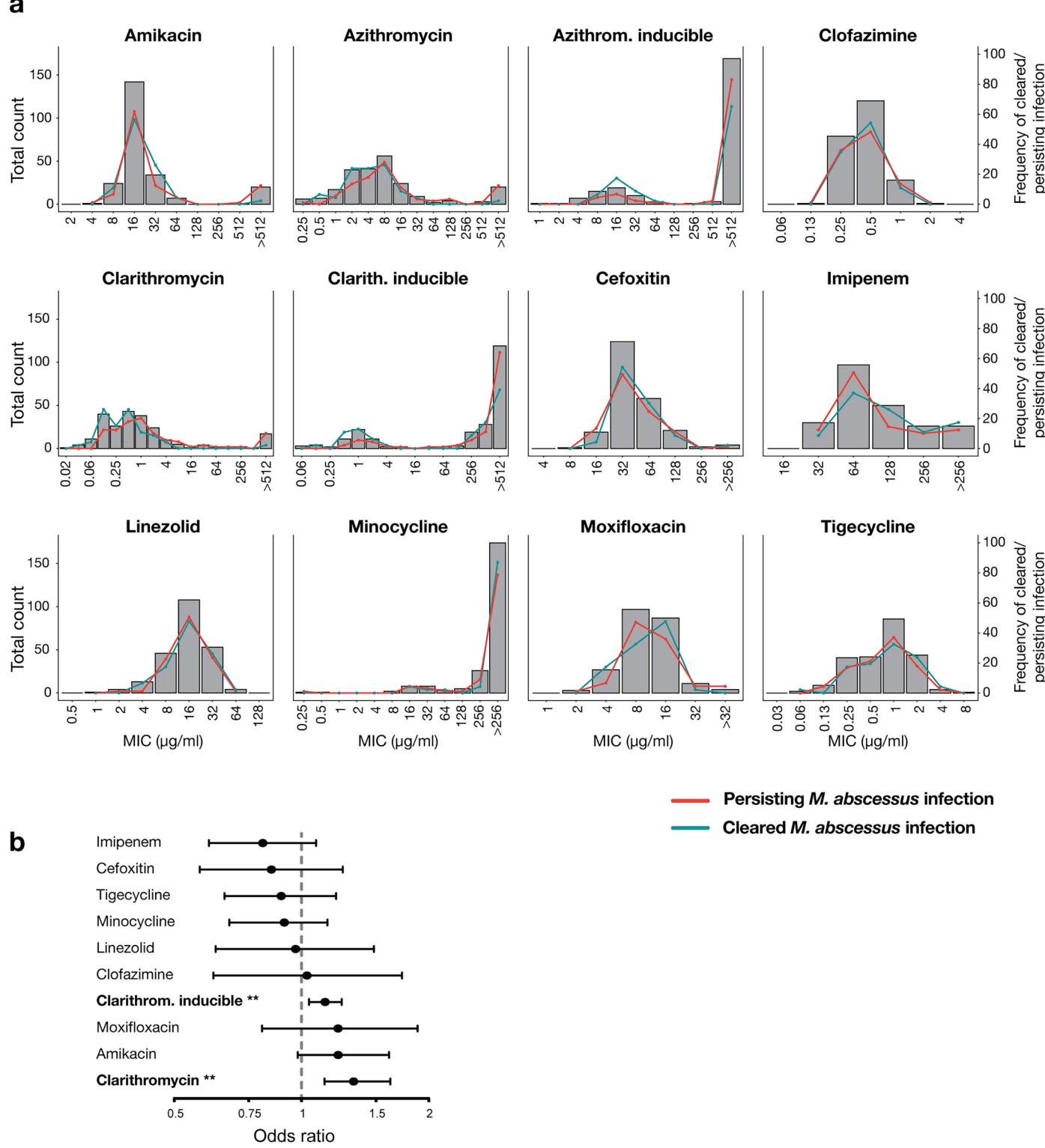

**Extended Data Fig. 9 | Association of *M. abscessus* MICs with individual patient outcomes.** (**a**) MIC distributions of 229 clinical isolates and the frequency of isolates associated with cleared (n = 46) and persisting infection (n = 89). (**b**) Odds ratios of *M. abscessus* minimum inhibitory concentrations for predicting treatment failures (lack of culture conversion) in 135 patients.

Odds ratios of individual MIC measures were assessed with logistic regression (clarithromycin: P = 0.0019; clarithromycin inducible: P = 0.0045). Each dot represents the odds ratio, and whiskers highlight the 95% confidence interval. ** indicates the two-sided P < 0.01.

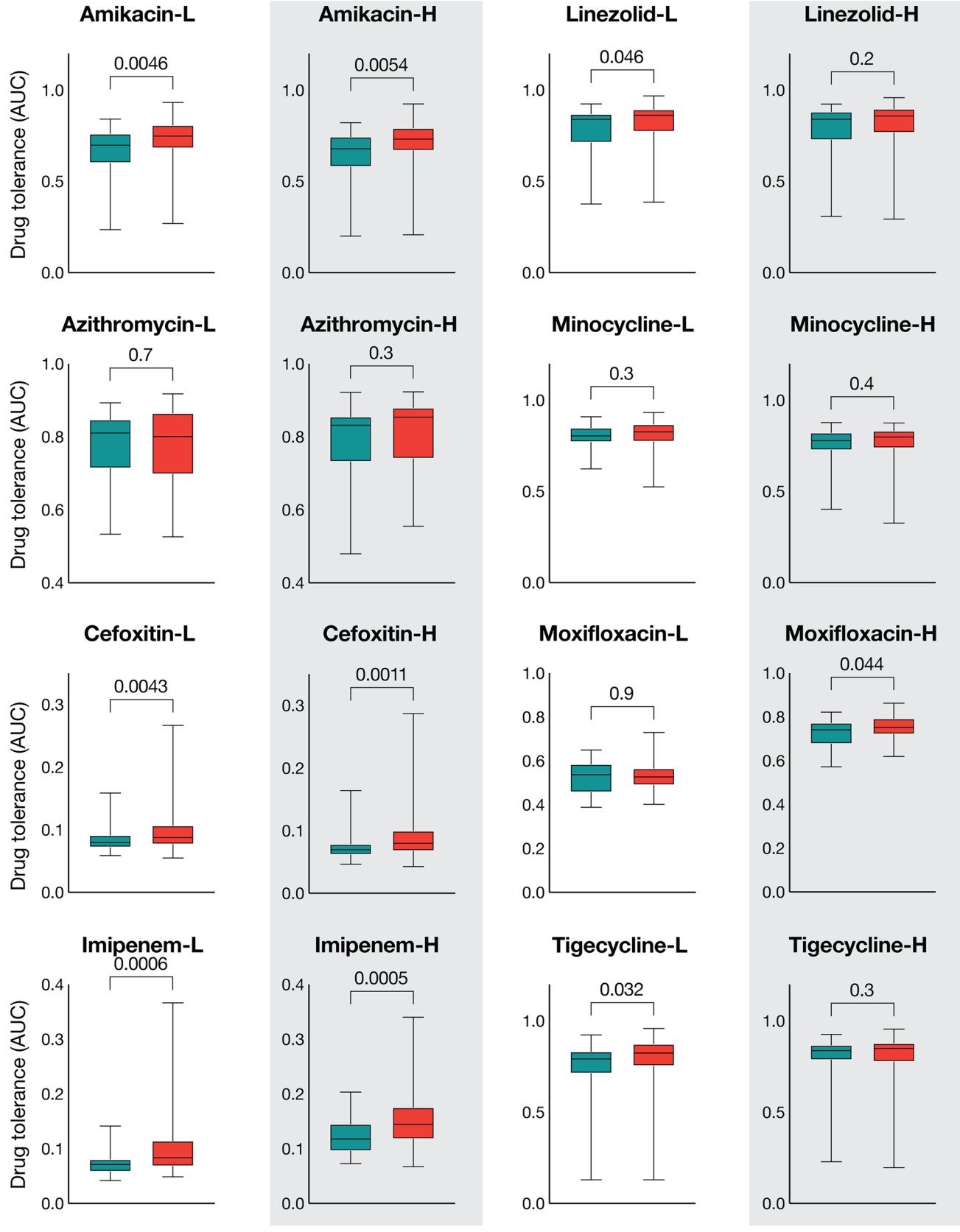

**Persisting *M. abscessus* infection**   **Cleared *M. abscessus* infection**

**Extended Data Fig. 10 | Association of *M. abscessus* drug tolerance phenotypes with individual patient outcomes.** Comparison of drug tolerance between *M. abscessus* isolates from patients with poor versus favourable clinical outcomes (n = 135, excluding growing isolates), using a two-sided Mann-Whitney U test (P indicated). Boxplots show the median, interquartile range and total range (central line, box, and whiskers, respectively) of AUC values. L refers to low drug concentration, and H to high drug concentration.

# Reporting Summary

## Statistics

For all statistical analyses, confirm that the following items are present in the figure legend, table legend, main text, or Methods section.

| n/a | Confirmed | |
|---|---|---|
| ☐ | ☒ | The exact sample size (*n*) for each experimental group/condition, given as a discrete number and unit of measurement |
| ☐ | ☒ | A statement on whether measurements were taken from distinct samples or whether the same sample was measured repeatedly |
| ☐ | ☒ | The statistical test(s) used AND whether they are one- or two-sided *Only common tests should be described solely by name; describe more complex techniques in the Methods section.* |
| ☒ | ☐ | A description of all covariates tested |
| ☐ | ☒ | A description of any assumptions or corrections, such as tests of normality and adjustment for multiple comparisons |
| ☐ | ☒ | A full description of the statistical parameters including central tendency (e.g. means) or other basic estimates (e.g. regression coefficient) AND variation (e.g. standard deviation) or associated estimates of uncertainty (e.g. confidence intervals) |
| ☐ | ☒ | For null hypothesis testing, the test statistic (e.g. *F*, *t*, *r*) with confidence intervals, effect sizes, degrees of freedom and *P* value noted *Give P values as exact values whenever suitable.* |
| ☒ | ☐ | For Bayesian analysis, information on the choice of priors and Markov chain Monte Carlo settings |
| ☐ | ☒ | For hierarchical and complex designs, identification of the appropriate level for tests and full reporting of outcomes |
| ☐ | ☒ | Estimates of effect sizes (e.g. Cohen's *d*, Pearson's *r*), indicating how they were calculated |

*Our web collection on statistics for biologists contains articles on many of the points above.*

## Software and code

Policy information about availability of computer code

| Data collection | No software was used for data collection. |
|---|---|
| Data analysis | BaSiC v1, Bcftools v1.14, FastTree v2.1, Fiji v2.9.9, Genome Analysis Toolkit v4.2, Graph Pad Prism v10.1.1, Ilastik version v1.3.3, Itol v5, Matlab version, Python v3, Pyseer v1.3.11, R v4.1.2, R caret package v6.0.94, Sambamba v1.0, SNPeffects v4.3.1. All original code is deposited at GitHub (https://github.com/BoeckLab/ASCT). |

For manuscripts utilizing custom algorithms or software that are central to the research but not yet described in published literature, software must be made available to editors and reviewers. We strongly encourage code deposition in a community repository (e.g. GitHub). See the Nature Portfolio guidelines for submitting code & software for further information.

## Data

Policy information about availability of data

All manuscripts must include a data availability statement. This statement should provide the following information, where applicable:

- Accession codes, unique identifiers, or web links for publicly available datasets
- A description of any restrictions on data availability
- For clinical datasets or third party data, please ensure that the statement adheres to our policy

All sequencing data of this study is deposited at the European Nucleotide Archive (ENA) with respective accession codes provided in source data. Raw imaging data

reported in this study cannot be deposited in a public repository due to size constraints, but is available from the lead contact upon request. All downstream imaging data and all other data are provided with this paper.

# Research involving human participants, their data, or biological material

Policy information about studies with human participants or human data. See also policy information about sex, gender (identity/presentation), and sexual orientation and race, ethnicity and racism.

| | |
|---|---|
| Reporting on sex and gender | N/A |
| Reporting on race, ethnicity, or other socially relevant groupings | N/A |
| Population characteristics | 405 patients with Mycobacterium abscessus pulmonary infection. |
| Recruitment | Retrospective clinical metatdata of patients assessed during routine clinical assessments was used. No patient was recruited for this study. |
| Ethics oversight | Ethical approval to use clinical metadata was obtained from the National Research Ethics Service (NRES; REC reference: 12/EE/0158) and the National Information Governance Board (NIGB; ECC 3-03 (f)/2012) for centres in England and Wales; from NHS Scotland Multiple Board Caldicott Guardian Approval (NHS Tayside AR/SW) for Scottish centres; and respective review boards from Queensland (Australia) and the University of North Carolina (USA). |

Note that full information on the approval of the study protocol must also be provided in the manuscript.

# Field-specific reporting

Please select the one below that is the best fit for your research. If you are not sure, read the appropriate sections before making your selection.

☒ Life sciences ☐ Behavioural & social sciences ☐ Ecological, evolutionary & environmental sciences

For a reference copy of the document with all sections, see nature.com/documents/nr-reporting-summary-flat.pdf

# Life sciences study design

All studies must disclose on these points even when the disclosure is negative.

| | |
|---|---|
| Sample size | This was a retrospective study. The sample size was based on available clinical samples. |
| Data exclusions | Replicates of M. abscessus time-kill kinetics not fulfilling the quality criteria were excluded (as outlined in the methods). |
| Replication | All experiments were assessed in replicates and all replicates analysed, except M. abscessus time-kill kinetics not fulfilling the quality criteria were excluded (as outlined in the methods). |
| Randomization | N/A. Samples were not allocated to experimental groups. |
| Blinding | N/A. Samples were not allocated to experimental groups. |

# Reporting for specific materials, systems and methods

We require information from authors about some types of materials, experimental systems and methods used in many studies. Here, indicate whether each material, system or method listed is relevant to your study. If you are not sure if a list item applies to your research, read the appropriate section before selecting a response.

## Materials & experimental systems

| n/a | Involved in the study |
|---|---|
| ☒ | ☐ Antibodies |
| ☒ | ☐ Eukaryotic cell lines |
| ☒ | ☐ Palaeontology and archaeology |
| ☒ | ☐ Animals and other organisms |
| ☐ | ☒ Clinical data |
| ☒ | ☐ Dual use research of concern |
| ☒ | ☐ Plants |

## Methods

| n/a | Involved in the study |
|---|---|
| ☒ | ☐ ChIP-seq |
| ☒ | ☐ Flow cytometry |
| ☒ | ☐ MRI-based neuroimaging |

# Clinical data

Policy information about clinical studies

All manuscripts should comply with the ICMJE guidelines for publication of clinical research and a completed CONSORT checklist must be included with all submissions.

| | |
|---|---|
| Clinical trial registration | Retrospective observational study. |
| Study protocol | This was a retrospective study done with previously obtained clinical samples and metadata. |
| Data collection | Retrospective clinical metatdata of patients assessed during routine clinical assessments was used. |
| Outcomes | Pulmonary M. abscessus infections were considered cleared, in case of sustained M. abscessus culture conversion after six months of treatment. Otherwise, the M. abscessus infection was considered persistent. |

# Plants

| | |
|---|---|
| Seed stocks | N/A |
| Novel plant genotypes | N/A |
| Authentication | N/A |

Policy information about clinical studies

