## [Peer Review File · Nature Microbiology]

Large-scale testing of antimicrobial lethality at single-cell resolution predicts mycobacterial infection outcomes

Corresponding Author: Dr Lucas Boeck

Version 0:

Reviewer comments:

Reviewer #1

(Remarks to the Author)

Overall Comment

The authors argue that growth-based methods may be limited in predicting treatment outcomes, and propose bacterial killing as a potentially more informative alternative to MIC determination for *Mycobacterium tuberculosis* and *Mycobacterium abscessus* infections. Their approach seems well-suited to studying bacterial tolerance at scale and evaluating treatment effectiveness. The amount and elegance of the work are commendable, and it addresses an important and often underexplored area in Mycobacteria research, which is gaining recognition for its relevance to treatment response. That said, it is suggested that this method can predict to individual treatment outcomes in a large extent, at least deserving clinical considerations, but I am not sure if this is fully supported by the data as presented. The authors do convincingly highlight the relevance of bacterial killing in understanding treatment responses, but the connection to individual clinical outcomes could benefit from further development or clarification. Below, I outline some major and minor points that may help strengthen the manuscript:

- 1) In the *M. tuberculosis* experiments, non-clinical strains are used to compare treatments that approximate the standard of care (SoC) versus those considered better. As such, the analysis focuses more on regimen performance than individual treatment outcomes. Additionally, the tested regimens involve combinations rather than individual clinical outcomes, and the strains are not derived from patient samples. In contrast, the *M. abscessus* experiments involve individual strains linked to treatment outcomes (e.g., treatment duration), yet the predictive utility of tolerance at the individual level remains unclear. While the results are highly relevant, the current framing of the study could be reconsidered — perhaps shifting from individual outcome prediction to a more general evaluation of regimen efficacy, especially given the differing approaches across the two models.
- 2) For example, in the *M. tuberculosis* section, the authors state: “we found that drug-specific killing dynamics rather than bacterial growth inhibition predict infection outcomes in mice and humans.” While this interpretation may be plausible, it may not be directly tested within the present dataset. Since the strains used are not linked to specific patient outcomes, it's difficult to assess how representative they are of real-world infections. The role of killing in understanding treatment combinations is well demonstrated, but the connection to actual infection outcomes might be more nuanced than the current wording suggests.
- 3) In the summary, the manuscript states: “We show that drug tolerance is conserved across similar drug targets, identify a phage protein that modulates antibiotic killing, and demonstrate that strain-specific killing dynamics shape individual patient outcomes — similar to, but independent of, drug resistance.” This raises an important conceptual point: is the phenomenon strain-specific (as suggested by the *M. abscessus* findings) or more broadly conserved (as implied by the *M. tuberculosis* data)? The latter interpretation may require more cautious language, and it might be helpful to reframe the *M. tuberculosis* data in terms of regimen efficacy rather than individualized outcome prediction. If I've misunderstood this distinction, I'm happy to be corrected.
- 4) Regarding Figure 2B — it's somewhat surprising that a clearly sterilizing regimen like RIF-PYR appears to perform relatively poorly, given its well-established efficacy in the literature. Similarly, BPaL is not generally considered superior to such regimens. In addition, why most bedaquiline-containing combinations seem to outperform others?, this is worth noting. It may be useful to address this discrepancy or clarify how this figure aligns with existing clinical data.
- 5) On the clinical trial data: it would be helpful to provide clear references to the source studies. The current citation does not seem to include human data. It appears that the analysis draws on phase IIA and IIB trials, where culture conversion — rather than relapse — is typically the primary endpoint. Since relapse might correlate more closely with bacterial killing, this could affect the interpretation. It may be helpful to clarify what is meant by “clinical outcome” in the manuscript and how closely the trial endpoints align with this definition.
- 6) One point that could benefit from clarification relates to the assumption that clinical outcomes can be predicted by in vitro metrics like MIC or bacterial killing. While bacterial factors are clearly under-investigated — and this study offers valuable insight — treatment outcomes are influenced by a wide range of factors, including host responses,

pharmacokinetics/pharmacodynamics, adverse events, loss to follow-up, and more. For instance, two regimens with similar in vitro killing profiles could differ significantly in real-world effectiveness due to non-bacterial factors. Acknowledging this complexity might help frame the results in a more balanced and clinically relevant way.

7) In the *M. abscessus* experiments, the predictive link between tolerance and clinical outcome might be strengthened by ensuring that MIC and killing data correspond only to the drugs actually used for each patient. From the text, it appears that all drugs were tested across all strains. If treatment-specific drug-strain pairs are not available, it may help to note this as a limitation. If this matching was done but not clearly described, a brief explanation could clarify.

8) Regarding the role of MIC: while the authors suggest it is a poor predictor of treatment outcomes, it is important to recognize that resistance — typically defined by an MIC above a breakpoint — remains a useful clinical tool for guiding therapy. This is especially true in cases where resistance is well characterized. While bacterial killing may indeed offer additional predictive value, it might be helpful to more explicitly acknowledge the existing utility of MIC-based breakpoints and the rationale behind combination therapies.

9) As mentioned earlier, the *M. tuberculosis* experiments seem more focused on evaluating the correlation between bacterial killing and regimen efficacy rather than predicting individual outcomes. These are not clinical strains tied to treated patients, and no direct real-world validation of individual outcomes is included.

10) The grouping of regimens into SoC and BSoC is clear, but since there is overlap in bacterial killing values between the groups, showing individual-level comparisons may offer additional insight. The pooled data suggests a strong correlation between killing and regimen classification, but given that individual values are available, it might be useful to also show paired SoC-BSoC comparisons. Are there instances where SoC regimens performed better? Or cases where the difference was not statistically significant? This could also help contextualize the statement: "Overall, 11 out of 12 associations between *M. tuberculosis* killing in starvation and clinical outcomes reached statistical significance." Including individual data or related statistical tests might strengthen this conclusion.

11) Finally, returning to the broader theme — the complexity of clinical outcomes — it may be helpful to consider whether two regimens with similar in vitro killing could still diverge in real-world performance due to non-bacterial factors (e.g., side effects, adherence, drug access). If so, interpreting regimen efficacy solely based on bacterial killing might oversimplify the clinical picture. This nuance could be acknowledged to round out the interpretation of the findings.

11) The correlation between the method used to assess bacterial killing and survival determined via a standard killing curve appears to be quite low (0.14, Fig ED_FIG4_B). This result isn't discussed in much detail in the manuscript. Could the authors comment on whether this discrepancy might influence the overall conclusions? Additionally, it's not fully clear why day 7 was chosen as the point of correlation rather than MDK99 for example. I guess it is related to starvation but clarification of this choice would help the reader better understand the rationale.

12) In validating that propidium iodide (PI) reliably stains dead cells, the authors report that only "1–11% of the PI-negative bacteria regrew in antibiotic-free media" during the first 24 hours post-treatment. It would be helpful to discuss why the remaining 89–99% of PI-negative cells did not grow. Does this imply that these cells were viable but non-culturable, or are they likely dead? This raises the question of how definitively PI staining alone can be used to indicate cell death. Additional discussion of this point would clarify how robust PI is as a proxy for viability in this context.

13) The *M. abscessus* work is a strong part of the manuscript and comes closer to linking bacterial factors with individual clinical outcomes, using patient isolates. However, I have some concerns about the interpretation of susceptibility to cefoxitin and imipenem. Drug susceptibility testing (DST) for these agents in *M. abscessus* is known to be unreliable in many settings. Could this affect some of the downstream analyses involving these drugs? In fact, in Fig 3B, they show remarkably high activity compared to others — it might be useful to comment on this result in the context of DST variability.

14) Regarding the use of in vitro bacterial killing to predict patient outcomes in *M. abscessus* cases: how well do the antibiotics tested in vitro match the treatments given to each patient? Are the authors testing individual drugs used in each patient's regimen, or using combinations similar to the *M. tuberculosis* part of the study? Clarifying whether killing data reflect the specific drugs administered to each patient would strengthen the link between the experimental data and clinical outcomes.

15) The statement that "drug tolerance is heritable" is an interesting and valuable contribution, as it supports the idea that drug tolerance has genetic determinants that can be studied. That said, some of the heritability values presented in Fig 4 raised questions. For example, if I'm interpreting the figure correctly, is the heritability of MIC for some antibiotics essentially zero? It might be helpful to briefly comment on this, especially in contrast to the heritability estimates for tolerance.

16) In reference to the conclusion that "drug tolerance predicts individual infection outcomes" (Fig 5 and related text): while the data do suggest an association, it's not entirely clear that individual outcome prediction was directly tested. For instance, in Fig 5a, it's hard to determine whether the data are presented at the individual level. A statement like "contributes to predicting" or "is associated with improved prediction of" might be more accurate unless further evidence is provided. Additionally, the ROC value for imipenem (0.68) suggests only moderate predictive performance. It might strengthen the analysis to include other performance metrics, such as positive and negative predictive values (PPV and NPV), to help the reader better understand the extent to which tolerance adds to clinical prediction

(Remarks on code availability)

(Remarks to the Author)

In this study, Jovanovic et al. present a high-throughput method, ASCT, for measuring antibiotic killing in mycobacteria at single-cell level, and apply it in two clinically relevant areas. First, they probe 65 combinatorial antibiotic regimens used to treat *M. tuberculosis* (Mtb) and show that in-vitro killing of starved/non-growing Mtb cells, as measured by ASCT, captures better pathogen clearance in animal models and clinical studies, compared to traditional MIC measures or killing measured in growing cells or by conventional CFU/ml measures. Second, they move to *M. abscessus* and measure killing kinetics of 8 antibiotics in a collection of 405 exponentially growing clinical isolates using ASCT. They use these data to compare how well they predict clinical outcomes and how they are linked to growth characteristics of the strains and to the drug mode-of-action. They then genotype all strains to identify genetic elements that drive distinct tolerance elements, and follow up on a specific phage-tail protein that (partially) drives tolerance to protein-synthesis inhibitors. Overall, this is an impressive method with potential to change both our fundamental understanding of tolerance mechanisms and clinical practice guidelines – at least for Mycobacterial infections. It is accompanied by 2 tour-de-force applications of the method to benchmark the quality of the data it produces and to exemplify its power. My complements to the authors – all my points below are geared towards clarifying some of the aspects of the study design, as well the results and their interpretation.

Main points

1. One of the main points of the study (in abstract – lines 7-9, but also in intro, main text and discussion) is that killing dynamics predict better infection outcomes when compared to growth inhibition. However authors show that killing dynamics of exponentially growing cells predict equally “bad” Mtb infection outcomes as growth inhibition. So rather than killing vs growth inhibition, it’s the state of the cells that recapitulates better the infection state and hence antibiotic efficacy during infection. I find this is an important distinction that needs to be put more forward. This is particularly important since for other pathogens dormant/not growing states may be less central to their treatment as compared to Mtb –(see also point 4 about this).
2. The new method gives an impressive alternative in terms to throughput and information to cumbersome CFU measurements for assessment of killing kinetics, allowing for use of single-cell data. Nevertheless, the manuscript would benefit from expand the discussion on some of the caveats and areas of improvement/expansion of the new method:
 - (i) other high-throughput assays have been used to assess killing – e.g. resazurin in bulk assays, or cytological profiling in image-based assays. All these assays may have caveats compared to current method, but should be mentioned/discussed here.
 - (ii) authors apply the new method exclusively to mycobacteria, but do note that it is agnostic to species tested. This is an interesting point that deserves more discussion – e.g. how much adjustment is needed for the automated cell tracking and PI ability to capture dead cells? does faster growth limit the time-scale that killing kinetics can be followed (suppressors being a bigger problem)? would other species be measured with similar killing kinetics?
 - (iii) authors acknowledge caveats of the PI stain, but they seem to underplay the fact that only 1-11% of PI-negative cells regrow in antibiotic-free media both in results and discussion. Are there caveats of how they measure regrowth? Are there alternatives/orthogonal measures to PI? These are points that all need to be discussed more extensively.
 - (iv) comparison to CFU measures (ED4) is unfair, as CFUs are measured in 2 time-points and compared to continuous and more robust measurements (AUCs) using PI. To do this in a fairer way, the authors should perform a sensitivity analysis with the PI data, taking single-time points from the ASCT curve and testing their predictive power for in vivo outcomes of antibiotic regimens – as they do for CFUs in ED Fig 4c.
3. MIC-measures (mean, median, lowest) are not good predictors of in vivo outcomes of antibiotic regimens in Mtb (ED Fig. 5). However since this is to predict the efficacy of combinatorial treatments, the authors may want to test how interaction metrics (FICI, Bliss scores) perform.
4. The motivation to move from Mtb to *M. abscessus* experiments is well-explained – latter allows for use of large clinical isolate collections (since biosafety level 2), and thereby opens the path for genotype associations to tolerance. However, the choices of the drugs and the assays are less well motivated. Is there are reason that these 8 antibiotics were tested – rather than newer Mtb-specific drugs with different mode-of-actions and more activity to dormant cells? Even more importantly, is there a reason that tolerance is probed in exponentially growing *M. abscessus*, when the authors have just shown that for Mtb starved cells have much better prediction power of clinical outcomes? At the very least this latter part should be better acknowledged and discussed in manuscript.
5. The authors make a point that correlations between *M. abscessus* kill curves of exponential cells (as measured by ASCT) and growth dynamics (Fig. 3c,d; ED Fig. 6) are largely absent. This goes against current knowledge, and I would be careful for strong statements here, as they may be confounded by technical reasons.

First, authors surprisingly choose to measure growth by OD in batch culture – rather than using their ability to track single cells, so they can measure single cell growth. This would have had several advantages: same set-up; additional single-cell readouts to growth (cell morphology metrics) and ability to even compare growth and tolerance of same cell if tracking was done continuously (so correlations can be built by comparing thousands of cells directly rather than comparing bulk measurements to AUCs of pooled-single cell data).

Second, the measures they extract from growth curve are somewhat questionable. What they measure as lag phase is “apparent lag phase”, a composite of true lag phase and growth rate below the level of detection (if they start from 10^5 cells they have several doublings below the level of detection). Hence no surprise that 2 metrics (growth rate and lag phase) strongly correlate. They can use the growth rate and initial inoculum to deduce the true lag phase. Moreover, it’s unclear how good are the Gompertz fits of their curves, how long do the different strains grow with a constant growth rate, and at which point of this growth curve have the isolates been treated with the antibiotic.

6. Genetic nature of tolerance and specificity to drug class. Both of these aspects are exciting, but in both cases, as the authors acknowledge, there have been prior reports. What is even more exciting are the GWAS associations for the drug tolerance

phenotypes, which yield strong association to 43 genes. The authors should show all genes in Fig 6b, rather than just the one they follow-up, and provide their functional groupings (to back-up the claim that they are enriched in functional groups related to antibiotic target – lines 192-193).

In terms of follow up, I understand that identifying the mechanism of the phage tail protein in protein synthesis killing may go beyond the scope of this study, but some discussion on this topic would be helpful.

Minor points

1. lines 5--7: it's important to specify that bacteria = mycobacteria.
2. lines 11-12: "drug tolerance is conserved across similar drug targets" – consider revising as it's unclear what you mean. Tolerance patterns, mechanisms or kinetics are similar for drug with same target?
3. lines 25-26: some mechanisms of drug tolerance are known - both in vitro and in infections. One could cite dozens of papers.
4. line 53: isn't it better to say that this is tested in 1536-plates? More than 1,000 conditions sounds quite random.
6. Fig 1c – this cannot be the time-kill curve of Tigecycline, as it directly contradicts traces shown in Fig. 1d. Please also specify which clinical isolate of *M. abscessus* is profiled.
7. lines 89-91: why are those three drugs expected to act better on starved cells? It's not straightforward to explain based on their target/mode-of-action.
8. lines 127-129 – it may be good to mention how much antibiotic concentration (and it's relation to MIC) is expected to affect killing kinetics.
9. lines 152-153 – many more papers beyond mycobacteria, and I would find important to cite PMID 25043002 and 28183996, which have been seminal in this regard.
10. lines 158-159: why resistance of unstable drugs (unstable in what sense? chemically unstable?) or drugs with less variable MIC are expected to be less heritable?
11. lines 168-170 (and lines 242-244): increased resistance in some antibiotics resulting as a trade-off to decreased tolerance to other antibiotics, is reminiscent of collateral sensitivity and opens the path for combinatorial use. Maybe the link can be made in discussion.
12. Fig 5a: please provide n for box plots. If same as in ED. Fig 8 (46 + 89), then please correct total in the legend (135, not 136).
13. lines 180-183 and Fig 5B: it's unclear why data from tolerance on imipenem is combined with resistance to macrolides to benchmark predictions. Why this combination of tolerance and resistance vs any other?
14. line 198: most of the protein synthesis inhibitors tested here are not known to cause mistranslation, and only inhibit translation.
15. line 201; this is not entirely true for linezolid (complemented mutant has also a significantly different time-kill curve compared to wildtype too).
16. lines 231-233: this has been challenged before. See for example PMID: 34646011 - there is shown that antibiotics can change from bacteriostatic action to rapid killing depending on the species and stain tested.

(Remarks on code availability)

Reviewer #3

(Remarks to the Author)

In the present report, Jovanovic and collaborators report an exciting and novel antimicrobial single cell testing to overcome historical limitations of measuring susceptibility in bulk bacterial populations. This is an impressive tour de force new method that have the potential to significantly improve our understanding of how antibiotics work and clinical decision-making. The quality of the data and the analysis is very high.

There are, however, some aspects that need to be considered to strength the message of the manuscript (and the approach). I have some suggestions for improving the manuscript:

The cell viability is measured by propidium iodide accumulation after penetrating the damaged bacterial cell wall. Imipenem and cefoxitin, which target the cell wall stands out as the more effective drugs against both *M. tuberculosis* (Figure 1d) and *M. abscessus* (Figure 3b). Could antibiotics targeting cell wall facilitate PI penetration, possibly leading to overestimation of death rates in those conditions? On the other hand, is possible an underreporting of cell death by non-membrane-disrupting drugs? (increasing the chance of false negative)

The experiments presented in Figure 1D do need an untreated control as it is important to know how the system affects cell viability in general. There are no SEM values for Linezolid, Imipenem and Cefoxitin. Would it be because there is no variability? I understand the authors do not want to assess single-cell heterogeneity in this manuscript as mentioned in the main text. However, it would be interesting to assess the variability between antibiotic at the bulk level as well as the temporal level. For

example, if we consider Amikacin or Moxifloxacin, it seems that the variability (here shown by the SEM) is higher in later timepoints. Could it suggest that within experiments, early-death (24h) mediated by ATBs is less variable than late-death? Could the authors comment on this.

Bactericidal and bacteriostatic effects can switch depending on antibiotic concentration. Hence, I suggest reinforcing the Figure 1 to add for one or two antibiotic a growth assessment using different concentrations.

The authors state that INH, RIF and EMB are more active against exponentially growing *M. tuberculosis*. I would change the wording, as we know that at the single-cell level *Mtb* seems to have a linear growth (Chung et al. 2024), even though I understand that the authors refer to bulk growth.

In Figure 2, the manuscript would benefit if the authors can show all the combinations possible of the 6 antibiotics in both rich and starvation media. Moreover, how the authors can exclude that in the comparison between rich and starvation media, the drugs are binding to media components and do not enter the bacterial cell?

Regarding the Figure 2 and the link between the ASCT in vitro killing efficacy under bacterial starvation (PBS and pantothenate) and the predicted in vivo outcome. Dormant/metabolically inactive phenotypes generated in vitro could mirror drug-tolerant bacilli in vivo. However, in vivo lesions have multiple microenvironments (oxygen tension, pH, lipids) that cannot be addressed by the in vitro ASCT starvation assay. The in vitro assay uses constant concentrations of antibiotics, which don't reflect dynamic drug exposure that happens in vivo. These limitations (as mentioned in the discussion "Furthermore, our study did not specifically mimic complex mycobacterial infection environments, such as intracellular, niches or granulomas, which may alter tolerance phenotypes") may affect the comparison between the in vitro results in this work and the in vivo (mouse and clinical trials) results from references 28 and 29.

Moreover, a limitation of the correlation between the in vitro data presented and the results from mice/clinical trials relies on the strain of *Mtb* used. I acknowledge the "tour-de-force" of the method, however, as mentioned by others, antibiotic efficacy can be affected by *Mtb* localization in cells (Santucci et al, 2021), therefore potentially in vivo. This is not discussed in the results section or discussion. The authors should discuss this and acknowledge the limitations of this correlation.

In the main text, L119-120, "which probably shares fundamental principles of antibiotic survival with other mycobacteria", it is a bold sentence, not referring to any study. Also, *Mtb* and *Mab* share their genus, but differs through their growth rate. I would tone down this sentence as it can be misleading.

One criticism that could be made about the "genetic" bacterial background is that we do not know, within those clinical isolates, if patients have been differentially treated with different antibiotics that could have generate tolerance, unrelated to the genetic background. Could the authors comment on the distribution of the strains?

In Figure 6, The data regarding MAB_0233 are quite interesting, did the authors looked for similar genes in other mycobacteria (maybe some of them that are more tolerant to those drugs, in clinics)?

In order to validate the method, the use of other markers of cell death or metabolic activity (for example resazurin) could be implemented into the Antimicrobial Single-Cell Testing (ASCT) to compare results.

(Remarks on code availability)

It works

Reviewer #4

(Remarks to the Author)

This manuscript addresses a critical issue in the treatment of Mycobacterium infections: the disconnect between in vitro assessments of antibiotic susceptibility and actual in vivo outcomes. The authors develop a microscopy-based method, termed Antimicrobial Single-Cell Testing (ASCT), to measure antibiotic killing dynamics in real time at single-cell resolution, and correlate these measurements with infection outcomes in vivo. They further perform genetic analysis to identify determinants of drug tolerance.

Overall, I find this manuscript to be both timely and of significant potential interest. The authors thoroughly characterize in vitro efficacy and, through statistical analysis, relate it to clinical outcomes. A wide range of antibiotics and mutants are examined in order to identify generalizable features that explain treatment outcomes. This integrative approach has value for the evidence-based design of diagnostics and treatments.

That said, I found some parts of the manuscript difficult to follow. In several instances, explanations are either insufficient or potentially misleading due to a lack of clarity or contextualization. Moreover, the manuscript does not clearly distinguish which findings are novel contributions of this study versus reiterations or confirmations of previously established knowledge.

Taking an abstract as an example, the description of ASCT as a "large-scale imaging approach that quantifies bacterial killing in real time at single-cell resolution". Many recent studies have applied single-cell imaging approaches to monitor antibiotic killing. It would be helpful if the authors clarified what specifically differentiates ASCT from these existing methods—whether it's the throughput, resolution, experimental design, or analytical framework.

Likewise, the statement that "drug-specific killing dynamics – rather than bacterial growth inhibition – predict infection outcomes" is ambiguous. Does this imply that bactericidal antibiotics are more predictive than bacteriostatic ones? Or by bacterial growth inhibition, do authors allude to slow death (tolerance)?

The authors also state that "antibiotic killing is not merely a drug effect, but a genetically encoded bacterial trait (drug tolerance) distinct from classical drug resistance." Does this statement try to mean that there are genetic mechanisms for drug tolerance? Indeed, the existence of a genetic base of tolerance is a well-accepted idea in microbiology. I saw the author cited "43. Levin, B. R. & Rozen, D. E. Non-inherited antibiotic resistance. *Nat. Rev. Microbiol.* 4, 556– 367 562 (2006)." to support that drug tolerance is non-genetic. But, this paper is more than 20 years old. There have been significant advances since then.

I agree with the central premise that bacterial tolerance may play an underappreciated role in mycobacterial infections and that MIC values may not correlate well with clinical success. However, I caution against generalizing this finding to all bacterial infections. There is ample evidence in other systems where high MIC values are associated with treatment failure. The manuscript should be more precise in delimiting the scope of its claims.

The treatment of heritability also warrants more explanation. While a technical paper is cited (ref 48), the meaning of "heritability" in this context remains unclear to me. If the authors are arguing for genetic inheritance of drug tolerance, does it warrant addressing possible confounding factors such as horizontal gene transfer, repeated spontaneous emergence, or clonal spread of strains within communities?

Some figures are hard to understand and not clear. I think authors should strengthen figure captions. These are just a few that stood out the most.

- Fig 1d. What is the sample size of each rep? Similar to Fig. 1c? I vaguely remember in suppl materials that wells with <500 or <1000 cells discarded so I'm assuming sample size is quite large.
- Fig2b. It is difficult to evaluate the graph without standard deviation. The vertical lines are distracting. I am not sure what information the black and white dots show other than drug combinations.
- Fig 3a. Does color mean anything?
- Fig 3e. R² value is very low and slopes are small shown in Ext Fig 6. Are these statistically meaningful?
- Fig 4a. Standard error should be added. Growing isolates should be added since they are the true resistant strains. I'm assuming "resistance" and "tolerance" are selected based on MIC and AUC? It should be clarified in legend.
- Fig 5a. Is there a reason that individual data (like in Fig. 2cd, which were very helpful) is missing here?

(Remarks on code availability)

Decision Letter:

29th April 2025

Dear Lucas,

Thank you for your patience while your manuscript "Antibiotic lethality dictates mycobacterial infection outcomes" was under peer-review at Nature Microbiology. It has now been seen by 4 referees, whose expertise and comments you will find at the end of this email. Although they find your work of some potential interest, they have raised a number of concerns that will need to be addressed before we can consider publication of the work in Nature Microbiology.

In particular, referee #1 says that the clinical relevance of the work should be further developed or clarified (for example via additional discussion), and feels that the current framing of the study could be reconsidered towards a more general evaluation of regimen efficacy. The referee also says that the predictive link between tolerance and clinical outcome might be strengthened by ensuring that MIC and killing data correspond only to the drugs actually used for each patient. Referee #2 says it should be made clearer that rather than killing vs growth inhibition, it is the state of the cells that recapitulates better the infection state and hence antibiotic efficacy during infection. This referee also suggests expanding the discussion on some of the caveats and areas of improvement/expansion of the new method. Furthermore, referee #2 suggests to test how interaction metrics (FIC_i, Bliss scores) perform. Referee #3 says that the experiments presented in Figure 1D need an untreated control. The referee also feels it would be interesting to assess the variability between antibiotic at the bulk level as well as the temporal level. Further the referee suggests reinforcing Figure 1 to add for one or two antibiotic a growth assessment using different concentrations. For Figure 2, the referee states that the manuscript would benefit from showing all the combinations possible of the 6 antibiotics in both rich and starvation media. Similar to referee #2, referee #3 also asks to better describe the limitations of the study. Referee #4 stresses that some parts of the manuscript and some of the figures should be simplified for non-experts.

Should further experimental data allow you to address these criticisms, we would be happy to look at a revised manuscript.

Please include a data availability statement as a separate section after Methods but before references, under the heading "Data Availability". This section should inform readers about the availability of the data used to support the conclusions of your study. This information includes accession codes to public repositories (data banks for protein, DNA or RNA sequences, microarray, proteomics data etc...), references to source data published alongside the paper, unique identifiers such as URLs to data repository entries, or data set DOIs, and any other statement about data availability. At a minimum, you should include the following statement: "The data that support the findings of this study are available from the corresponding author upon request",

mentioning any restrictions on availability. If DOIs are provided, we also strongly encourage including these in the Reference list (authors, title, publisher (repository name), identifier, year). For more guidance on how to write this section please see: <http://www.nature.com/authors/policies/data/data-availability-statements-data-citations.pdf>

* If you have not done so already we suggest that you begin to revise your manuscript so that it conforms to our Article format instructions at <http://www.nature.com/nmicrobiol/info/final-submission>. Refer also to any guidelines provided in this letter.

When submitting the revised version of your manuscript, please pay close attention to our [href="https://www.nature.com/nature-portfolio/editorial-policies/image-integrity">Digital Image Integrity Guidelines.](https://www.nature.com/nature-portfolio/editorial-policies/image-integrity) and to the following points below:

EXTENDED DATA FIGURES

Link Redacted

Note: This url links to your confidential homepage and associated information about manuscripts you may have submitted or be reviewing for us. If you wish to forward this e-mail to co-authors, please delete this link to your homepage first.

Nature Microbiology is committed to improving transparency in authorship. As part of our efforts in this direction, we are now requesting that all authors identified as 'corresponding author' on published papers create and link their Open Researcher and Contributor Identifier (ORCID) with their account on the Manuscript Tracking System (MTS), prior to acceptance. This applies to primary research papers only. ORCID helps the scientific community achieve unambiguous attribution of all scholarly contributions. You can create and link your ORCID from the home page of the MTS by clicking on 'Modify my Springer Nature account'. For more information please visit www.springernature.com/orcid.

If you wish to submit a suitably revised manuscript we would hope to receive it within 4 months. If you cannot send it within this time, please let us know.

Yours sincerely,

Reviewer Expertise:

- Referee #1: Computational biology, Mycobacteria
- Referee #2: Antimicrobial resistance and tolerance, systems biology
- Referee #3: Mycobacteria
- Referee #4: Single cell analysis, computational modeling

Reviewer Comments:

Reviewer #1 (Remarks to the Author):

Overall Comment

The authors argue that growth-based methods may be limited in predicting treatment outcomes, and propose bacterial killing as a potentially more informative alternative to MIC determination for *Mycobacterium tuberculosis* and *Mycobacterium abscessus* infections. Their approach seems well-suited to studying bacterial tolerance at scale and evaluating treatment effectiveness. The amount and elegance of the work are commendable, and it addresses an important and often underexplored area in *Mycobacteria* research, which is gaining recognition for its relevance to treatment response. That said, it is suggested that this method can predict to individual treatment outcomes in a large extent, at least deserving clinical considerations, but I am not sure if this is fully supported by the data as presented. The authors do convincingly highlight the relevance of bacterial killing in understanding treatment responses, but the connection to individual clinical outcomes could benefit from further development or clarification. Below, I outline some major and minor points that may help strengthen the manuscript:

- 1) In the *M. tuberculosis* experiments, non-clinical strains are used to compare treatments that approximate the standard of care (SoC) versus those considered better. As such, the analysis focuses more on regimen performance than individual treatment outcomes. Additionally, the tested regimens involve combinations rather than individual clinical outcomes, and the strains are not derived from patient samples. In contrast, the *M. abscessus* experiments involve individual strains linked to treatment outcomes (e.g., treatment duration), yet the predictive utility of tolerance at the individual level remains unclear. While the results are highly relevant, the current framing of the study could be reconsidered — perhaps shifting from individual outcome prediction to a more general evaluation of regimen efficacy, especially given the differing approaches across the two models.
- 2) For example, in the *M. tuberculosis* section, the authors state: “we found that drug-specific killing dynamics rather than bacterial growth inhibition predict infection outcomes in mice and humans.” While this interpretation may be plausible, it may not be directly tested within the present dataset. Since the strains used are not linked to specific patient outcomes, it’s difficult to assess how representative they are of real-world infections. The role of killing in understanding treatment combinations is well demonstrated, but the connection to actual infection outcomes might be more nuanced than the current wording suggests.
- 3) In the summary, the manuscript states: “We show that drug tolerance is conserved across similar drug targets, identify a phage protein that modulates antibiotic killing, and demonstrate that strain-specific killing dynamics shape individual patient outcomes — similar to, but independent of, drug resistance.” This raises an important conceptual point: is the phenomenon strain-specific (as suggested by the *M. abscessus* findings) or more broadly conserved (as implied by the *M. tuberculosis* data)? The latter interpretation may require more cautious language, and it might be helpful to reframe the *M. tuberculosis* data in terms of regimen efficacy rather than individualized outcome prediction. If I’ve misunderstood this distinction, I’m happy to be corrected.
- 4) Regarding Figure 2B — it’s somewhat surprising that a clearly sterilizing regimen like RIF-PYR appears to perform relatively poorly, given its well-established efficacy in the literature. Similarly, BPaL is not generally considered superior to such regimens. In addition, why most bedaquiline-containing combinations seem to outperform others?, this is worth noting. It may be useful to address this discrepancy or clarify how this figure aligns with existing clinical data.
- 5) On the clinical trial data: it would be helpful to provide clear references to the source studies. The current citation does not seem to include human data. It appears that the analysis draws on phase IIA and IIB trials, where culture conversion — rather than relapse — is typically the primary endpoint. Since relapse might correlate more closely with bacterial killing, this could affect the interpretation. It may be helpful to clarify what is meant by “clinical outcome” in the manuscript and how closely the trial endpoints align with this definition.
- 6) One point that could benefit from clarification relates to the assumption that clinical outcomes can be predicted by in vitro metrics like MIC or bacterial killing. While bacterial factors are clearly under-investigated — and this study offers valuable insight — treatment outcomes are influenced by a wide range of factors, including host responses, pharmacokinetics/pharmacodynamics, adverse events, loss to follow-up, and more. For instance, two regimens with similar in vitro killing profiles could differ significantly in real-world effectiveness due to non-bacterial factors. Acknowledging this complexity might help frame the results in a more balanced and clinically relevant way.
- 7) In the *M. abscessus* experiments, the predictive link between tolerance and clinical outcome might be strengthened by ensuring that MIC and killing data correspond only to the drugs actually used for each patient. From the text, it appears that all drugs were tested across all strains. If treatment-specific drug-strain pairs are not available, it may help to note this as a limitation. If this matching was done but not clearly described, a brief explanation could clarify.
- 8) Regarding the role of MIC: while the authors suggest it is a poor predictor of treatment outcomes, it is important to recognize that resistance — typically defined by an MIC above a breakpoint — remains a useful clinical tool for guiding therapy. This is especially true in cases where resistance is well characterized. While bacterial killing may indeed offer additional predictive value, it might be helpful to more explicitly acknowledge the existing utility of MIC-based breakpoints and the rationale behind combination therapies.
- 9) As mentioned earlier, the *M. tuberculosis* experiments seem more focused on evaluating the correlation between bacterial killing and regimen efficacy rather than predicting individual outcomes. These are not clinical strains tied to treated patients, and no direct real-world validation of individual outcomes is included.
- 10) The grouping of regimens into SoC and BSoC is clear, but since there is overlap in bacterial killing values between the groups, showing individual-level comparisons may offer additional insight. The pooled data suggests a strong correlation between killing and regimen classification, but given that individual values are available, it might be useful to also show paired SoC-BSoC comparisons. Are there instances where SoC regimens performed better? Or cases where the difference was not statistically significant? This could also help contextualize the statement: “Overall, 11 out of 12 associations between *M.*

tuberculosis killing in starvation and clinical outcomes reached statistical significance.” Including individual data or related statistical tests might strengthen this conclusion.

11) Finally, returning to the broader theme — the complexity of clinical outcomes — it may be helpful to consider whether two regimens with similar in vitro killing could still diverge in real-world performance due to non-bacterial factors (e.g., side effects, adherence, drug access). If so, interpreting regimen efficacy solely based on bacterial killing might oversimplify the clinical picture. This nuance could be acknowledged to round out the interpretation of the findings.

11) The correlation between the method used to assess bacterial killing and survival determined via a standard killing curve appears to be quite low (0.14, Fig ED_FIG4_B). This result isn't discussed in much detail in the manuscript. Could the authors comment on whether this discrepancy might influence the overall conclusions? Additionally, it's not fully clear why day 7 was chosen as the point of correlation rather than MDK99 for example. I guess it is related to starvation but clarification of this choice would help the reader better understand the rationale.

12) In validating that propidium iodide (PI) reliably stains dead cells, the authors report that only “1–11% of the PI-negative bacteria regrew in antibiotic-free media” during the first 24 hours post-treatment. It would be helpful to discuss why the remaining 89–99% of PI-negative cells did not grow. Does this imply that these cells were viable but non-culturable, or are they likely dead? This raises the question of how definitively PI staining alone can be used to indicate cell death. Additional discussion of this point would clarify how robust PI is as a proxy for viability in this context.

13) The *M. abscessus* work is a strong part of the manuscript and comes closer to linking bacterial factors with individual clinical outcomes, using patient isolates. However, I have some concerns about the interpretation of susceptibility to cefoxitin and imipenem. Drug susceptibility testing (DST) for these agents in *M. abscessus* is known to be unreliable in many settings. Could this affect some of the downstream analyses involving these drugs? In fact, in Fig 3B, they show remarkably high activity compared to others — it might be useful to comment on this result in the context of DST variability.

14) Regarding the use of in vitro bacterial killing to predict patient outcomes in *M. abscessus* cases: how well do the antibiotics tested in vitro match the treatments given to each patient? Are the authors testing individual drugs used in each patient's regimen, or using combinations similar to the *M. tuberculosis* part of the study? Clarifying whether killing data reflect the specific drugs administered to each patient would strengthen the link between the experimental data and clinical outcomes.

15) The statement that “drug tolerance is heritable” is an interesting and valuable contribution, as it supports the idea that drug tolerance has genetic determinants that can be studied. That said, some of the heritability values presented in Fig 4 raised questions. For example, if I'm interpreting the figure correctly, is the heritability of MIC for some antibiotics essentially zero? It might be helpful to briefly comment on this, especially in contrast to the heritability estimates for tolerance.

16) In reference to the conclusion that “drug tolerance predicts individual infection outcomes” (Fig 5 and related text): while the data do suggest an association, it's not entirely clear that individual outcome prediction was directly tested. For instance, in Fig 5a, it's hard to determine whether the data are presented at the individual level. A statement like “contributes to predicting” or “is associated with improved prediction of” might be more accurate unless further evidence is provided. Additionally, the ROC value for imipenem (0.68) suggests only moderate predictive performance. It might strengthen the analysis to include other performance metrics, such as positive and negative predictive values (PPV and NPV), to help the reader better understand the extent to which tolerance adds to clinical prediction

Reviewer #2 (Remarks to the Author):

In this study, Jovanovic et al. present a high-throughput method, ASCT, for measuring antibiotic killing in mycobacteria at single-cell level, and apply it in two clinically relevant areas. First, they probe 65 combinatorial antibiotic regimens used to treat *M. tuberculosis* (Mtb) and show that in-vitro killing of starved/non-growing Mtb cells, as measured by ASCT, captures better pathogen clearance in animal models and clinical studies, compared to traditional MIC measures or killing measured in growing cells or by conventional CFU/ml measures. Second, they move to *M. abscessus* and measure killing kinetics of 8 antibiotics in a collection of 405 exponentially growing clinical isolates using ASCT. They use these data to compare how well they predict clinical outcomes and how they are linked to growth characteristics of the strains and to the drug mode-of-action. They then genotype all strains to identify genetic elements that drive distinct tolerance elements, and follow up on a specific phage-tail protein that (partially) drives tolerance to protein-synthesis inhibitors. Overall, this is an impressive method with potential to change both our fundamental understanding of tolerance mechanisms and clinical practice guidelines – at least for Mycobacterial infections. It is accompanied by 2 tour-de-force applications of the method to benchmark the quality of the data it produces and to exemplify its power. My compliments to the authors – all my points below are geared towards clarifying some of the aspects of the study design, as well the results and their interpretation.

Main points

1. One of the main points of the study (in abstract – lines 7-9, but also in intro, main text and discussion) is that killing dynamics predict better infection outcomes when compared to growth inhibition. However authors show that killing dynamics of exponentially growing cells predict equally “bad” Mtb infection outcomes as growth inhibition. So rather than killing vs growth inhibition, it's the state of the cells that recapitulates better the infection state and hence antibiotic efficacy during infection. I find this is an important distinction that needs to be put more forward. This is particularly important since for other pathogens dormant/not growing states may be less central to their treatment as compared to Mtb –(see also point 4 about this).

2. The new method gives an impressive alternative in terms to throughput and information to cumbersome CFU measurements for assessment of killing kinetics, allowing for use of single-cell data. Nevertheless, the manuscript would benefit from expand the discussion on some of the caveats and areas of improvement/expansion of the new method:

(i) other high-throughput assays have been used to assess killing – e.g. resazurin in bulk assays, or cytological profiling in

image-based assays. All these assays may have caveats compared to current method, but should be mentioned/discussed here. (ii) authors apply the new method exclusively to mycobacteria, but do note that it is agnostic to species tested. This is an interesting point that deserves more discussion – e.g. how much adjustment is needed for the automated cell tracking and PI ability to capture dead cells? does faster growth limit the time-scale that killing kinetics can be followed (suppressors being a bigger problem)? would other species be measured with similar killing kinetics?

(iii) authors acknowledge caveats of the PI stain, but they seem to underplay the fact that only 1-11% of PI-negative cells regrow in antibiotic-free media both in results and discussion. Are there caveats of how they measure regrowth? Are there alternatives/orthogonal measures to PI? These are points that all need to be discussed more extensively.

(iv) comparison to CFU measures (ED4) is unfair, as CFUs are measured in 2 time-points and compared to continuous and more robust measurements (AUCs) using PI. To do this in a fairer way, the authors should perform a sensitivity analysis with the PI data, taking single-time points from the ASCT curve and testing their predictive power for in vivo outcomes of antibiotic regimens – as they do for CFUs in ED Fig 4c.

3. MIC-measures (mean, median, lowest) are not good predictors of in vivo outcomes of antibiotic regimens in Mtb (ED Fig. 5). However since this is to predict the efficacy of combinatorial treatments, the authors may want to test how interaction metrics (FICI, Bliss scores) perform.

4. The motivation to move from Mtb to M. abscessus experiments is well-explained – latter allows for use of large clinical isolate collections (since biosafety level 2), and thereby opens the path for genotype associations to tolerance. However, the choices of the drugs and the assays are less well motivated. Is there a reason that these 8 antibiotics were tested – rather than newer Mtb-specific drugs with different mode-of-actions and more activity to dormant cells? Even more importantly, is there a reason that tolerance is probed in exponentially growing M. abscessus, when the authors have just shown that for Mtb starved cells have much better prediction power of clinical outcomes? At the very least this latter part should be better acknowledged and discussed in manuscript.

5. The authors make a point that correlations between M. abscessus kill curves of exponential cells (as measured by ASCT) and growth dynamics (Fig. 3c,d; ED Fig. 6) are largely absent. This goes against current knowledge, and I would be careful for strong statements here, as they may be confounded by technical reasons.

First, authors surprisingly choose to measure growth by OD in batch culture – rather than using their ability to track single cells, so they can measure single cell growth. This would have had several advantages: same set-up; additional single-cell readouts to growth (cell morphology metrics) and ability to even compare growth and tolerance of same cell if tracking was done continuously (so correlations can be built by comparing thousands of cells directly rather than comparing bulk measurements to AUCs of pooled-single cell data).

Second, the measures they extract from growth curve are somewhat questionable. What they measure as lag phase is “apparent lag phase”, a composite of true lag phase and growth rate below the level of detection (if they start from 10^5 cells they have several doublings below the level of detection). Hence no surprise that 2 metrics (growth rate and lag phase) strongly correlate. They can use the growth rate and initial inoculum to deduce the true lag phase. Moreover, it's unclear how good are the Gompertz fits of their curves, how long do the different strains grow with a constant growth rate, and at which point of this growth curve have the isolates been treated with the antibiotic.

6. Genetic nature of tolerance and specificity to drug class. Both of these aspects are exciting, but in both cases, as the authors acknowledge, there have been prior reports. What is even more exciting are the GWAS associations for the drug tolerance phenotypes, which yield strong association to 43 genes. The authors should show all genes in Fig 6b, rather than just the one they follow-up, and provide their functional groupings (to back-up the claim that they are enriched in functional groups related to antibiotic target – lines 192-193).

In terms of follow up, I understand that identifying the mechanism of the phage tail protein in protein synthesis killing may go beyond the scope of this study, but some discussion on this topic would be helpful.

Minor points

1. lines 5–7: it's important to specify that bacteria = mycobacteria.

2. lines 11-12: “drug tolerance is conserved across similar drug targets” – consider revising as it's unclear what you mean. Tolerance patterns, mechanisms or kinetics are similar for drug with same target?

3. lines 25-26: some mechanisms of drug tolerance are known - both in vitro and in infections. One could cite dozens of papers.

4. line 53: isn't it better to say that this is tested in 1536-plates? More than 1,000 conditions sounds quite random.

6. Fig 1c – this cannot be the time-kill curve of Tigecycline, as it directly contradicts traces shown in Fig. 1d. Please also specify which clinical isolate of M. abscessus is profiled.

7. lines 89-91: why are those three drugs expected to act better on starved cells? It's not straightforward to explain based on their target/mode-of-action.

8. lines 127-129 – it may be good to mention how much antibiotic concentration (and it's relation to MIC) is expected to affect killing kinetics.

9. lines 152-153 – many more papers beyond mycobacteria, and I would find important to cite PMID 25043002 and 28183996, which have been seminal in this regard.

10. lines 158-159: why resistance of unstable drugs (unstable in what sense? chemically unstable?) or drugs with less variable MIC are expected to be less heritable?
11. lines 168-170 (and lines 242-244): increased resistance in some antibiotics resulting as a trade-off to decreased tolerance to other antibiotics, is reminiscent of collateral sensitivity and opens the path for combinatorial use. Maybe the link can be made in discussion.
12. Fig 5a: please provide n for box plots. If same as in ED. Fig 8 (46 + 89), then please correct total in the legend (135, not 136).
13. lines 180-183 and Fig 5B: it's unclear why data from tolerance on imipenem is combined with resistance to macrolides to benchmark predictions. Why this combination of tolerance and resistance vs any other?
14. line 198: most of the protein synthesis inhibitors tested here are not known to cause mistranslation, and only inhibit translation.
15. line 201; this is not entirely true for linezolid (complemented mutant has also a significantly different time-kill curve compared to wildtype too).
16. lines 231-233: this has been challenged before. See for example PMID: 34646011 - there is shown that antibiotics can change from bacteriostatic action to rapid killing depending on the species and strain tested.

Reviewer #3 (Remarks to the Author):

In the present report, Jovanovic and collaborators report an exciting and novel antimicrobial single cell testing to overcome historical limitations of measuring susceptibility in bulk bacterial populations. This is an impressive tour de force new method that have the potential to significantly improve our understanding of how antibiotics work and clinical decision-making. The quality of the data and the analysis is very high.

There are, however, some aspects that need to be considered to strength the message of the manuscript (and the approach). I have some suggestions for improving the manuscript:

The cell viability is measured by propidium iodide accumulation after penetrating the damaged bacterial cell wall. Imipenem and cefoxitin, which target the cell wall stands out as the more effective drugs against both *M. tuberculosis* (Figure 1d) and *M. abscessus* (Figure 3b). Could antibiotics targeting cell wall facilitate PI penetration, possibly leading to overestimation of death rates in those conditions? On the other hand, is possible an underreporting of cell death by non-membrane-disrupting drugs? (increasing the chance of false negative)

The experiments presented in Figure 1D do need an untreated control as it is important to know how the system affects cell viability in general. There are no SEM values for Linezolid, Imipenem and Cefoxitin. Would it be because there is no variability? I understand the authors do not want to assess single-cell heterogeneity in this manuscript as mentioned in the main text. However, it would be interesting to assess the variability between antibiotic at the bulk level as well as the temporal level. For example, if we consider Amikacin or Moxifloxacin, it seems that the variability (here shown by the SEM) is higher in later timepoints. Could it suggest that within experiments, early-death (24h) mediated by ATBs is less variable than late-death? Could the authors comment on this.

Bactericidal and bacteriostatic effects can switch depending on antibiotic concentration. Hence, I suggest reinforcing the Figure 1 to add for one or two antibiotic a growth assessment using different concentrations.

The authors state that INH, RIF and EMB are more active against exponentially growing *M. tuberculosis*. I would change the wording, as we know that at the single-cell level *Mtb* seems to have a linear growth (Chung et al. 2024), even though I understand that the authors refer to bulk growth.

In Figure 2, the manuscript would benefit if the authors can show all the combinations possible of the 6 antibiotics in both rich and starvation media. Moreover, how the authors can exclude that in the comparison between rich and starvation media, the drugs are binding to media components and do not enter the bacterial cell?

Regarding the Figure 2 and the link between the ASCT in vitro killing efficacy under bacterial starvation (PBS and pantothenate) and the predicted in vivo outcome. Dormant/metabolically inactive phenotypes generated in vitro could mirror drug-tolerant bacilli in vivo. However, in vivo lesions have multiple microenvironments (oxygen tension, pH, lipids) that cannot be addressed by the in vitro ASCT starvation assay. The in vitro assay uses constant concentrations of antibiotics, which don't reflect dynamic drug exposure that happens in vivo. These limitations (as mentioned in the discussion "Furthermore, our study did not specifically mimic complex mycobacterial infection environments, such as intracellular, niches or granulomas, which may alter tolerance phenotypes") may affect the comparison between the in vitro results in this work and the in vivo (mouse and clinical trials) results from references 28 and 29.

Moreover, a limitation of the correlation between the in vitro data presented and the results from mice/clinical trials relies on the strain of *Mtb* used. I acknowledge the "tour-de-force" of the method, however, as mentioned by others, antibiotic efficacy can be affected by *Mtb* localization in cells (Santucci et al, 2021), therefore potentially in vivo. This is not discussed in the results section or discussion. The authors should discuss this and acknowledge the limitations of this correlation.

In the main text, L119-120, "which probably shares fundamental principles of antibiotic survival with other mycobacteria", it is a bold sentence, not referring to any study. Also, *Mtb* and *Mab* share their genus, but differs through their growth rate. I would tone down this sentence as it can be misleading.

One criticism that could be made about the "genetic" bacterial background is that we do not know, within those clinical isolates, if patients have been differentially treated with different antibiotics that could have generate tolerance, unrelated to the genetic

background. Could the authors comment on the distribution of the strains?

In Figure 6, The data regarding MAB_0233 are quite interesting, did the authors looked for similar genes in other mycobacteria (maybe some of them that are more tolerant to those drugs, in clinics)?

In order to validate the method, the use of other markers of cell death or metabolic activity (for example resazurin) could be implemented into the Antimicrobial Single-Cell Testing (ASCT) to compare results.

Reviewer #3 (Remarks on code availability):

It works

Reviewer #4 (Remarks to the Author):

This manuscript addresses a critical issue in the treatment of Mycobacterium infections: the disconnect between in vitro assessments of antibiotic susceptibility and actual in vivo outcomes. The authors develop a microscopy-based method, termed Antimicrobial Single-Cell Testing (ASCT), to measure antibiotic killing dynamics in real time at single-cell resolution, and correlate these measurements with infection outcomes in vivo. They further perform genetic analysis to identify determinants of drug tolerance.

Overall, I find this manuscript to be both timely and of significant potential interest. The authors thoroughly characterize in vitro efficacy and, through statistical analysis, relate it to clinical outcomes. A wide range of antibiotics and mutants are examined in order to identify generalizable features that explain treatment outcomes. This integrative approach has value for the evidence-based design of diagnostics and treatments.

That said, I found some parts of the manuscript difficult to follow. In several instances, explanations are either insufficient or potentially misleading due to a lack of clarity or contextualization. Moreover, the manuscript does not clearly distinguish which findings are novel contributions of this study versus reiterations or confirmations of previously established knowledge.

Taking an abstract as an example, the description of ASCT as a “large-scale imaging approach that quantifies bacterial killing in real time at single-cell resolution”. Many recent studies have applied single-cell imaging approaches to monitor antibiotic killing. It would be helpful if the authors clarified what specifically differentiates ASCT from these existing methods—whether it's the throughput, resolution, experimental design, or analytical framework.

Likewise, the statement that “drug-specific killing dynamics – rather than bacterial growth inhibition – predict infection outcomes” is ambiguous. Does this imply that bactericidal antibiotics are more predictive than bacteriostatic ones? Or by bacterial growth inhibition, do authors allude to slow death (tolerance)?

The authors also state that “antibiotic killing is not merely a drug effect, but a genetically encoded bacterial trait (drug tolerance) distinct from classical drug resistance.” Does this statement try to mean that there are genetic mechanisms for drug tolerance? Indeed, the existence of a genetic base of tolerance is a well-accepted idea in microbiology. I saw the author cited “43. Levin, B. R. & Rozen, D. E. Non-inherited antibiotic resistance. Nat. Rev. Microbiol. 4, 556– 367 562 (2006).” to support that drug tolerance is non-genetic. But, this paper is more than 20 years old. There have been significant advances since then.

I agree with the central premise that bacterial tolerance may play an underappreciated role in mycobacterial infections and that MIC values may not correlate well with clinical success. However, I caution against generalizing this finding to all bacterial infections. There is ample evidence in other systems where high MIC values are associated with treatment failure. The manuscript should be more precise in delimiting the scope of its claims.

The treatment of heritability also warrants more explanation. While a technical paper is cited (ref 48), the meaning of “heritability” in this context remains unclear to me. If the authors are arguing for genetic inheritance of drug tolerance, does it warrant addressing possible confounding factors such as horizontal gene transfer, repeated spontaneous emergence, or clonal spread of strains within communities?

Some figures are hard to understand and not clear. I think authors should strengthen figure captions. These are just a few that stood out the most.

- Fig 1d. What is the sample size of each rep? Similar to Fig. 1c? I vaguely remember in suppl materials that wells with <500 or <1000 cells discarded so I'm assuming sample size is quite large.
- Fig2b. It is difficult to evaluate the graph without standard deviation. The vertical lines are distracting. I am not sure what information the black and white dots show other than drug combinations.
- Fig 3a. Does color mean anything?
- Fig 3e. R² value is very low and slopes are small shown in Ext Fig 6. Are these statistically meaningful?
- Fig 4a. Standard error should be added. Growing isolates should be added since they are the true resistant strains. I'm assuming “resistance” and “tolerance” are selected based on MIC and AUC? It should be clarified in legend.
- Fig 5a. Is there a reason that individual data (like in Fig. 2cd, which were very helpful) is missing here?

Version 1:

Reviewer comments:

Reviewer #1

(Remarks to the Author)

Dear Authors,

I just want to congratulate you for a great piece of work. I highly appreciate the detailed responses which themselves are highly valuable. My concerns have been addressed and I hope I have been able to help to clarify some minor points of this great work

(Remarks on code availability)

Reviewer #2

(Remarks to the Author)

This revised version is a considerable improvement of an already very strong first submission - including new experiments and analysis. All my comments have been adequately addressed. Kudos to the authors for the excellent work.

(Remarks on code availability)

Reviewer #3

(Remarks to the Author)

Many thanks for the authors to address my concerns and suggestions. They have done an outstanding job in answering point by point to the reviewers concerns and the response is very strong and convincing. I am happy to recommend publication.

(Remarks on code availability)

It works

Reviewer #4

(Remarks to the Author)

I am happy with the author's revision.

Minsu Kim

(Remarks on code availability)

Decision Letter:

Our ref: NMICROBIOL-25031020A

21st October 2025

Dear Lucas,

Thank you for submitting your revised manuscript "Antibiotic lethality dictates mycobacterial infection outcomes" (NMICROBIOL-25031020A). It has now been seen by the original referees and their comments are below. The reviewers find that the paper has improved in revision, and therefore we'll be happy in principle to publish it in Nature Microbiology, pending minor revisions to comply with our editorial and formatting guidelines.

If the current version of your manuscript is in a PDF format, please email us a copy of the file in an editable format (Microsoft Word)-- we can not proceed with PDFs at this stage.

Thank you again for your interest in Nature Microbiology. Please do not hesitate to contact me if you have any questions.

Sincerely,

Reviewer #1 (Remarks to the Author):

Dear Authors,

I just want to congratulate you for a great piece of work. I highly appreciate the detailed responses which themselves are highly valuable. My concerns have been addressed and I hope I have been able to help to clarify some minor points of this great work

Reviewer #2 (Remarks to the Author):

This revised version is a considerable improvement of an already very strong first submission - including new experiments and analysis. All my comments have been adequately addressed. Kudos to the authors for the excellent work.

Reviewer #3 (Remarks to the Author):

Many thanks for the authors to address my concerns and suggestions. They have done an outstanding job in answering point by point to the reviewers concerns and the response is very strong and convincing. I am happy to recommend publication.

Reviewer #3 (Remarks on code availability):

It works

Reviewer #4 (Remarks to the Author):

I am happy with the author's revision.

Minsu Kim

Version 2:

Decision Letter:

11th November 2025

Dear Lucas,

I am pleased to accept your Article "Large-scale testing of antimicrobial lethality at single-cell resolution predicts mycobacterial infection outcomes" for publication in Nature Microbiology. Thank you for having chosen to submit your work to us and many congratulations.

Authors may need to take specific actions to achieve compliance with funder and institutional open access mandates. If your research is supported by a funder that requires immediate open access (e.g. according to [Plan S principles](https://www.springernature.com/gp/open-science/plan-s-compliance) or the [NIH public access policy](https://www.springernature.com/gp/open-science/us-federal-agency-compliance)) then you should select the gold OA route, and we will direct you to the compliant route where possible. Because authors warrant under our subscription licensing terms that they haven't committed to licensing any version of their article under a licence inconsistent with the terms of our agreement – including the applicable embargo period – publication under the subscription model isn't suitable for authors whose funders require no embargo.

Congratulations once again and I look forward to seeing the article published.

With kind regards,

P.S. Click on the following link if you would like to recommend Nature Microbiology to your librarian
<http://www.nature.com/subscriptions/recommend.html#forms>

** Visit the Springer Nature Editorial and Publishing website at http://editorial-jobs.springernature.com?utm_source=ejp_NMicro_email&utm_medium=ejp_NMicro_email&utm_campaign=ejp_NMicro for more information about our career opportunities. If you have any questions please click [here](mailto:editorial.publishing.jobs@springernature.com).**

Reviewer #1 (Remarks to the Author):

Overall Comment

The authors argue that growth-based methods may be limited in predicting treatment outcomes, and propose bacterial killing as a potentially more informative alternative to MIC determination for *Mycobacterium tuberculosis* and *Mycobacterium abscessus* infections. Their approach seems well-suited to studying bacterial tolerance at scale and evaluating treatment effectiveness. The amount and elegance of the work are commendable, and it addresses an important and often underexplored area in Mycobacteria research, which is gaining recognition for its relevance to treatment response. That said, it is suggested that this method can predict to individual treatment outcomes in a large extent, at least deserving clinical considerations, but I am not sure if this is fully supported by the data as presented. The authors do convincingly highlight the relevance of bacterial killing in understanding treatment responses, but the connection to individual clinical outcomes could benefit from further development or clarification. Below, I outline some major and minor points that may help strengthen the manuscript:

We thank the reviewer for the thorough evaluation of our manuscript and for highlighting its potential relevance in mycobacterial infections. We appreciate the recognition of the scale and novelty of our approach, as well as the broader relevance of this work for understanding antibiotic efficacy. We also acknowledge the reviewer's concerns, specifically regarding individual patient outcomes.

1) In the *M. tuberculosis* experiments, non-clinical strains are used to compare treatments that approximate the standard of care (SoC) versus those considered better. As such, the analysis focuses more on regimen performance than individual treatment outcomes. Additionally, the tested regimens involve combinations rather than individual clinical outcomes, and the strains are not derived from patient samples. In contrast, the *M. abscessus* experiments involve individual strains linked to treatment outcomes (e.g., treatment duration), yet the predictive utility of tolerance at the individual level remains unclear. While the results are highly relevant, the current framing of the study could be reconsidered — perhaps shifting from individual outcome prediction to a more general evaluation of regimen efficacy, especially given the differing approaches across the two models.

Ad 1.1) The reviewer is correct that in our *M. tuberculosis* experiments (Fig. 2), we show that *in vitro* time-kill kinetics of different drug regimens are predictive of *in vivo* treatment outcomes at the population level, in mouse models and patient cohorts. These findings suggest that drug-related (regimen-related) killing dynamics, as quantified by ASCT, are relevant and could be used to prioritise more effective drug combinations for tuberculosis.

While this first part of the study focuses on regimen comparisons using the same non-clinical *M. tuberculosis* strain, the second part (from Fig. 3 onward) shifts to assessing and comparing different clinical *M. abscessus* isolates from individual patients (strain-level), all tested under identical drug exposures. For example, at a fixed concentration of imipenem, we observed that isolates with high imipenem tolerance were more likely to persist in patients (i.e. remained culture positive after 6 months of treatment), whereas low-tolerance isolates were associated with sustained culture-conversion. These associations with clinical outcome were observed across multiple drugs at multiple concentrations, and the predictive effect size of drug tolerance was similar to macrolide resistance (the only MIC–outcome association), highlighting the robustness of this finding. Considering that there exists little direct evidence linking drug tolerance to clinical outcomes, our findings represent some of the strongest data to date supporting drug tolerance as a relevant mechanism underlying treatment failures. Moreover, these findings establish drug tolerance as a measurable phenotype (analogous to MICs) that could eventually be used to personalise antibiotic treatments.

Overall, our study demonstrates that antibiotic killing and survival, as measured by ASCT, are critical determinants of treatment outcomes, no matter whether driven by drug properties (as in *M. tuberculosis* regimen comparisons) or by bacterial traits (as in *M. abscessus* drug tolerance). We acknowledge that

this shift from evaluating drug effects at a population level in *M. tuberculosis* to assessing isolate variation at the individual level in *M. abscessus* may be challenging to the reader. To address this, we have revised both sections of the manuscript.

2) For example, in the *M. tuberculosis* section, the authors state: “we found that drug-specific killing dynamics rather than bacterial growth inhibition predict infection outcomes in mice and humans.” While this interpretation may be plausible, it may not be directly tested within the present dataset. Since the strains used are not linked to specific patient outcomes, it's difficult to assess how representative they are of real-world infections. The role of killing in understanding treatment combinations is well demonstrated, but the connection to actual infection outcomes might be more nuanced than the current wording suggests.

Ad 1.2) This sentence was indeed imprecisely phrased and may have implied individual-level information that our *M. tuberculosis* data do not support. The *M. tuberculosis* experiments were conducted using non-clinical strains and were designed to compare the bactericidal activity of different drug regimens. As such, they were not intended to predict individual patient outcomes, but rather to assess whether *in vitro* killing dynamics could inform on treatment outcomes at the population level.

The goal of these experiments was to determine if drug-specific killing patterns are associated with treatment efficacy of previously published mouse models and human clinical trials. In drug development, MIC values from a single or a few bacterial strains are used to estimate a drug's overall potency, not to predict individual patient outcomes, but to inform on expected population level response rates. In our study we found that MICs (including drugs that are considered active) did not correlate well with treatment outcomes, whereas ASCT-derived time-kill kinetics did. We showed that two genetically distinct avirulent *M. tuberculosis* strains (H37Ra and mc²7000) yielded *in vitro* killing patterns that were strongly associated with treatment efficacy *in vivo*, which was assessed with virulent strains. This suggests that such differences in killing, even when assessed with avirulent strains, can be informative for identifying promising regimens and prioritising them for further development.

We have revised the manuscript accordingly to clarify that our *M. tuberculosis* data support associations between *in vitro* killing dynamics and population-level treatment efficacy, without implying direct prediction of patient level outcomes.

3) In the summary, the manuscript states: “We show that drug tolerance is conserved across similar drug targets, identify a phage protein that modulates antibiotic killing, and demonstrate that strain-specific killing dynamics shape individual patient outcomes — similar to, but independent of, drug resistance.” This raises an important conceptual point: is the phenomenon strain-specific (as suggested by the *M. abscessus* findings) or more broadly conserved (as implied by the *M. tuberculosis* data)? The latter interpretation may require more cautious language, and it might be helpful to reframe the *M. tuberculosis* data in terms of regimen efficacy rather than individualized outcome prediction. If I've misunderstood this distinction, I'm happy to be corrected.

Ad 1.3) The reviewer is correct in distinguishing between drug-driven effects, as shown in *M. tuberculosis*, and strain-specific bacterial effects, as demonstrated in *M. abscessus*. In antibiotic development, drug activity is typically assessed using MIC values from a limited number of bacterial strains, providing an overall estimate of potency. However, once this drug is applied to a diverse bacterial population (particularly under selective pressures) its efficacy is influenced by strain variation and adaptation. This principle applies not only to MICs, but also to bacterial killing: while some drugs (or regimens) have superior overall killing properties, the actual killing effect is modulated by the characteristics of individual bacterial strains. This concept is reflected in the distinction between bacteriostatic and bactericidal drugs, which describe general drug class behaviour but do not capture the variability in killing across different strains. In the revised manuscript we more clearly distinguish between drug-specific and individual strain-specific killing effects.

4) Regarding Figure 2B — it's somewhat surprising that a clearly sterilizing regimen like RIF-PYR appears to perform relatively poorly, given its well-established efficacy in the literature. Similarly, BPaL is not generally considered superior to such regimens. In addition, why most bedaquiline-containing combinations seem to outperform others?, this is worth noting. It may be useful to address this discrepancy or clarify how this figure aligns with existing clinical data.

Ad 1.4) We agree that the rifampicin-pyrazinamide combination is considered sterilising and was critical in shortening treatment to the current standard six-month treatment. However, in clinical practice these two drugs are always given as part of a multidrug regimen, and have hardly been investigated alone. In our data, regimens containing rifampicin showed improved killing under metabolically active conditions, but we did not observe an additional benefit from pyrazinamide. This may reflect a limitation of our current setup, which does not reflect host environments, such as compartments with low pH where pyrazinamide is most active. This limitation is now discussed in the revised manuscript. Regarding bedaquiline- and pretomanid-containing regimens, several animal and clinical studies (e.g. BPaZ or BPaMZ) have demonstrated the treatment-shortening potential¹⁻⁴, consistent with our finding that such combinations show strong killing under starvation conditions.

5) On the clinical trial data: it would be helpful to provide clear references to the source studies. The current citation does not seem to include human data. It appears that the analysis draws on phase IIA and IIB trials, where culture conversion — rather than relapse — is typically the primary endpoint. Since relapse might correlate more closely with bacterial killing, this could affect the interpretation. It may be helpful to clarify what is meant by "clinical outcome" in the manuscript and how closely the trial endpoints align with this definition.

Ad 1.5) We used the previously established standard-of care (SOC) versus better-than-SOC classification from Larkins-Ford and colleagues⁵, which is based on ~30 prior publications (all referenced in data S2 of that article). Given the citation limits and considering that this represents only a part of our study, we cannot cite each original reference individually.

We apologise for not being clear on *M. tuberculosis* clinical outcomes in phase 2a and 2b clinical trials. As the reviewer notes, these studies assess microbiological endpoints (bactericidal activity) rather than clinical cure. We now specify in the manuscript that clinical outcome in this context refers to bactericidal activity. A systematic analysis of phase 3 studies assessing clinical relapse was not within the scope of this study, would have been prone to selection bias, and would likely yield smaller numbers.

6) One point that could benefit from clarification relates to the assumption that clinical outcomes can be predicted by in vitro metrics like MIC or bacterial killing. While bacterial factors are clearly under-investigated — and this study offers valuable insight — treatment outcomes are influenced by a wide range of factors, including host responses, pharmacokinetics/pharmacodynamics, adverse events, loss to follow-up, and more. For instance, two regimens with similar in vitro killing profiles could differ significantly in real-world effectiveness due to non-bacterial factors. Acknowledging this complexity might help frame the results in a more balanced and clinically relevant way.

Ad 1.6) By advancing from bulk growth measures to single-cell viability measures, we show that antibiotic killing (drug- and strain-specific) is a critical but yet underappreciated marker of treatment outcomes. However, similar to MICs, time-kill kinetics alone cannot fully account for real-world treatment efficacy, as it does not capture other key contributors to treatment failure. These include factors such as limited drug penetration, toxicity, heterogeneous infection environments (e.g. granulomas and intracellular niches), and host immune responses. We now highlight several of these non-bacterial factors in the revised discussion.

7) In the *M. abscessus* experiments, the predictive link between tolerance and clinical outcome might be strengthened by ensuring that MIC and killing data correspond only to the drugs actually used for each patient. From the text, it appears that all drugs were tested across all strains. If treatment-specific drug-strain pairs are not available, it may help to note this as a limitation. If this matching was done but not clearly described, a brief explanation could clarify.

Ad 1.7) The reviewer raises an important point. ASCT-based drug tolerance measurements were generated for each clinical *M. abscessus* isolate across a fixed panel of antibiotics, independent of the specific treatments received by each patient. This lack of drug-strain matching may have led to an underestimation of the true effect size of tolerance on clinical outcome. There were two main reasons that did not allow us to address drug tolerance in the context of the treatments used. First, for most patients detailed treatment data were not available, as collecting and retrospectively standardising such data is highly time-consuming. Second, treatment regimens in *M. abscessus* infections are typically complex (see example below), making consistent interpretation extremely challenging. Nevertheless, given the central role of amikacin and beta lactams in *M. abscessus* treatment (Table), we assume that a large proportion of patients received amikacin (in some studies up to 100%) and many were likely treated with a beta lactam (imipenem or ceftazidime, tolerance of both highly correlated). We now acknowledged this in the discussion and clarified this point in the methods.

Figure: Example of antibiotic regimen used to treat *M. abscessus* pulmonary disease in a single patient.

	De Mello et al.	Ellender et al.	Jarand et al.	Koh et al. 2016	Koh et al. 2016	Namko ong et al.	Park et al.	Van Ingen et al.
Amikacin (%)	69	100	75	100	100	100	78	88
Ceftazidime (%)	0	77	17	17	2	0	42	13
Imipenem (%)	4	0	51	24	48	31	42	63

Table. Treatment characteristics of patients with *M. abscessus* pulmonary disease, modified from Kwak et al. ⁶

8) Regarding the role of MIC: while the authors suggest it is a poor predictor of treatment outcomes, it is important to recognize that resistance — typically defined by an MIC above a breakpoint — remains a useful clinical tool for guiding therapy. This is especially true in cases where resistance is well characterized.

Ad 1.8A) We completely agree that MIC testing (resistance testing) is a cornerstone of antibiotic therapy and essential for clinical decision-making, as highlighted in MDR/XDR tuberculosis and macrolide-resistant *M. abscessus*. However, antibiotic treatment can also fail despite bacteria being susceptible, suggesting MICs alone do not fully capture treatment outcomes. Our findings provide evidence that drug tolerance contributes to treatment failure in *M. abscessus* infections, even in the absence of resistance. In the revised manuscript we have clarified that MICs are indispensable for identifying resistance and guiding therapy, while antibiotic tolerance represents an additional, complementary predictor of treatment efficacy (see also response 4.5 to reviewer 4).

While bacterial killing may indeed offer additional predictive value, it might be helpful to more explicitly acknowledge the existing utility of MIC-based breakpoints and the rationale behind combination therapies.

Ad 1.8B) While MIC breakpoints are well established for *M. tuberculosis*, most *M. abscessus* breakpoints (apart from macrolides and aminoglycosides) are derived from MIC distributions rather than clinical outcome data, limiting their clinical utility.

The rationale for combination therapy is indeed manifold, including reduced resistance evolution, targeting diverse host niches and bacterial populations, and achieving synergistic antibacterial effects. Importantly, our killing assessments capture antibiotic activity in a single environment, but do not account for all mechanisms contributing to *in vivo* efficacy.

9) As mentioned earlier, the *M. tuberculosis* experiments seem more focused on evaluating the correlation between bacterial killing and regimen efficacy rather than predicting individual outcomes. These are not clinical strains tied to treated patients, and no direct real-world validation of individual outcomes is included.

Ad 1.9) As noted above, the goal of assessing drug regimens in non-clinical *M. tuberculosis* strains was to streamline drugs and drug combinations for development and clinical trials. Personalised treatment decisions would require testing individual patient samples, which was not the aim of our *M. tuberculosis* experiments. However, we performed such analysis for *M. abscessus*, where we showed that ASCT-derived time-kill kinetics are associated with real-world patient outcomes in *M. abscessus* pulmonary disease. We have now clarified this point throughout the manuscript.

10) The grouping of regimens into SoC and BSoC is clear, but since there is overlap in bacterial killing values between the groups, showing individual-level comparisons may offer additional insight. The pooled data suggests a strong correlation between killing and regimen classification, but given that individual values are available, it might be useful to also show paired SoC-BSoC comparisons.

Ad 1.10) It is not entirely clear to us whether the reviewer's reference to "pooled data" and "individual-level comparisons" refers to drug-strain pairs, drug regimens or our outcome data.

Regarding strain level-data: In our current *M. tuberculosis* analysis, individual drug-strain pairs are not available. Animal experiments were conducted using a single virulent *M. tuberculosis* strain, while the human data are based on different clinical strains. Unfortunately, these clinical *M. tuberculosis* strains cannot be assessed in our biosafety-2 setup. However, we agree that future prospective studies, (e.g. within a randomised controlled trial), that incorporate defined drug regimens and patient isolates, would be ideal to link drug- and strain-specific killing effects.

Regarding drug combinations: Each data point shown in the relevant figures (e.g. Fig 2) represents a unique drug combination tested using ASCT (averaged between replicates).

Regarding outcome data: In order to minimise selection bias and ensure consistency, we used two publications that already summarised outcomes across multiple animal and clinical studies. Given the heterogeneity in study design and outcome metrics, the authors classified the regimens into SoC and BSoC categories rather than continuous outcome measures.

Are there instances where SoC regimens performed better? Or cases where the difference was not statistically significant? This could also help contextualize the statement: “Overall, 11 out of 12 associations between *M. tuberculosis* killing in starvation and clinical outcomes reached statistical significance.” Including individual data or related statistical tests might strengthen this conclusion.

The reviewer is correct that *in vitro* time-kill kinetics were not fully discriminative of *in vivo* outcomes. For example, moxifloxacin-pretomanid-pyrazinamide performed well in starved models, but was classified as SOC in phase 2 clinical studies (though better-than-SOC in mouse models); similarly, bedaquiline-pretomanid, which was classified as SOC in the bactericidal mouse model but better-than-SOC in the relapsing mouse model. Several factors may explain these exceptions:

- As noted in 1.6, ASCT quantifies bacterial killing only and does not account for non-bacterial determinants of treatment success such as host immunity, pharmacokinetics or diverse infection environments. These additional factors (including drug resistance) can substantially contribute to treatment outcomes.
- The mouse models used to define SoC and better-than-SoC are imperfect predictors of clinical efficacy in human ^{7,8}. In our dataset, correlations between different mouse models, as well as mouse models and clinical phase 2 studies were only modest (Table) ^{5,9}. Despite this, killing of starved *M. tuberculosis* showed strong and consistent associations with preclinical models and clinical outcomes, even if not perfectly aligned.

The one exception where ASCT in starvation did not correlate with *in vivo* outcomes was the association of the PBS-starved H37Ra strain with outcomes in the bactericidal C3HeB/FeJ mouse model (p=0.16). This analysis included only 15 drug regimens, limiting statistical power to detect moderate effect sizes. All individual data are provided in Source Data Fig. 2.

	RMM	BMM	gBMM	Clinical trials
RMM	1.00	0.68	0.60	0.31
BMM	0.68	1.00	0.66	0.31
gBMM	0.60	0.66	1.00	0.45
Clinical trials	0.31	0.31	0.45	1.00

Table: Pearson correlation coefficient between *in vivo* mouse and clinical outcome data. RMM: relapsing mouse model, BMM: bactericidal mouse model, gBMM: granulomatous bactericidal mouse model.

11) Finally, returning to the broader theme — the complexity of clinical outcomes — it may be helpful to consider whether two regimens with similar *in vitro* killing could still diverge in real-world performance due to non-bacterial factors (e.g., side effects, adherence, drug access). If so, interpreting regimen efficacy solely based on bacterial killing might oversimplify the clinical picture. This nuance could be acknowledged to round out the interpretation of the findings.

Ad 1.11A) We addressed this important comment in our response to comment 6, and have incorporated it into the discussion of the revised manuscript.

11) The correlation between the method used to assess bacterial killing and survival determined via a standard killing curve appears to be quite low (0.14, Fig ED_FIG4_B). This result isn’t discussed in much detail in the manuscript. Could the authors comment on whether this discrepancy might influence the

overall conclusions? Additionally, it's not fully clear why day 7 was chosen as the point of correlation rather than MDK99 for example. I guess it is related to starvation but clarification of this choice would help the reader better understand the rationale.

Ad 1.11B) The main goal of this work was to identify clinically relevant *in vitro* measures of antibiotic activity. Our rationale is that markers capable of predicting regimen-level or individual patient outcomes can inform on drug development and clinical treatment decisions. Despite its limited evidence^{10,11}, bactericidal drug activity is considered a favourable drug feature, while drug tolerance is assumed to limit treatment efficacy. Traditionally, both are measured using colony forming unit (CFU) assays.

While CFUs are the gold standard for assessing viability, distinguishing live from dead bacteria death is inherently difficult. Actively dividing cells clearly indicate viability, and lysed cells are dead; however, a large number of bacteria are neither dividing nor lysed, especially in mycobacteria¹². CFU assays measure the ability to form colonies and are thus a proxy for replicating bacteria (a live marker, where a signal indicates living bacteria). However, cells that do not generate a colony are not necessarily dead. Some, stressed or starved bacteria may form colonies after a prolonged lag or remain viable but not culturable (VBNCs), while still metabolically active and potentially clinically relevant. Others might be culturable only in conditions with specific supplements (differentially culturable bacteria), potentially increasing bacterial numbers by several log steps¹³. Additionally, CFU assessments are made after antibiotic washout and macroscopic bacterial regrowth, introducing a delayed readout and further uncertainty due to post-antibiotic effects or regrowth-related killing¹⁴.

In contrast, ASCT uses viability staining (PI; propidium iodide) as a real-time marker of membrane integrity and cell death. We show that PI-positive cells do not regrow after antibiotic removal, validating PI as a reliable death marker. Moreover, in the revised manuscript (discussed in more detail in 1.12) we also show that after beta-lactam or quinolone treatment and antibiotic washout most PI-negative cells are able to express a fluorophore upon transcriptional induction. Given this conceptual (CFUs: live marker; ASCT: uses PI, a marker of bacterial death) and temporal differences (CFUs: delay of several days; ASCT: real-time), it is not surprising that the correlation between CFU-based and ASCT-based viability was low (both were assessed under starvation conditions; now clarified in the figure legend). But in contrast to CFUs, ASCT-based killing assessments in starved bacteria were highly predictive for *in vivo* treatment outcomes.

We compared CFUs (live cell fraction at day 7) with ASCT-based time kill kinetics using a measure of overall killing over a 7-day period (area under the 7-day time kill curve). Due to limited sampling (only two CFU timepoints), calculating a CFU-based AUC was not meaningful. The minimum duration to kill 99% (MDK99) could be used as an alternative¹⁵; however, was not applicable to ASCT due to the lesser degree of overall killing. CFU-derived MDK99 values correlated well with 7-day live cell fractions (Pearson R: 0.86), showed a similarly low correlation with ASCT AUCs (Pearson R: 0.36, R²: 0.13) and a poor association with *in vivo* outcomes (Figure). We also calculated ASCT-based live cell fractions at day 3 and day 7 (comment 2.2iv in response to reviewer 2). These static values were, similarly to AUC values, predictive of *in vivo* outcomes, supporting the robustness of the ASCT approach across multiple metrics.

Figure: Association of CFU-based MDK99 metrics with *in vivo* outcomes. RMM: relapsing mouse model, BMM: bactericidal mouse model

12) In validating that propidium iodide (PI) reliably stains dead cells, the authors report that only “1–11% of the PI-negative bacteria regrew in antibiotic-free media” during the first 24 hours post-treatment. It would be helpful to discuss why the remaining 89–99% of PI-negative cells did not grow. Does this imply that these cells were viable but non-culturable, or are they likely dead? This raises the question of how definitively PI staining alone can be used to indicate cell death. Additional discussion of this point would clarify how robust PI is as a proxy for viability in this context.

Ad 1.12) The reviewer highlights an important point regarding the interpretation of PI staining as a marker of viability, which was to some extent addressed in comment 1.11. Given that we did not find that PI-positive cells regrow after antibiotic washout, they can be considered dead. However, as the reviewer points out, the viability state of PI-negative cells is less clear. Below, and in the revised manuscript, we outline several plausible reasons why PI-negative bacteria (assumed viable in our assay) may fail to regrow in antibiotic-free media (assumed dead in the CFU assay). In addition, we now provide new experiments that support PI as a real-time, single-cell proxy of viability, while highlighting fundamental limitations of delayed regrowth-based readouts.

- Some bacteria may remain metabolically active but are unable to regrow under the given conditions. This may occur due to prior starvation or stress responses resulting in prolonged lag or specific media requirements (e.g. resuscitation factors). These cells may be in a dormant state, capable of survival but not replication. As such, while PI-negative, they would fail to form colonies and would be incorrectly classified as dead in CFU-based assays. We have now assessed if PI-negative cells are able to synthesise new proteins after antibiotic washout, by expressing a fluorophore (GFP) upon transcriptional induction (**Figure**). We observed that after beta-lactam or fluoroquinolone treatment, almost all PI-negative bacteria (considered alive using PI) can express new GFP, and therefore should be considered alive. For example, while 43% of bacteria were able to express new GFP after cefoxitin treatment, only 4.2% regrew after antibiotic washout. Thus, regrowth measures were underestimating bacterial survival ~10-fold.

Figure: New GFP expression and PI fluorescence after 12-hour cefoxitin treatment and antibiotic washout. (f) Brightfield, GFP, and PI fluorescence images, with merged overlays, acquired ~1 hour after antibiotic removal and GFP induction. (g) Distribution of bacterial fluorescence intensities with gating of PI- and GFP-positive and negative populations.

- Bacteria may remain viable at the time of antibiotic removal (PI-negative), but die shortly thereafter. This death in antibiotic-free conditions could result from incomplete antibiotic washout (e.g. complete washout is almost impossible for aminoglycosides), post-antibiotic effects (where damage continues after drug removal), oxidative or nutrient shock following the exposure to nutrient-rich media, or the activation of programmed death pathways. Our new analysis of GFP induction and PI staining after antibiotic washout supports the presence of post-exposure killing. While a large fraction of bacteria was able to synthesise GFP immediately after washout, the proportion of PI-positive cells increased steadily over the following 12 hours (**Figure**). This indicates that many bacteria initially capable of new protein synthesis die during the recovery phase in nutrient rich media.

Figure: GFP and PI fluorescence intensities of *M. abscessus* populations treated with amikacin (AMK), cefoxitin (FOX), or moxifloxacin (MXF) following antibiotic removal. Time (hours) indicates imaging time points, with 0 set to the start of imaging (~1 hour after GFP induction).

- It is also possible that some bacteria are dead but not immediately detected by PI. Death may occur without sufficient or immediate membrane damage to allow PI entry. Nevertheless, we argue that regardless of the precise mechanisms of bacterial death every dead cell will ultimately lose membrane integrity. Thus, some bacteria classified as PI-negative during antibiotic exposure may already be dead and only became PI-positive later (e.g. after the washout). However, we would not expect such cells to express new GFP but rather to directly transition from PI-negative (GFP-neg) to PI-positive (GFP-positive). In contrast, we observed that most PI-negative bacteria expressed GFP at some point, and only later transitioned to PI positivity, suggesting that the majority died after antibiotic removal. While now presented as one of many Extended Data Figures, this finding is important, as it indicates that bacterial viability cannot be fully captured by delayed regrowth readouts. We recognise however that deciphering live versus death is an incredibly challenging task that goes far beyond the scope of this work, and must be addressed in much more detail in the future. ASCT, for example in combination with additional viability markers, could be a highly valuable tool to more precisely dissect these transitions.

In summary, PI- and CFU-based viability assessments have inherent limitations. CFUs underestimate viability while PI staining may overestimate it. We recognise that PI accumulation can be delayed, especially when killing is indirect (i.e. due to other mechanisms than cell wall damage, see comment

3.2). However, given that killing can be rapid regardless of the underlying mechanism, and considering our experiments extended over 3 to 7 days, such delays are unlikely to substantially influence our conclusions. Importantly, the goal of ASCT is not to replicate CFU-based viability, but to provide a real-time, single-cell and highly scalable proxy of bacterial killing that better correlates with *in vivo* treatment outcomes. While PI-based viability is not perfect, our data show that it yields insights that predict treatment success more robustly than CFUs. Moreover, the ASCT platform is adaptable and can be further refined by integrating additional viability markers, such as cell morphology, complementary fluorescence dyes or reporter strains. As also suggested by the reviewers 2 and 3, we have now evaluated resazurin-based metabolic activity as an independent viability marker (see comment 3.14A).

13) The *M. abscessus* work is a strong part of the manuscript and comes closer to linking bacterial factors with individual clinical outcomes, using patient isolates. However, I have some concerns about the interpretation of susceptibility to cefoxitin and imipenem. Drug susceptibility testing (DST) for these agents in *M. abscessus* is known to be unreliable in many settings. Could this affect some of the downstream analyses involving these drugs? In fact, in Fig 3B, they show remarkably high activity compared to others — it might be useful to comment on this result in the context of DST variability.

Ad 1.13) We agree that drug susceptibility testing of cefoxitin and imipenem in *M. abscessus* is relatively unreliable. This is supported by the poor agreement across DST testing¹⁶ and is likely a main factor for the low heritability observed for these drugs in our analysis (Figure 4a). Both drugs are chemically unstable in standard bacterial growth media, and this instability becomes increasingly problematic over prolonged incubation. Since MIC values are typically read between days 3 and 5, substantial degradation by that time may introduce considerable variability. In contrast, our killing assessments were performed using drug concentrations that were well above the MIC. Although degradation still occurs, its impact on killing is smaller since i) effective concentrations remain above the MIC (we did not observe growth) and ii) most killing occurs within the first 12 hours, before major degradation occurs. We observed strong correlations between time-kill kinetics of both imipenem (Pearson R: 0.89) and both cefoxitin concentrations (Pearson R: 0.97), despite analysing low and high drug concentrations in different experiments, supporting the robustness of our measurements. Furthermore, our data show that among drugs with confirmed activity, MIC values are poorly correlated with time-kill kinetics. In other words, a drug with a low MIC does not necessarily exhibit rapid killing, whereas a drug with a relatively high MIC may still have good killing properties (given that tested concentrations are above the MIC). The low stability of the beta-lactams cefoxitin and imipenem is now mentioned in the manuscript.

14) Regarding the use of *in vitro* bacterial killing to predict patient outcomes in *M. abscessus* cases: how well do the antibiotics tested *in vitro* match the treatments given to each patient? Are the authors testing individual drugs used in each patient's regimen, or using combinations similar to the *M. tuberculosis* part of the study? Clarifying whether killing data reflect the specific drugs administered to each patient would strengthen the link between the experimental data and clinical outcomes.

Ad 1.14) This important point is addressed in comment 1.7.

15) The statement that “drug tolerance is heritable” is an interesting and valuable contribution, as it supports the idea that drug tolerance has genetic determinants that can be studied. That said, some of the heritability values presented in Fig 4 raised questions. For example, if I'm interpreting the figure correctly, is the heritability of MIC for some antibiotics essentially zero? It might be helpful to briefly comment on this, especially in contrast to the heritability estimates for tolerance.

Ad 1.15) In our analysis, the MIC heritability for several antibiotics (particularly cefoxitin and imipenem) is very low. There are several potential explanations for this observation:

- MIC testing, especially for chemically unstable compounds, like imipenem and ceftazidime, is prone to significant variability. This is consistent with the well-documented challenges of imipenem/ceftazidime DST in *M. abscessus* (see also 1.13).
- Drugs that showed broad resistance ranges (high-level resistance), generally had higher heritability scores (e.g. macrolides and aminoglycosides), while drugs with low MIC variability tended to show lower heritability. A single log₂-fold difference is within the accepted range of reproducibility.
- MICs may be influenced by complex or subtle genetic mechanisms, including low-frequency variants and gene-copy number variation, that were not captured in our analysis. These factors can obscure genotype-phenotype association and reduce apparent heritability.

In contrast, ASCT-derived drug tolerance testing consistently showed higher heritability, suggesting that killing dynamics are strain-specific and under genetic control. In the revised manuscript we now discuss this contrast between MIC and tolerance heritability.

16) In reference to the conclusion that “drug tolerance predicts individual infection outcomes” (Fig 5 and related text): while the data do suggest an association, it's not entirely clear that individual outcome prediction was directly tested. For instance, in Fig 5a, it's hard to determine whether the data are presented at the individual level. A statement like “contributes to predicting” or “is associated with improved prediction of” might be more accurate unless further evidence is provided. Additionally, the ROC value for imipenem (0.68) suggests only moderate predictive performance. It might strengthen the analysis to include other performance metrics, such as positive and negative predictive values (PPV and NPV), to help the reader better understand the extent to which tolerance adds to clinical prediction

Ad 1.16) We show in Figure 5a the association between drug tolerance of individual *M. abscessus* patient isolates and the corresponding treatment outcomes in the same patients. In Figure 5b we assess the predictive performance of this association using the area under the receiver operating characteristic curve (AUC-ROC), a standard and well-established method for evaluating classification performance.

We believe the term “predictive” is appropriate and statistically justified, as AUC-ROC provides a threshold-independent metric that quantifies the ability of tolerance to distinguish between treatment success and failure. Vague terms like “contributes to prediction” or “associated with prediction” risks misinterpreting the analysis.

While an AUC-ROC of 0.68 might be considered modest under highly controlled experimental conditions, our analysis was conducted using real-world clinical data from patients with highly diverse lung pathologies, comorbidities, co-infections, variable antibiotic treatments and side effects. In this heterogeneous setting, having an *in vitro* phenotype of antibiotic activity, that is unrelated to MICs, but yields an AUC comparable to that of macrolide resistance (the only marker strongly associated with outcomes) is a clinically very meaningful result that is far beyond of what we expected. These results may not be only helpful to improve individual treatments but challenge a prevailing assumption that MIC-unrelated antibiotic treatment failures are mainly host related. We now additionally report threshold-dependent metrics such as positive and negative predictive values.

Model	AUC	Sensitivity	Specificity	PPV	NPV	F1_Score
Imipenem tolerance	0.68	0.59	0.79	0.85	0.67	0.69
Clarithromycin MIC	0.69	0.62	0.70	0.82	0.66	0.68
Combined MIC/tolerance	0.78	0.67	0.80	0.88	0.73	0.75

Reviewer #2 (Remarks to the Author):

In this study, Jovanovic et al. present a high-throughput method, ASCT, for measuring antibiotic killing in mycobacteria at single-cell level, and apply it in two clinically relevant areas. First, they probe 65 combinatorial antibiotic regimens used to treat *M. tuberculosis* (Mtb) and show that in-vitro killing of starved/non-growing Mtb cells, as measured by ASCT, captures better pathogen clearance in animal models and clinical studies, compared to traditional MIC measures or killing measured in growing cells or by conventional CFU/ml measures. Second, they move to *M. abscessus* and measure killing kinetics of 8 antibiotics in a collection of 405 exponentially growing clinical isolates using ASCT. They use these data to compare how well they predict clinical outcomes and how they are linked to growth characteristics of the strains and to the drug mode-of-action. They then genotype all strains to identify genetic elements that drive distinct tolerance elements, and follow up on a specific phage-tail protein that (partially) drives tolerance to protein-synthesis inhibitors. Overall, this is an impressive method with potential to change both our fundamental understanding of tolerance mechanisms and clinical practice guidelines – at least for Mycobacterial infections. It is accompanied by 2 tour-de-force applications of the method to benchmark the quality of the data it produces and to exemplify its power. My complements to the authors – all my points below are geared towards clarifying some of the aspects of the study design, as well the results and their interpretation.

We are very grateful for the reviewer's assessment and appreciate the recognition of the work's potential to change the understanding of drug tolerance and clinical practice. The reviewer's suggestions have helped to clarify key aspect of our results and interpretation, and have clearly strengthened the manuscript.

Main points

1. One of the main points of the study (in abstract – lines 7-9, but also in intro, main text and discussion) is that killing dynamics predict better infection outcomes when compared to growth inhibition. However authors show that killing dynamics of exponentially growing cells predict equally “bad” Mtb infection outcomes as growth inhibition. So rather than killing vs growth inhibition, it's the state of the cells that recapitulates better the infection state and hence antibiotic efficacy during infection. I find this is an important distinction that needs to be put more forward. This is particularly important since for other pathogens dormant/not growing states may be less central to their treatment as compared to Mtb →(see also point 4 about this).

Ad 2.1) The reviewer raises an important point that in *M. tuberculosis*, killing dynamics predicted treatment outcomes only when assessed in starvation conditions, but not following normal growth. Therefore, killing dynamics and whether these dynamics are predictive of *in vivo* outcomes is context-dependent (i.e. depends on the physiological conditions under which killing is tested). We have now clarified this distinction. Rather than highlighting growth inhibition versus killing, we now emphasise that the physiological state of bacteria is critical for outcome prediction. In the revised manuscript we also mention that this relationship and the relevance of drug tolerance in general may differ across pathogens.

2. The new method gives an impressive alternative in terms to throughput and information to cumbersome CFU measurements for assessment of killing kinetics, allowing for use of single-cell data. Nevertheless, the manuscript would benefit from expand the discussion on some of the caveats and areas of improvement/expansion of the new method:

Ad 2.2) In the revised manuscript, we now discuss the limitations of ASCT in detail, particularly those related to PI-based viability assessments and the absence host-related infection environments. We also outline potential future strategies to improve ASCT, including adaptations to other bacterial species, incorporation of alternative viability markers, and expansion to real-time, large-scale assessments of single-cell behaviours.

(i) other high-throughput assays have been used to assess killing – e.g. resazurin in bulk assays, or cytological profiling in image-based assays. All these assays may have caveats compared to current method, but should be mentioned/discussed here.

Ad 2.2i) A large number of viability markers have been suggested, each with strengths and limitations. These include bulk methods such as CFU and resazurin-based assays, as well as imaging-based techniques like cytological profiling. In the revised manuscript we now address ASCT in the context of other large-scale methods. As also suggested by other reviewers, we have now additionally assessed all *M. tuberculosis* drug regimens using resazurin (see comment 3.14A).

(ii) authors apply the new method exclusively to mycobacteria, but do note that it is agnostic to species tested. This is an interesting point that deserves more discussion – e.g. how much adjustment is needed for the automated cell tracking and PI ability to capture dead cells? does faster growth limit the time-scale that killing kinetics can be followed (suppressors being a bigger problem)? would other species be measured with similar killing kinetics?

Ad 2.2ii) While our study focused on mycobacteria, ASCT is in principal agnostic to the bacterial species tested. However, applying ASCT to most other species requires adjustments in experimental preparation, image acquisition and/or analysis. For example, faster growth and killing dynamics requires shorter imaging intervals; differences in cell shape requires retraining segmentation models; motile bacteria may need alternative immobilisation methods (e.g. higher agar concentrations) and tracking strategies. The performance of PI may also depend on species-specific membrane and cell wall properties and death modes. Nonetheless, preliminary results in *E. coli* and *P. aeruginosa* suggest that ASCT can be successfully adapted to other bacterial species without major modifications. We have added a brief discussion of the adaptation of ASCT to other species in the revised manuscript.

Suppressors can be captured, but will quickly lead to bacterial overgrowth and therefore do not allow to quantify time-kill kinetics of the remaining cells (more on bacterial growth in 2.5ii).

(iii) authors acknowledge caveats of the PI stain, but they seem to underplay the fact that only 1-11% of PI-negative cells regrow in antibiotic-free media both in results and discussion. Are there caveats of how they measure regrowth? Are there alternatives/orthogonal measures to PI? These are points that all need to be discussed more extensively.

Ad 2.2iii) The reviewer raises an important point which was also noted by other reviewers. While our initial focus was more on the *in vivo* relevance of our viability measure (i.e. its predictive value), we have now included additional experiments to validate PI against alternative readouts.

- Regrowth: We have previously shown that PI-positive bacteria do not resume growth after antibiotic removal, indicating that PI-positive mycobacteria are truly dead.
- GFP-induction: The state of PI-negative bacteria was less clear, since only 1-11% regrew in antibiotic-free media. To address this, we used a reporter strain in which GFP expression can be transcriptionally induced. We treated this *M. abscessus* strain with antibiotics, washed the antibiotics off and then induced GFP expression. After treatments with cefoxitin and moxifloxacin most PI-negative cells were able to newly express GFP, indicating that most of the PI-negative were alive (a similar pattern observed for amikacin later after antibiotic removal). In contrast, only a small subset of these cells regrew in antibiotic-free media, suggesting that regrowth-based assays severely underestimate survival. We also observed continued or even accelerated killing after antibiotic removal, explaining part of the discrepancy between real-time PI-based assessments and delayed regrowth readouts (see also comments 1.12 for more detail).

- Resazurin: In addition, as a bulk viability alternative we performed resazurin assays. These correlated with ASCT-based viability measures in starved *M. tuberculosis* and like ASCT, showed that some starvation conditions were associated with *in vivo* outcomes. However, while resazurin showed mostly trends towards more killing in better-than-SOC treatments, these associations were more consistent and robust in ASCT-based viability assessments (more in comment 3.14A)

(iv) comparison to CFU measures (ED4) is unfair, as CFUs are measured in 2 time-points and compared to continuous and more robust measurements (AUCs) using PI. To do this in a fairer way, the authors should perform a sensitivity analysis with the PI data, taking single-time points from the ASCT curve and testing their predictive power for *in vivo* outcomes of antibiotic regimens – as they do for CFUs in ED Fig 4c.

Ad 2.2iv) We do not think that comparing the 29 imaging frames of ASCT with the two time-points of CFUs is “unfair”, but rather a technological advancement that allows assessing time-kill kinetics at much higher temporal resolution. Nonetheless, the reviewer raises an important point, whether improved prediction of individual patient outcomes is due to the underlying strategy (i.e. PI-based single-cell assessments of viability) or from its more frequent sampling. To address this, we performed the suggested sensitivity analysis using ASCT data restricted to live-cell fractions at day 3 and day 7, matching CFU sampling. Even with this limited resolution, ASCT-derived live-cell fractions are associated with clinical outcomes. Specifically, 11 out of 12 associations were statistically significant at both time points, identical to what was observed when quantifying AUCs. This indicates that ASCT’s predictive performance is not dependent on high-frequency sampling. Due to figure space limitations, we have included the results of this analysis (p-value of associations) in Extended Data.

Figure: Day 3 live-cell fractions of *M. tuberculosis* under nutrient-rich (growing) and nutrient-starved conditions assessed by ASCT across 65 drug regimens. Associations with *in vivo* treatment outcomes are shown. Analyses are identical to those presented in Fig. 2c.

Figure: Day 7 live-cell fractions of *M. tuberculosis* under nutrient-rich (growing) and nutrient-starved conditions assessed by ASCT across 65 drug regimens. Associations with *in vivo* treatment outcomes are shown. Analyses are identical to those presented in Fig. 2c.

	3-day live-cell fractions				7-day live-cell fractions			
	RMM	BMM	gBMM	Clinical	RMM	BMM	gBMM	Clinical
H37Ra log phase	0.018	0.023	1	0.23	0.048	0.055	0.8	0.9
mc2700 log phase	0.9	1	0.06	0.7	0.9	0.7	0.19	0.14
H37Ra PBS starved	0.0013	0.015	0.13	0.043	0.0008	0.0043	0.099	0.029
mc2700 PBS starved	0.0010	0.0035	0.0047	0.020	0.0017	0.011	0.0013	0.020
mc2700 PA starved	0.0026	0.0070	0.028	0.0080	0.0026	0.010	0.040	0.020

Table: Sensitivity analysis using ASCT-based live-cell fractions at day 3 and 7 after antibiotic exposure. Statistical differences were assessed using a two-sided Mann-Whitney U test; corresponding P values are shown.

3. MIC-measures (mean, median, lowest) are not good predictors of *in vivo* outcomes of antibiotic regimens in Mtb (ED Fig. 5). However since this is to predict the efficacy of combinatorial treatments, the authors may want to test how interaction metrics (FICI, Bliss scores) perform.

Ad 2.3) The reviewer is correct that our initial analysis focused on measures of drug activity of single drugs and did not account for potential drug-drug interactions. Quantifying high-order interactions is technically challenging, and to our knowledge no generally accepted strategy exists. Nevertheless, to address this we measured all pairwise interactions among the drugs used in the 65 *M. tuberculosis* regimens. We combined 13 x 12 drugs, and tested each across 9 x 9 concentrations in quadruplicate (in total ~25,000 assays). From these pairwise data we predicted high-order drug effects and compared them with Bliss expectations derived from single-drug effects^{17,18}. Interactions were generally weak, and often antagonistic, with Bliss scores mostly between -0.05 and 0.05, resulting in minimal impact on MICs. Interaction metrics alone were not associated with *M. tuberculosis in vivo* outcomes, except for clinical trial data where antagonism correlated with better-than-SOC classifications. Considering that interactions

modulate rather than reflect absolute drug activity, we combined MIC and interaction metrics in logistic regression models. This showed moderate predictive performance in the relapsing but not the bactericidal mouse model.

Extended Data Fig. 7

Figure. (a) Pairwise *M. tuberculosis* drug-drug interactions of 13 x 13 drugs, each performed in 9x9 checkerboards, highlighting expected and observed growth inhibition. Pairwise interactions are quantified by the Loewe interaction score. (b) Measured pairwise and predicted high-order (derived from pairwise data) drug interactions of 52 *M. tuberculosis* drug regimens. B indicates bedaquiline, C clofazimine, D delamanid, E ethambutol, H isoniazid, L linezolid, M moxifloxacin, Pa pretomanid, Z pyrazinamide, R rifampicin, P rifapentine, Sq SQ109 and Su sutezolid. (c) Drug interactions were compared across similar-to-SOC versus better-than-SOC classifications, based on outcomes in relapsing mouse models (RMM), bactericidal mouse models (BMM) of common mouse strains, the granulomatous C3HeB/FeJ strain and clinical studies^{5,9}. Comparisons were performed using a two-sided Mann-Whitney U test (P values indicated). (d) The performance of combined MIC (median regimen MIC) and drug interaction measures for predicting in vivo outcomes was assessed using logistic regression models. Fifty ROC curves were generated for every condition by randomly selecting 80% of the samples. The black line represents the mean receiver operating characteristic (ROC) curve, the shaded area one standard deviation. Mean area under the ROC curves and 95% confidence intervals are presented.

4. The motivation to move from *Mtb* to *M. abscessus* experiments is well-explained – latter allows for use of large clinical isolate collections (since biosafety level 2), and thereby opens the path for genotype associations to tolerance. However, the choices of the drugs and the assays are less well motivated. Is there a reason that these 8 antibiotics were tested – rather than newer *Mtb*-specific drugs with different mode-of-actions and more activity to dormant cells?

Ad 2.4i) *M. abscessus* is intrinsically resistant to all first-line drugs of *M. tuberculosis*, which are therefore not used clinically for *M. abscessus*. Since our main goal was to evaluate if *in vitro* killing predicts clinical infection outcomes, which requires that patients were treated with these drugs or at least drug classes, we focused on antibiotics that were used in treatment. The eight selected antibiotics are commonly used, recommended in *M. abscessus* treatments, and represent a range of mechanisms of action. While newer drugs such as bedaquiline are of great interest, they were not included in this study because they were not part of the treatment regimens of the patients in our cohort. We have previously mentioned the rationale for our drug selection in the methods section, but have now clarified it in the main text of the revised manuscript.

Even more importantly, is there a reason that tolerance is probed in exponentially growing *M. abscessus*, when the authors have just shown that for *Mtb* starved cells have much better prediction power of clinical outcomes? At the very least this latter part should be better acknowledged and discussed in manuscript.

Ad 2.4ii) The reviewer raises an important point regarding our tolerance profiling in *M. abscessus* clinical isolates following growth, in contrast to the starvation conditions that strongly predicted outcomes in *M. tuberculosis*. There were several reasons for this:

- *M. tuberculosis* and *M. abscessus* experiments were conducted in parallel, and the superior predictive strength of starvation conditions in *M. tuberculosis* only became apparent after large parts of the *M. abscessus* data had already been acquired.
- Preparing single-cell suspensions in mycobacteria is technically challenging due to clumping and cord formation. While this is manageable in nutrient-rich media, clumping becomes extensive in starvation conditions, especially in rough *M. abscessus* clinical isolates. Although we were able to adapt the protocols for the three starvation models in *M. tuberculosis* (mainly by preparing much larger cultures) applying this workflow to over 400 *M. abscessus* isolates was not feasible.

- We assessed all *M. abscessus* isolates simultaneously within a single plate. This was only possible by preparing single-cell suspensions (for some isolates we had to repeat this step several times to achieve sufficient cell numbers) and freezing them in advance. After thawing, agar immobilisation and a 30-minute equilibration, bacteria were physiologically most likely in a lag phase rather than in active exponential growth when the drugs were added.
- Finally, the two conditions tested in *M. tuberculosis* represent physiological extremes rather than authentic infection environments. The same is true for *M. abscessus*. Their predictive value nonetheless suggests that it captured critical features of infection and antibiotic treatment. Extending this approach to more infection-relevant conditions is an important future direction, which may further enhance predictive performance.

5. The authors make a point that correlations between *M. abscessus* kill curves of exponential cells (as measured by ASCT) and growth dynamics (Fig. 3c,d; ED Fig. 6) are largely absent. This goes against current knowledge, and I would be careful for strong statements here, as they may be confounded by technical reasons.

Ad 2.5i) We agree with the reviewer that the weak correlations between ASCT-based tolerance phenotypes and growth dynamics (growth rate and lag time) were unexpected and are not in line with prior studies that strongly link slow growth with increased tolerance. We now acknowledge in the manuscript that technical factors may contribute to these findings, including potential inaccuracies in growth fitting and especially the use of OD measurements, which may be particularly affected by the clumping characteristics of *M. abscessus* isolates. Therefore, we have also more cautiously interpreted these findings.

First, authors surprisingly choose to measure growth by OD in batch culture – rather than using their ability to track single cells, so they can measure single cell growth. This would have had several advantages: same set-up; additional single-cell readouts to growth (cell morphology metrics) and ability to even compare growth and tolerance of same cell if tracking was done continuously (so correlations can be built by comparing thousands of cells directly rather than comparing bulk measurements to AUCs of pooled-single cell data).

Ad 2.5ii) Indeed, correlating single-cell dynamics and antibiotic killing in the same setup could offer more robust and informative insights. However, in the current implementation of ASCT, long-term tracking of growing mycobacteria is technically constrained. While ASCT enables continuous tracking for 2-3 generations, growing mycobacteria eventually form microcolonies with overlapping cells, which cannot be resolved with our high-speed image acquisition setup (widefield, no z-stack acquisition, 40x objective). Moreover, the small well area (~1.5 x 1.5 mm) does not allow prolonged growth assessments. As a result, single-cell tracking becomes unreliable beyond early growth phases. To quantify overall bacterial replication, we therefore used the total object area. This approach allowed us to distinguish and exclude growing isolates in downstream analyses. Notably, in other species like *P. aeruginosa*, we observed extended 2D growth between the gel and plate surface, enabling longer single-cell tracking during growth. Similar growth patterns may be achievable by tuning hydrogel properties to force mycobacterial growth within a single focal plane. Moreover, automatic tracking of growing cells and their phylogenies was not the focus of this study. Overall, the current strength of ASCT is its high-throughput tracking of non-growing or only transiently dividing cells. Extending single-cell growth tracking over multiple generations will require further experimental and analytical optimisation. We now clarify these points in the manuscript.

Second, the measures they extract from growth curve are somewhat questionable. What they measure as lag phase is “apparent lag phase”, a composite of true lag phase and growth rate below the level of detection (if they start from 10^5 cells they have several doublings below the level of detection). Hence

no surprise that 2 metrics (growth rate and lag phase) strongly correlate. They can use the growth rate and initial inoculum to deduce the true lag phase.

Ad 2.5iii) We agree that measuring the lag time as the “time to detectable OD” is highly flawed. To avoid this, we fitted a Gompertz growth model that incorporates the initial inoculum (N_0), growth rate (C) and carrying capacity (A):

$$\lambda = -\ln\left(\ln\left(\frac{A}{N_0}\right)/B\right)/C$$

This approach provides a model-based estimate of true lag time, derived from the known bacterial inoculum and the fitted growth curve, rather than an “apparent lag” from detection limits. Importantly, in our data longer lag times correlated with fast growth, which is the opposite of what would be expected if lag times were determined by undetected early growth.

Moreover, it’s unclear how good are the Gompertz fits of their curves, how long do the different strains grow with a constant growth rate, and at which point of this growth curve have the isolates been treated with the antibiotic.

Ad 2.5iv) We compared three different models to quantify growth and lag times:

- Logistic equation: $N(t) = \frac{K}{\left(1 + \frac{K}{K-N_0}e^{-r(t-T)}\right)}$
- Gompertz equation: $N(t) = Ae^{-Be^{-Ct}}$
- Hybrid Logistic-Monod equation: $t = \frac{Y_K+K}{rK^2} \ln\left(\frac{(K-N_0)N(t)}{N_0(K-N(t))}\right) + \frac{Y_K}{rK} \left(\frac{1}{K-N(t)} - \frac{1}{K-N_0}\right) + T$
where Y_K is the biomass yield coefficient, N_0 is the initial inoculum and $N(t)$ is the population at time t .

To quantify goodness of fit, we calculated the residual sum of squares (RSS) between predicted and observed OD values across 226 clinical isolates. The RSS was 0.71 for the logistic model, 1.10 for the Gompertz model and 2.13 for the hybrid logistic-Monod model. While the logistic model had the lowest RSS, its lag times were sometimes negative and not reproducible. Therefore, we selected the Gompertz model, which provided robust data and interpretable lag times.

As a sigmoid function, the Gompertz equation does not assume a constant growth rate at any interval. Growth phenotypes were assessed in bulk cultures (OD measurements) without antibiotics using, while antibiotic tolerance was evaluated separately by single-cell imaging. As discussed in comment 2.4ii) *M. abscessus* clinical isolates were in lag phase when exposed to antibiotics, while *M. tuberculosis* was treated either in growing (we aimed for mid-log) or starvation conditions.

6. Genetic nature of tolerance and specificity to drug class. Both of these aspects are exciting, but in both cases, as the authors acknowledge, there have been prior reports. What is even more exciting are the GWAS associations for the drug tolerance phenotypes, which yield strong association to 43 genes. The authors should show all genes in Fig 6b, rather than just the one they follow-up, and provide their functional groupings (to back-up the claim that they are enriched in functional groups related to antibiotic target – lines 192-193).

Ad 2.6) We have now included all genes in Fig. 6b. Since most of these genes are poorly characterised (which is common for *M. abscessus*), we did not perform functional enrichment analysis. Therefore, we cannot claim that tolerance-associated genes are enriched in specific functional groups. The sentence “The top genes were consistently enriched within antibiotic groups targeting protein synthesis, DNA or the cell wall.” might have been misleading and was consequently changed.

In terms of follow up, I understand that identifying the mechanism of the phage tail protein in protein synthesis killing may go beyond the scope of this study, but some discussion on this topic would be helpful.

Ad 2.7) Unfortunately, dissecting the exact mechanism of how MAB_0233 modulates killing goes beyond the scope of the present study, especially as there are many additional candidate genes that warrant mechanistic investigation. Nevertheless, we have now included a brief discussion of possible explanations in the revised manuscript.

Minor points

1. lines 5--7: it's important to specify that bacteria = mycobacteria.

Ad 2.min1) We have revised this text.

2. lines 11-12: "drug tolerance is conserved across similar drug targets" – consider revising as it's unclear what you mean. Tolerance patterns, mechanisms or kinetics are similar for drug with same target?

Ad 2.min2) We have revised this text.

3. lines 25-26: some mechanisms of drug tolerance are known - both in vitro and in infections. One could cite dozens of papers.

Ad 2.min3) We agree with the reviewer that many mechanisms underlying drug tolerance have been described. However, compared to resistance, our mechanistic understanding of drug tolerance in human infections is poor. We have revised this sentence.

4. line 53: isn't it better to say that this is tested in 1536-plates? More than 1,000 conditions sounds quite random.

Ad 2.min4) We have revised this text.

6. Fig 1c – this cannot be the time-kill curve of Tigecycline, as it directly contradicts traces shown in Fig. 1d. Please also specify which clinical isolate of *M. abscessus* is profiled.

Ad 2.min6) The previous figure 1c and 1d showed different clinical isolates with different tigecycline time-kill kinetics. We understand that this is confusing. Therefore, we now changed the figures 1c-e, in a way that they are derived from the same clinical isolate (Isolate 328).

7. lines 89-91: why are those three drugs expected to act better on starved cells? It's not straightforward to explain based on their target/mode-of-action.

Ad 2.min7) We agree that the enhanced killing of starved *M. tuberculosis* by bedaquiline, clofazimine and pretomanid may not be immediately intuitive from their known targets, especially for clofazimine where the precise target is not known. Pretomanid, which is a prodrug, that once activated releases nitric oxide and disrupts the respiratory chain, a process that is particularly harmful to non-replicating bacteria relying on minimal respiration. Bedaquiline inhibits ATP synthase, and clofazimine interferes with electron transport and redox balance. These mechanisms are especially effective in metabolically inactive cells. But regardless of their exact targets, all three drugs have shown to have substantial activity under non-replicating or nutrient-starved conditions, as shown in the referenced articles.

8. lines 127-129 – it may be good to mention how much antibiotic concentration (and it's relation to MIC) is expected to affect killing kinetics.

Ad 2.min8) This is an interesting question that could be readily be addressed with ASCT. But in this project we did not systematically explore a wide range of antibiotic concentrations. In general, killing is assumed to increase once drug levels exceed the MIC and to plateau at high concentrations. Here, we analysed only two relatively high concentrations (10x and 20x the MIC of the reference strain), which generated similar and strongly correlated overall kill kinetics (Table). These findings indicate that in our conditions, time-kill kinetics were predominantly shaped by strain-intrinsic tolerance phenotypes, while drug concentrations had a smaller role.

AMK	AZM	FOX	IPM	MIN	MXF	LIN	TGC
0.99	0.97	0.97	0.89	0.87	0.83	0.97	0.97

Table: Pearson correlation coefficient between killing (AUCs) of *M. abscessus* isolates exposed to high and low drug concentrations

9. lines 152-153 – many more papers beyond mycobacteria, and I would find important to cite PMID 25043002 and 28183996, which have been seminal in this regard.

Ad 2.min9) We agree with the reviewer and have now included both suggested references at this position (the first reference was previously cited elsewhere in the manuscript and is now also cited here).

10. lines 158-159: why resistance of unstable drugs (unstable in what sense? chemically unstable?) or drugs with less variable MIC are expected to be less heritable?

Ad 2.min10) We apologise for the ambiguous wording. We have clarified that drugs with lower chemical stability tended to show lower heritability estimates, likely due to lower reproducibility. In addition, when MIC variability is small, a substantial proportion of the observed variation may reflect experimental noise (a single log₂ step is according to clinical microbiology guidelines acceptable). Therefore, minor but truly genetically conferred phenotypic changes are hard to detect, whereas high-level resistance mechanisms are easier to capture (see also comment 1.15).

11. lines 168-170 (and lines 242-244): increased resistance in some antibiotics resulting as a trade-off to decreased tolerance to other antibiotics, is reminiscent of collateral sensitivity and opens the path for combinatorial use. Maybe the link can be made in discussion.

Ad 2.min11) The reviewer raises an interesting question, whether high tolerance to one drug could simultaneously lead to low-tolerance or low MICs to another, creating exploitable vulnerabilities. We observed that some isolates with high-level aminoglycoside or macrolide resistance also had a low

tigecycline tolerance. However, this was not consistent across all high-level resistance mutations, indicating different underlying mechanisms. Though, such patterns could still favour selection of low-tolerance mutants. We now include a summary figure of all resistance-tolerance correlations (Extended Data Fig. 7), showing several weak positive and negative associations that may reflect cross-resistance/tolerance or collateral sensitivity, respectively. But given the large genetic diversity of our clinical isolates, our dataset is not optimal for such analysis. Optimally, such associations should be explored using experimental evolution or mutagenesis in a uniform genetic background. ASCT could be used in such endeavour, for example to systematically profile drug tolerance alongside resistance in an arrayed mutant library.

12. Fig 5a: please provide n for box plots. If same as in ED. Fig 8 (46 + 89), then please correct total in the legend (135, not 136).

Ad 2.min12) As suggested by reviewer 4 and addressed in comment 4.12, we have added individual data points to the box plots. We have now also included sample numbers, which are 135 for all MICs (we apologise for the mistake). For drug tolerance, growing isolates were excluded, which particularly reduced the numbers for azithromycin, since most *M. abscessus* isolates harbour an inducible macrolide resistance.

13. lines 180-183 and Fig 5B: it's unclear why data from tolerance on imipenem is combined with resistance to macrolides to benchmark predictions. Why this combination of tolerance and resistance vs any other?

Ad 2.min13) Macrolide MICs were the only resistance phenotypes associated with clinical outcomes, which is why no other MICs were included. Among tolerance measures, imipenem showed one of the strongest associations with patient outcomes and was thus selected as a representative tolerance

measure. To clarify this point, we now include the AUCs of all MIC-tolerance combinations in the Extended Data Fig. 8.

	AMK toleran. (AUC)	AZM tolerance (AUC)	FOX tolerance (AUC)	IPM tolerance (AUC)	LZD tolerance (AUC)	MIN tolerance (AUC)	MXF tolerance (AUC)	TGC tolerance (AUC)	Resistance - single feature
AMK resistance (MIC)	0.643 (0.617, 0.670)	0.640 (0.537, 0.743)	0.663 (0.644, 0.682)	0.696 (0.675, 0.717)	0.638 (0.615, 0.661)	0.631 (0.581, 0.682)	0.506 (0.450, 0.562)	0.636 (0.605, 0.666)	0.525 (0.503, 0.547)
AZM resistance (MIC)	0.670 (0.641, 0.699)	0.589 (0.516, 0.661)	0.684 (0.663, 0.705)	0.731 (0.709, 0.753)	0.663 (0.639, 0.686)	0.696 (0.665, 0.726)	0.603 (0.562, 0.644)	0.680 (0.650, 0.710)	0.642 (0.618, 0.666)
CLR resistance (MIC)	0.737 (0.709, 0.766)	0.661 (0.584, 0.739)	0.730 (0.704, 0.755)	0.775 (0.757, 0.794)	0.727 (0.700, 0.754)	0.747 (0.717, 0.776)	0.680 (0.649, 0.712)	0.732 (0.710, 0.754)	0.692 (0.668, 0.715)
FOX resistance (MIC)	0.649 (0.624, 0.674)	0.636 (0.541, 0.731)	0.684 (0.658, 0.710)	0.702 (0.676, 0.727)	0.595 (0.555, 0.636)	0.600 (0.554, 0.646)	0.532 (0.491, 0.574)	0.582 (0.535, 0.628)	0.539 (0.516, 0.562)
IPM resistance (MIC)	0.647 (0.621, 0.672)	0.629 (0.560, 0.698)	0.698 (0.672, 0.725)	0.697 (0.672, 0.721)	0.603 (0.561, 0.645)	0.630 (0.589, 0.671)	0.565 (0.528, 0.602)	0.632 (0.592, 0.672)	0.580 (0.554, 0.606)
LZD resistance (MIC)	0.649 (0.622, 0.676)	0.621 (0.518, 0.724)	0.643 (0.621, 0.664)	0.672 (0.649, 0.695)	0.628 (0.601, 0.654)	0.561 (0.519, 0.603)	0.545 (0.488, 0.601)	0.606 (0.580, 0.631)	0.517 (0.503, 0.532)
MIN resistance (MIC)	0.674 (0.648, 0.700)	0.648 (0.590, 0.706)	0.663 (0.637, 0.689)	0.700 (0.675, 0.725)	0.625 (0.594, 0.656)	0.592 (0.549, 0.636)	0.521 (0.468, 0.575)	0.631 (0.608, 0.653)	0.529 (0.508, 0.549)
MXF resistance (MIC)	0.645 (0.621, 0.669)	0.764 (0.729, 0.799)	0.681 (0.655, 0.707)	0.705 (0.674, 0.736)	0.582 (0.546, 0.618)	0.538 (0.470, 0.607)	0.527 (0.479, 0.575)	0.564 (0.519, 0.608)	0.517 (0.487, 0.546)
TGC resistance (MIC)	0.667 (0.644, 0.690)	0.633 (0.562, 0.705)	0.640 (0.614, 0.667)	0.681 (0.656, 0.705)	0.608 (0.581, 0.635)	0.610 (0.551, 0.669)	0.610 (0.585, 0.635)	0.589 (0.551, 0.628)	0.542 (0.519, 0.566)
Tolerance - single feature	0.652 (0.630, 0.675)	0.540 (0.472, 0.608)	0.650 (0.627, 0.673)	0.677 (0.658, 0.697)	0.610 (0.584, 0.636)	0.558 (0.528, 0.589)	0.510 (0.481, 0.540)	0.612 (0.586, 0.637)	

14. line 198: most of the protein synthesis inhibitors tested here are not known to cause mistranslation, and only inhibit translation.

Ad 2.min14) We agree that most protein synthesis inhibitors included here primarily inhibit translation and are not classically associated with mistranslation. We speculated that MAB_0233 could influence drug accumulation, or for certain drugs such as aminoglycosides, modulate drug-induced mistranslation. But as the reviewer points out this would not explain the tolerance mechanisms for all drugs. As suggested in comment 2.7, we have expanded the discussion of potential mechanisms underlying these tolerance phenotypes.

15. line 201; this is not entirely true for linezolid (complemented mutant has also a significantly different time-kill curve compared to wildtype too).

Ad 2.min15) The reviewer is right that our sentence “*This effect on killing was lost following MAB_0233 complementation*” is not completely correct. While we show increased tolerance to amikacin, tigecycline and linezolid in the MAB_0233 knockout strain, complementation fully restored the low-tolerance phenotype only for amikacin and tigecycline, but not for linezolid. The complemented strain still had an altered time-kill profile compared to the wild-type for linezolid, suggesting either partial complementation or the involvement of additional factors specifically affecting linezolid tolerance. This was clarified in the revised manuscript.

16. lines 231-233: this has been challenged before. See for example PMID: 34646011 - there is shown that antibiotics can change from bacteriostatic action to rapid killing depending on the species and stain tested.

Ad 2.min16) The traditional classification of antibiotics as either bactericidal or bacteriostatic, mostly based on the study of a few strains in nutrient-rich media, fails to capture the true complexity of antibiotic activity. Notably, studies have also failed to show that the use of bactericidal drugs translates into improved clinical outcomes^{10,19}. Nevertheless, the prevailing view especially in medicine remains that the drug itself is the primary, if not only, determinant of bacterial killing. However, our findings, and those of others challenge this assumption. The study cited by the reviewer nicely shows that the same drug can lead to massively different kill-kinetics across different bacterial species. Our data now show that such variability in killing extends even to isolates within the same bacterial species.

Reviewer #3 (Remarks to the Author):

In the present report, Jovanovic and collaborators report an exciting and novel antimicrobial single cell testing to overcome historical limitations of measuring susceptibility in bulk bacterial populations. This is an impressive tour de force new method that have the potential to significantly improve our understanding of how antibiotics work and clinical decision-making. The quality of the data and the analysis is very high.

Ad 3.1) We greatly appreciate the thorough evaluation, positive feedback and suggestions to improve our work.

There are, however, some aspects that need to be considered to strength the message of the manuscript (and the approach). I have some suggestions for improving the manuscript: The cell viability is measured by propidium iodide accumulation after penetrating the damaged bacterial cell wall. Imipenem and cefoxitin, which target the cell wall stands out as the more effective drugs against both *M. tuberculosis* (Figure 1d) and *M. abscessus* (Figure 3b). Could antibiotics targeting cell wall facilitate PI penetration, possibly leading to overestimation of death rates in those conditions? On the other hand, Is possible an underreporting of cell death by non-membrane-disrupting drugs? (increasing the chance of false negative)

Ad 3.2) The reviewer raises an important point – that cell wall targeting antibiotics could enhance membrane permeability and thus facilitate propidium iodide (PI) entry, while antibiotics that do not directly compromise the cell envelope might underestimate killing.

- Following beta lactam exposure and antibiotic washout, PI-positive bacteria did not regrow, indicating that these cells are truly dead. In contrast, nearly all PI-negative bacteria could induce GFP expression, indicating an intact transcription and translation machinery and thus viability. These new data on GFP induction are now included in the main manuscript (Fig. 1) and Extended Data Fig. 2 (discussed in more detail in comment 3.14).
- Non-cell wall targeting drugs can also induce rapid PI uptake. However, uptake typically occurs with a ~3 hour delay compared to beta lactams. Given the long duration of our experiments (up to 168 hours), this modest lag is unlikely to affect overall conclusions.
- In *M. abscessus*, beta-lactams are not uniformly the most bactericidal drugs. In some clinical isolates, amikacin, moxifloxacin or tigecycline generated stronger killing than cefoxitin or imipenem.
- In *M. tuberculosis*, PI-positive fractions were not higher with cell-wall targeting drugs compared to others classes (Figure). Figure 1d mentioned by the reviewer shows time kill-kinetics of *M. abscessus*.

Figure: Area under the 7 day time-kill curve of cell-wall/membrane targeting drugs (clofazimine, delamanid, ethambutol, isoniazid, pretomanid, SQ109) and non-cell wall targeting drugs (bedaquiline, linezolid, moxifloxacin, rifapentine, rifampicin, sutezolid, pyrazinamide) in growing and starved *M. tuberculosis*.

- PI-based viability dynamics of cell wall targeting and non-cell wall targeting drugs correlate well with *in vivo* outcomes in *M. tuberculosis* and *M. abscessus*, supporting the biological relevance of our measurements.

Overall, while membrane integrity loss likely occurs with slightly different kinetics depending on the antibiotic mechanism, our data indicate that PI-based viability assessments are a robust proxy for bacterial death. We now discuss this potential limitation and its implications in the revised manuscript.

The experiments presented in Figure 1D do need an untreated control as it is important to know how the system affects cell viability in general.

Ad 3.3) We agree that it is important to assess whether the ASCT system itself affects bacterial viability. In nutrient rich medium (e.g. 7H9) *M. abscessus* grows rapidly, and due to the size of a single well in the 1536-well plate (~1.5 x 1.5 mm), growth quickly exceeds the well limits. With our imaging approach (40x widefield, no z-stacks) we can follow initial cell divisions but not distinguish individual bacteria once multi-layered growth occurs. Still, by using the total object area we can identify growth and roughly quantify growth kinetics (which we use to identify growing isolates) until the area is fully covered with bacteria (in *M. abscessus* after approximately 1 day). For this reason, our platform is optimised for assessing bacterial killing rather than quantifying detailed growth kinetics.

We now also assessed and included a figure of untreated bacteria immobilised in ASCT, showing that in nutrient-rich conditions 95.2% of cells, and in starvation conditions 90.7% of all cells either grew or remain PI-negative after 48 hours (Figure). Similarly, in starved broth cultures 95.1% of cells were PI-negative cells at 48 hours. Because broth cells were not continuously tracked, the number of dead bacteria may be slightly underestimated. Overall, these results suggest that ASCT immobilisation does not substantially impair bacterial viability.

Figure: Fraction of viable *M. abscessus* immobilised in ASCT under nutrient-rich and starvation conditions (dividing and PI-negative cells were considered alive). Yellow dots indicate the fraction of PI-negative bacteria in starved broth culture at 24 and 48 hours.

There are no SEM values for Linezolid, Imipenem and Cefoxitin. Would it be because there is no variability?

Ad 3.4) In Figure 1d we present time-kill curves as mean \pm SEM of three replicates. For linezolid, imipenem and cefoxitin the SEM is very small, and therefore not visible when plotted with thicker lines.

Below we present the new Fig. 1d with thinner lines, although in some cases the SEM remains smaller than the line thickness.

I understand the authors do not want to assess single-cell heterogeneity in this manuscript as mentioned in the main text. However, it would be interesting to assess the variability between antibiotic at the bulk level as well as the temporal level. For example, if we consider Amikacin or Moxifloxacin, it seems that the variability (here shown by the SEM) is higher in later timepoints. Could it suggest that within experiments, early-death (24h) mediated by ATBs is less variable than late-death? Could the authors comment on this.

Ad 3.5) We have now analysed the variability of *M. abscessus* time-kill kinetics across antibiotics and over time. Overall variability was low but tended to increase at later time points, particularly for antibiotics with strong killing activity. This suggests that early antibiotic-killing is more consistent across replicates, while later killing may be influenced by additional factors (bacterial numbers, drug stability, etc.). Several other analyses, like the correlation between low and high drug concentrations and the very high heritability estimates for some drugs illustrate the high accuracy, since such findings wouldn't have been possible without high reproducibility.

This type of variability represents population-level heterogeneity, which can also be captured by bulk assays, like traditional time-kill curves. In contrast, single-cell heterogeneity refers to differences between individual bacteria such as morphology, cell behaviours (assessed with dyes or reporter strains), or single-cell growth dynamics. Assessing the trajectories of single-cells, including antibiotic killing, could for example be used to identify characteristics underlying persister cells. While such single-cell analyses are beyond the scope of this manuscript, ASCT is well-suited to allow such studies at large scale in the future.

	3h	6h	9h	12h	24h	36h	48h	60h	72h
AMK-L	0	0	1	1	2	2	3	3	3
AMK-H	0	1	1	1	2	3	4	4	4
AZM-L	0	1	1	1	2	2	3	3	3
AZM-H	0	0	1	1	1	2	2	2	3
FOX-L	6	9	10	13	16	18	19	19	20
FOX-H	7	10	11	12	15	16	17	17	17
IPM-L	6	6	6	7	9	11	12	12	13
IPM-H	6	8	8	9	11	12	12	13	14
LZD-L	0	1	1	1	1	2	2	2	3
LZD-H	0	1	1	1	2	3	3	3	4
MIN-L	0	1	1	1	2	2	2	3	4
MIN-H	0	1	1	2	3	4	5	6	6

MXF-L	2	2	2	2	5	6	7	7	8
MXF-H	1	2	2	4	8	10	10	10	11
TGC-L	1	1	2	2	4	5	5	6	6
TGC-H	1	1	1	2	3	4	4	5	5

Table: Mean coefficient of variation (in %) of all clinical *M. abscessus* isolate triplicates assessed over time, across eight drugs, each at low (L) and high (H) concentration.

Bactericidal and bacteriostatic effects can switch depending on antibiotic concentration. Hence, I suggest reinforcing the Figure 1 to add for one or two antibiotic a growth assessment using different concentrations.

Ad 3.6) We now added a panel to figure 1, highlighting the dose response in regard to antibiotic killing.

The authors state that INH, RIF and EMB are more active against exponentially growing *M. tuberculosis*. I would change the wording, as we know that at the single-cell level *Mtb* seems to have a linear growth (Chung et al. 2024), even though I understand that the authors refer to bulk growth.

Ad 3.7) Considering this work, we have now revised the manuscript and refer to “growth” versus “starved” *M. tuberculosis* conditions, rather than “exponential growth.”

In Figure 2, the manuscript would benefit if the authors can show all the combinations possible of the 6 antibiotics in both rich and starvation media. Moreover, how the authors can exclude that in the comparison between rich and starvation media, the drugs are binding to media components and do not enter the bacterial cell?

Ad 3.8) We assume the reviewer refers to the 6 antibiotics in Extended Data Fig. 2a-b. In the revised figure, we now present the associations of these 6 antibiotics under nutrient rich and starvation conditions. We compared regimens containing a specific drug and found that some were associated with higher overall killing, while a few were also linked to lower overall killing. Though, this does not imply that the drug itself reduces killing. For example, in nutrient rich conditions bedaquiline containing regimens performed on average worse than regimens lacking bedaquiline.

Numerous studies have shown that the physiologic bacterial state strongly influences antibiotic killing 14,20–23. While such physiological differences likely explain most of the observed effects, we cannot exclude that certain drugs may also bind to components of the nutrient-rich or starvation media, contributing to additional killing variability.

Regarding the Figure 2 and the link between the ASCT *in vitro* killing efficacy under bacterial starvation (PBS and pantothenate) and the predicted *in vivo* outcome. Dormant/metabolically inactive phenotypes generated *in vitro* could mirror drug-tolerant bacilli *in vivo*. However, *in vivo* lesions have multiple microenvironments (oxygen tension, pH, lipids) that cannot be addressed by the *in vitro* ASCT starvation assay. The *in vitro* assay uses constant concentrations of antibiotics, which don't reflect dynamic drug exposure that happens *in vivo*. These limitations (as mentioned in the discussion "Furthermore, our study did not specifically mimic complex mycobacterial infection environments, such as intracellular, niches or granulomas, which may alter tolerance phenotypes") may affect the comparison between the *in vitro* results in this work and the *in vivo* (mouse and clinical trials) results from references 28 and 29.

Ad 3.9) We completely agree with the reviewer that our strategy and platform cannot fully explain treatment outcome, since it does not address other critical parameters of bacterial infection and treatment. This includes, but is not limited to diverse host environments, immunity, dynamic drug exposures, genetic factors of the host, etc. . However, showing that our *in vitro* measures correlate with multiple infection outcomes across two species and several drug conditions, suggests that our approach likely captures highly relevant mechanisms of antibiotic treatment, even if it does not fully capture the *in vivo* infection environment. Therefore, we consider it an important advancement from traditional MIC testing.

Moreover, a limitation of the correlation between the *in vitro* data presented and the results from mice/clinical trials relies on the strain of Mtb used. I acknowledge the "tour-de-force" of the method, however, as mentioned by others, antibiotic efficacy can be affected by Mtb localization in cells (Santucci et al, 2021), therefore potentially *in vivo*. This is not discussed in the results section or discussion. The authors should discuss this and acknowledge the limitations of this correlation.

Ad 3.10) We completely agree that antibiotic activity is highly complex and cannot be fully captured by our *in vitro* findings, as it is determined not only by the drug and bacterium, but also by diverse host

environments (and many other factors), which influence pharmacokinetics and pharmacodynamics. As mentioned by the reviewer, the localisation of *M. tuberculosis* in distinct infection niches such as granulomas, intracellular compartments or caseum can strongly influence antibiotic efficacy²⁴. Unfortunately, apart from animal models, which themselves have limitations, there are no experimental models that address all of these factors simultaneously.

So far, most of clinical microbiology and much of drug discovery and development rely primarily on MICs. With our work we aim to move beyond MICs by capturing additional infection-relevant bacterial phenotypes. We acknowledge that our current approach does not model complex infection environments, which we now explicitly mention in the revised manuscript. Importantly, we believe that the value of *in vitro* assays is not determined by their complexity per se (whether it captures all these factors) but by their demonstrated relevance for infection outcomes, something that remains unknown for most *in vitro* models. Our work provides rare evidence of an *in vitro* tool assessing antibiotic efficacy (apart from MICs) that translates into *in vivo* outcomes in mouse models and patients, and ASCT is able to extend to more complex host environments, to further enhance its predictive performance.

In the main text, L119-120, “which probably shares fundamental principles of antibiotic survival with other mycobacteria”, it is a bold sentence, not referring to any study. Also, Mtb and Mab share their genus, but differs through their growth rate. I would tone down this sentence as it can be misleading.

Ad 3.11) We agree that this statement was ambiguous and have now removed it from the manuscript.

One criticism that could be made about the “genetic” bacterial background is that we do not know, within those clinical isolates, if patients have been differentially treated with different antibiotics that could have generate tolerance, unrelated to the genetic background. Could the authors comment on the distribution of the strains?

Ad 3.12) We have assessed drug tolerance across many bacterial *M. abscessus* strains in a single environment, and show that this variability is largely genetically driven. We acknowledge that these assessments do not capture the full complexity of the host environment, including prior treatments, drug combination treatment, or bacterial co-infections. However, despite not modelling this complexity, we find that single drug tolerance phenotypes measured *in vitro* are associated with patient outcomes. This important point is now discussed in the revised manuscript.

In Figure 6, The data regarding MAB_0233 are quite interesting, did the authors looked for similar genes in other mycobacteria (maybe some of them that are more tolerant to those drugs, in clinics)?

Ad 3.13) We identified orthologous genes to MAB_0233 in several other (less known) non-tuberculous mycobacteria, but not in *M. tuberculosis*. Importantly, MAB_0233 represents just one of many tolerance mechanisms in *M. abscessus*, given the larger target space for drug tolerance, compared to drug resistance. Functional validation of additional candidate genes, as well as evaluation of their conservation and relevance across other mycobacterial species is part of our ongoing work.

In order to validate the method, the use of other markers of cell death or metabolic activity (for example resazurin) could be implemented into the Antimicrobial Single-Cell Testing (ASCT) to compare results.

Ad 3.14A) Indeed, a major strength of ASCT is its versatility, allowing rapid integration of different bacterial reporter strains and additional dyes to test various aspects of bacterial physiology, including viability and metabolic activity. Resazurin is a blue non-fluorescent dye, that gets in metabolically active cells enzymatically reduced to the red and highly fluorescent resorufin. Resazurin is widely used to amplify signals of growing bacterial populations (e.g. in the MIC assay), or to assess the viability of bulk populations. However, resorufin does not bind to specific cellular structures and diffuses freely, making it unsuitable for single-cell imaging in ASCT. In the original manuscript we have shown that PI-positive

cells do not grow following antibiotic removal, indicating they are dead. Moreover, we show that PI-based viability assessments are related to *in vivo* outcomes. Now we strengthen our approach with two additional strategies:

- Since resazurin was also mentioned by reviewers 1 and 2 (comments 1.12 and 2.2i), we performed population-level resazurin assays in *M. tuberculosis* and compared the results to ASCT and *in vivo* infection outcomes. As expected for a metabolic readout, starved *M. tuberculosis* mc²7000 showed substantially lower resorufin levels than growing cultures. When testing all 65 *M. tuberculosis* drug regimens, the RFU change (between day 0 and 14) correlated with ASCT-derived viability only under starvation but not when growing cultures were exposed (**Figure**).

Figure: Correlation between resazurin-based killing (change in relative fluorescence units [RFU] from day 0 to 14) and ASCT-based killing of *M. tuberculosis* mc27000 (area under the time-kill curve). 65 drug regimens were tested.

Under starvation we also observed some associations with clinical outcomes, with a general trend towards a greater resorufin decline in better-than-SOC regimens. This trend was similar to the associations observed with ASCT, though mostly not statistically significant.

Figure: Comparisons of resazurin-based killing of *M. tuberculosis* mc27000 (change in RFU from day 0 to 14) across in vivo classifications of drug regimens as similar-to-standard of care (SOC) or better-than-SOC, based on outcomes in relapsing mouse models (RMM), bactericidal mouse models (BMM) of common mouse strains, the granulomatous C3HeB/FeJ strain and clinical studies^{5,9}. Groups were compared using a two-sided Mann-Whitney U test (P indicated). Boxplots show the median, interquartile range and total range. (RMM: n = 46, BMM common strains: n = 48, BMM C3HeB/FeJ: n = 15, clinical bactericidal activity: n = 14).

- As mentioned above, PI-positive cells do not regrow following antibiotic washout, suggesting that PI accumulation is a robust marker of mycobacterial death. However, the viability state of PI-negative cells, especially considering the discrepancy between the number of PI-negative and regrowing cells, is less clear. To address the viability of PI-negative cells we used an *M. abscessus* reporter strain that expresses GFP following transcriptional induction. We treated this *M. abscessus* strain with different antibiotics, removed the antibiotics and then induced GFP expression with anhydrotetracycline (ATc). After cefoxitin or moxifloxacin exposure most of the PI-negative bacteria were able to synthesise GFP, indicating that they are alive. The number of GFP expressing bacteria was much higher than overall bacterial regrowth, suggesting that a lot of bacteria newly expressing GFP are not able to regrow in the given conditions. Moreover, we observed after antibiotic removal substantial additional bacterial death, which explains some of the discrepancy between the high survival of real-time ASCT readouts, and the low survival of delayed CFU measures (discussed in more detail in comment 1.12).

Reviewer #3 (Remarks on code availability):

It works

Ad 3.15) We appreciate testing our code.

Reviewer #4 (Remarks to the Author):

This manuscript addresses a critical issue in the treatment of Mycobacterium infections: the disconnect between in vitro assessments of antibiotic susceptibility and actual in vivo outcomes. The authors develop a microscopy-based method, termed Antimicrobial Single-Cell Testing (ASCT), to measure antibiotic killing dynamics in real time at single-cell resolution, and correlate these measurements with infection outcomes in vivo. They further perform genetic analysis to identify determinants of drug tolerance.

Overall, I find this manuscript to be both timely and of significant potential interest. The authors thoroughly characterize in vitro efficacy and, through statistical analysis, relate it to clinical outcomes. A wide range of antibiotics and mutants are examined in order to identify generalizable features that explain treatment outcomes. This integrative approach has value for the evidence-based design of diagnostics and treatments.

That said, I found some parts of the manuscript difficult to follow. In several instances, explanations are either insufficient or potentially misleading due to a lack of clarity or contextualization. Moreover, the manuscript does not clearly distinguish which findings are novel contributions of this study versus reiterations or confirmations of previously established knowledge.

Ad 4.1) We thank the reviewer for the encouraging feedback and for the suggestions to improve our manuscript. We have accordingly revised the manuscript throughout to improve clarity, provide additional context, and more clearly highlight the novel contributions of this study.

Taking an abstract as an example, the description of ASCT as a “large-scale imaging approach that quantifies bacterial killing in real time at single-cell resolution”. Many recent studies have applied single-cell imaging approaches to monitor antibiotic killing. It would be helpful if the authors clarified what specifically differentiates ASCT from these existing methods—whether it's the throughput, resolution, experimental design, or analytical framework.

Ad 4.2) We agree with the reviewer that single-cell imaging approaches are increasingly applied to study antibiotic activity. Many new strategies focus on rapid species or resistance identification (PhAST, Selux, dRAST, Accelerate Pheno, etc.) while many others use microfluidics (e.g. the CellAsics platform or mother-machine) combined with dyes (live/dead, membrane potential, metabolic activity, etc.) or fluorescent reporter strains quantifying metabolic activity, transcription, translation, stress responses and more.

Our approach differs in several regards:

- Experimental throughput: By combining liquid handling with our plate-based immobilisation strategy, ASCT enables the simultaneous assessment of hundreds of different conditions in 1536-well plates.
- Automated image analysis: We established an automated high-content image acquisition and analysis pipeline that accurately segments, tracks and classifies the viability of millions of single bacteria at ~2-hour resolution. While most published single-cell imaging studies analyse hundreds to thousands of cells, ASCT successfully tracked over 100 million bacteria for 3 or 7 days.
- Benchmarking: There exists a large number tools assessing antibiotic activity. However, for most (especially those unrelated to MICs) the value is unknown. We show in mycobacteria, that ASCT-derived killing dynamics (across drug regimens and across bacterial strains) consistently translate into *in vivo* treatment outcomes in mice and patients.
- Versatility: Although we focus on antibiotic killing dynamics in mycobacteria (bactericidal drug activity and bacterial tolerance), ASCT is adaptable, with applications in basic biology (for studying drug tolerance, hetero-resistance, resistance evolution, etc.) drug discovery and development and potentially clinical diagnostics.

We highlight these unique strengths of ASCT more clearly in the revised manuscript.

Likewise, the statement that “drug-specific killing dynamics – rather than bacterial growth inhibition – predict infection outcomes” is ambiguous. Does this imply that bactericidal antibiotics are more predictive than bacteriostatic ones? Or by bacterial growth inhibition, do authors allude to slow death (tolerance)?

Ad 4.3) We agree that the original phrasing can be misinterpreted. We have now clarified that it is the phenotypes of killing dynamics (rapid vs slow killing) rather than growth related phenotypes (like MICs), that correlate with mycobacterial infection outcomes. The sentence has been changed accordingly.

The authors also state that “antibiotic killing is not merely a drug effect, but a genetically encoded bacterial trait (drug tolerance) distinct from classical drug resistance.” Does this statement try to mean that there are genetic mechanisms for drug tolerance? Indeed, the existence of a genetic base of tolerance is a well-accepted idea in microbiology. I saw the author cited “43. Levin, B. R. & Rozen, D. E. Non-inherited antibiotic resistance. *Nat. Rev. Microbiol.* 4, 556– 367 562 (2006).” to support that drug

tolerance is non-genetic. But, this paper is more than 20 years old. There have been significant advances since then.

Ad 4.4) We agree with the reviewer that genetic determinants of antibiotic tolerance have been reported in many studies across different bacterial species, including mycobacteria. Our statement aimed to strengthen this existing knowledge, as the view that tolerance is a primarily non-heritable, transient phenotypic trait remains common. While previous studies typically focused on single mutants or selected clinical isolates with known tolerance phenotypes, our work systematically assesses drug tolerance across hundreds of unselected patient isolates. This approach provides robust evidence that drug tolerance is genetically encoded on a population level. We have revised this sentence in the manuscript. The reference referring to the historical (traditional) perspective, has been removed.

I agree with the central premise that bacterial tolerance may play an underappreciated role in mycobacterial infections and that MIC values may not correlate well with clinical success. However, I caution against generalizing this finding to all bacterial infections. There is ample evidence in other systems where high MIC values are associated with treatment failure. The manuscript should be more precise in delimiting the scope of its claims.

Ad 4.5) We agree with the reviewer that MICs are the cornerstone of antibiotic susceptibility testing and are clinically indispensable, as exemplified in defining MDR/XDR tuberculosis. The intention of our study is not to diminish the relevance of MICs, but rather to complement them by highlighting bacterial killing dynamics as an additional, orthogonal predictor of treatment outcomes.

In our study, tuberculosis regimens with particularly low MICs did not necessarily outperform others. However, all tested regimens contained at least one active drug, underscoring the role of MICs in identifying these active agents. Similarly, in *M. abscessus*, macrolide resistance was predictive of poor outcomes, confirming the clinical value of resistance testing. High MICs (i.e. bacterial growth at therapeutic drug concentrations) are generally considered strong indicators of likely treatment failure across all bacterial infections. In contrast, drug tolerance may be particularly relevant in infections where standard treatments frequently fail despite low MICs. Our study demonstrates the added value of drug tolerance assessments in mycobacterial infections, but whether this also applies to other pathogens and clinical settings remains to be explored. We have revised the manuscript to more clearly state the essential role of MICs and to specify that our findings are limited to mycobacterial infections (see also response 1.8A to reviewer 1).

The treatment of heritability also warrants more explanation. While a technical paper is cited (ref 48), the meaning of "heritability" in this context remains unclear to me. If the authors are arguing for genetic inheritance of drug tolerance, does it warrant addressing possible confounding factors such as horizontal gene transfer, repeated spontaneous emergence, or clonal spread of strains within communities?

Ad 4.6) In our study, heritability refers to the proportion of phenotypic variation (e.g. variation in drug tolerance across clinical strains) that can be explained by underlying genetic variation. This estimate is derived from linear mixed models that quantify how much of the observed phenotype is attributable to genome-wide genetic differences. These models do not distinguish between different modes of inheritance, such as vertical transmission, horizontal gene transfer and spontaneous mutations, but rather provide a measure of the overall genetic contribution to the trait. We have clarified this definition in the revised manuscript.

Some figures are hard to understand and not clear. I think authors should strengthen figure captions. These are just a few that stood out the most.

As suggested by the reviewer we reviewed all figure captions.

- Fig 1d. What is the sample size of each rep? Similar to Fig. 1c? I vaguely remember in suppl materials that wells with <500 or <1000 cells discarded so I'm assuming sample size is quite large.

Ad 4.7) If by "sample size" the reviewer refers to the number bacteria tracked per well, each analysed well included ≥ 1000 single cells for *M. abscessus* and ≥ 500 single cells for *M. tuberculosis*. These thresholds were set to ensure that time-kill kinetics derived from single-cell data were stable and reproducible at the population level.

Figure 1d shows a representative time-kill curve of a *M. abscessus* isolate (Isolate 328), with a mean of 5999 (± 3340) tracked bacteria per well. The total number of tracked cells per well for all clinical isolates is summarised in Extended Data Fig. 1e, which illustrates the distribution of bacterial counts across the dataset. Each *M. abscessus* isolate-drug pair was tested in triplicate (i.e. in three wells).

- Fig2b. It is difficult to evaluate the graph without standard deviation. The vertical lines are distracting. I am not sure what information the black and white dots show other than drug combinations.

Ad 4.8) Figure 2b shows the average bacterial killing (AUC) across three starvation conditions for each regimen. In response to the reviewer, we have now included error bars to illustrate variability across these conditions. The black and white dots indicate the presence or absence of specific drugs in each regimen, which we believe provides a useful visual summary to help readers quickly interpret which drugs contribute to each regimen. We consider the lines quite important to link AUC values with the respective drug information below, but have now reduced their transparency. However, we are happy to remove them if the reviewer prefers. We have also applied the same strategy to Extended Data Fig. 2.

- Fig 3a. Does color mean anything?

Ad 4.9) Yes, the colours indicate the study site. For the UK and Ireland sites, this is shown on the map. For non-UK centres, the corresponding colours are shown in the figure. This has now been clarified in the figure legend.

- Fig 3e. R^2 value is very low and slopes are small shown in Ext Fig 6. Are these statistically meaningful?

Ad 4.10) Given the relatively large sample size, many of these associations are statistically significant (Figure 3e: significant associations highlighted in red or blue, non-significant associations in grey). Extended Data Figure 9 includes R^2 values and corresponding p-values. However, we agree, that statistically significant associations with very low effect sizes are unlikely biologically meaningful, supporting that drug tolerance is an independent marker of antibiotic activity.

- Fig 4a. Standard error should be added. Growing isolates should be added since they are the true resistant strains. I'm assuming "resistance" and "tolerance" are selected based on MIC and AUC? It should be clarified in legend.

Ad 4.11) Heritability estimates are based on bacterial phenotypes (the distribution of each phenotype across the phylogeny) of all clinical isolates and their corresponding genomic sequences. These estimates are derived from the entire dataset of strains with phenotypic and genomic information, yielding a single heritability value per trait, rather than a distribution from which standard errors could be calculated.

For drug resistance, we included all strains with an MIC value, meaning all isolates showed growth at one or more drug concentrations. In contrast, quantifying drug tolerance requires killing dynamics, which is not possible in isolates that continue to grow during drug exposure. Therefore, growing isolates were

excluded from the heritability analysis of tolerance, as including them (e.g. with no killing) could confound the results by attributing genetic determinants of resistance to tolerance.

The reviewer is correct that resistance and tolerance were quantified with MICs and AUCs, respectively. Since both are continuous traits (we quantified the heritability of these continuous traits) there was no selection (or breakpoints) involved. The phenotypes of resistance and tolerance were already stated on the x-axis and in the second sentence of the figure legend.

- Fig 5a. Is there a reason that individual data (like in Fig. 2cd, which were very helpful) is missing here?

Ad 4.12) In Figures 2c and 2d, we presented individual and summarised data, as the number of conditions was small. Figure 5a includes a larger number of conditions; therefore, individual datapoints become less illustrative and summary statistics gain importance. However, in line with the reviewer's suggestion, we have now included individual data points in the box plots of figure 5a.

References

1. Berg, A. *et al.* Model-Based Meta-Analysis of Relapsing Mouse Model Studies from the Critical Path to Tuberculosis Drug Regimens Initiative Database. *Antimicrob. Agents Chemother.* **66**, e01793-21 (2022).
2. Strydom, N. *et al.* Selection and prioritization of candidate combination regimens for the treatment of tuberculosis. *Sci. Transl. Med.* **17**, eadi4000 (2025).
3. Cevik, M. *et al.* Bedaquiline-pretomanid-moxifloxacin-pyrazinamide for drug-sensitive and drug-resistant pulmonary tuberculosis treatment: a phase 2c, open-label, multicentre, partially randomised controlled trial. *Lancet Infect. Dis.* **24**, 1003–1014 (2024).
4. Tweed, C. D. *et al.* Bedaquiline, moxifloxacin, pretomanid, and pyrazinamide during the first 8 weeks of treatment of patients with drug-susceptible or drug-resistant pulmonary tuberculosis: a multicentre, open-label, partially randomised, phase 2b trial. *Lancet Respir. Med.* **7**, 1048–1058 (2019).
5. Larkins-Ford, J., Degefu, Y. N., Van, N., Sokolov, A. & Aldridge, B. B. Design principles to assemble drug combinations for effective tuberculosis therapy using interpretable pairwise drug response measurements. *Cell Reports Medicine* 100737 (2022) doi:10.1016/j.xcrm.2022.100737.
6. Kwak, N. *et al.* Mycobacterium abscessus pulmonary disease: individual patient data meta-analysis. *Eur Respir J* **54**, 1801991 (2019).
7. Kramnik, I. & Beamer, G. Mouse models of human TB pathology: roles in the analysis of necrosis and the development of host-directed therapies. *Semin. Immunopathol.* **38**, 221–237 (2016).
8. Lanoix, J.-P., Chaisson, R. E. & Nuermberger, E. L. Shortening Tuberculosis Treatment With Fluoroquinolones: Lost in Translation? *Clin. Infect. Dis.* **62**, 484–490 (2016).
9. Larkins-Ford, J. *et al.* Systematic measurement of combination-drug landscapes to predict in vivo treatment outcomes for tuberculosis. *Cell Syst* **12**, 1046-1063.e7 (2021).
10. Nemeth, J., Oesch, G. & Kuster, S. P. Bacteriostatic versus bactericidal antibiotics for patients with serious bacterial infections: systematic review and meta-analysis. *The Journal of antimicrobial chemotherapy* **70**, 382–395 (2015).

11. Ronneau, S., Hill, P. W. & Helaine, S. Antibiotic persistence and tolerance: not just one and the same. *Curr Opin Microbiol* **64**, 76–81 (2021).
12. Gengenbacher, M. & Kaufmann, S. H. E. Mycobacterium tuberculosis: success through dormancy. *FEMS Microbiol. Rev.* **36**, 514–532 (2012).
13. Chengalroyen, M. D. *et al.* Detection and Quantification of Differentially Culturable Tubercle Bacteria in Sputum from Patients with Tuberculosis. *Am. J. Respir. Crit. Care Med.* **194**, 1532–1540 (2016).
14. Fanous, J. *et al.* Limited impact of Salmonella stress and persists on antibiotic clearance. *Nature* 1–9 (2025) doi:10.1038/s41586-024-08506-6.
15. Brauner, A., Shoresh, N., Fridman, O. & Balaban, N. Q. An Experimental Framework for Quantifying Bacterial Tolerance. *Biophysical Journal* **112**, 2664–2671 (2017).
16. Rominski, A., Schulthess, B., Müller, D. M., Keller, P. M. & Sander, P. Effect of β -lactamase production and β -lactam instability on MIC testing results for Mycobacterium abscessus. *J. Antimicrob. Chemother.* **72**, 3070–3078 (2017).
17. Cokol, M., Kuru, N., Bicak, E., Larkins-Ford, J. & Aldridge, B. B. Efficient measurement and factorization of high-order drug interactions in Mycobacterium tuberculosis. *Sci Adv* **3**, e1701881 (2017).
18. Katzir, I., Cokol, M., Aldridge, B. B. & Alon, U. Prediction of ultra-high-order antibiotic combinations based on pairwise interactions. *PLoS Comput. Biol.* **15**, e1006774 (2019).
19. Wald-Dickler, N., Holtom, P. & Spellberg, B. Busting the Myth of “Static vs Cidal”: A Systemic Literature Review. *Clin. Infect. Dis.* **66**, 1470–1474 (2018).
20. Block, A. M. *et al.* Mycobacterium tuberculosis Requires the Outer Membrane Lipid Phthiocerol Dimycocerosate for Starvation-Induced Antibiotic Tolerance. *mSystems* **8**, e00699-22 (2023).
21. Lobritz, M. A. *et al.* Antibiotic efficacy is linked to bacterial cellular respiration. *Proc. Natl. Acad. Sci.* **112**, 8173–8180 (2015).
22. Kwan, B. W., Valenta, J. A., Benedik, M. J. & Wood, T. K. Arrested Protein Synthesis Increases Persister-Like Cell Formation. *Antimicrob. Agents Chemother.* **57**, 1468–1473 (2013).
23. Wang, Y. *et al.* Inactivation of TCA cycle enhances Staphylococcus aureus persister cell formation in stationary phase. *Sci. Rep.* **8**, 10849 (2018).
24. Santucci, P. *et al.* Intracellular localisation of Mycobacterium tuberculosis affects efficacy of the antibiotic pyrazinamide. *Nat Commun* **12**, 3816 (2021).